# A Systematic Evaluation of the Planning and Scheduling Abilities of the Reasoning Model o1

**Karthik Valmeekam**[*]                                              *kvalmeek@asu.edu*
*School of Computing and AI*
*Arizona State University*

**Kaya Stechly**[*]                                                  *khstechl@asu.edu*
*School of Computing and AI*
*Arizona State University*

**Atharva Gundawar**                                                *agundawa@asu.edu*
*School of Computing and AI*
*Arizona State University*

**Subbarao Kambhampati**                                               *rao@asu.edu*
*School of Computing and AI*
*Arizona State University*

**Reviewed on OpenReview:** *https://openreview.net/forum?id=FkKBxp0FhR*

## Abstract

OpenAI claims that their recent o1 (Strawberry) model has been specifically constructed and trained to escape the normal limitations of autoregressive Large Language Models (LLMs)–making it a new kind of model: a Large Reasoning Model (LRM)–and be generally capable of tackling procedural reasoning tasks. We present the first comprehensive evaluation of these models on the fundamental tasks of planning and scheduling. Previous research attempted to use LLMs' expressive generation capabilities to solve these problems, but met with only limited success. We fill in the gaps in this literature by testing a larger suite of state-of-the-art LLMs on a set of large benchmarks, and then use this as a baseline to evaluate o1-preview and o1-mini. We see that while they can offer significant *accuracy* improvements over LLMs, this single metric is misleading and incomplete, as LRM queries demand large and unpredictable costs and take significant amounts of time to complete. We provide a case study demonstrating that, at those same price points, other methods of inference time scaling can do just as well. We also show that, contrary to OpenAI's injunctions, o1's performance can be improved further by embedding it in compound systems that separately, but complementarily, scale inference time further. Finally, while the paper is focused on o1, we provide similar evaluations of a more recent (and open-weight) LRM – DeepSeek R1.

## 1 Introduction

Large Language Models (LLMs) have captured the imagination of the AI research community by demonstrating linguistic behaviors that no one expected text completion systems to possess. Their seeming versatility initially led to claims that they are equally capable of deliberative behaviors such as planning and scheduling, but they have been called into question by a steady stream of more systematic investigations (Valmeekam et al., 2023; Silver et al., 2022; Xie et al.; Zheng et al., 2024; Mirzadeh et al., 2024). In some sense, this

---

[*]Equal Contribution

shouldn't be too surprising—the autoregressive "next token prediction in constant time" nature of LLMs, which makes them so popular, leaves little scope for runtime deliberation.

There have been several attempts to elicit reasoning from LLMs while keeping their essential auto-regressive nature intact. Early attempts—such as Chain of Thought and fine-tuning—were not particularly successful in robustly teasing planning behaviors out of LLMs (Stechly et al., 2024a; Valmeekam et al., 2024; Dziri et al., 2024). Compound systems which combine multiple LLM calls with verifiers, solvers, or other symbolic components have shown more promise (Yao et al., 2023b;a; Kambhampati et al., 2024) and demonstrate the potential of scaling the amount of LLM compute used at inference time to the problem at hand.

Against this background comes the recent release of OpenAI's o1 (Strawberry) (OpenAI, 2024a), which OpenAI claims to have trained specifically for tackling complex reasoning tasks (OpenAI, 2024a;b). Unlike vanilla LLMs–which can roughly be viewed as approximate retrievers–o1 seems to have been trained to be an approximate reasoner, capable of scaling the amount of compute it uses depending on the query.[1] Following OpenAI[2], we draw a distinction between previous Large Language Models (LLMs) and o1, a Large Reasoning Model (or LRM), as its new (unknown) architecture, operation, and capabilities all seem to be fundamentally different from those of vanilla LLMs—during both the pre-training phase and inference time.

Properly evaluating this new kind of model will require new tools and evaluation metrics. In this paper, we comprehensively evaluate the performance of o1 models (specifically o1-preview and o1-mini) on established planning and scheduling benchmarks. As a step towards better metrics and a clearer understanding of the utility and capabilities of LRMs, we build extensions of these benchmarks. We analyze not just the raw accuracy but the cost-efficiency trade-offs between these models and a baseline—a compound system that utilizes an off-the-shelf LLM and a sound verifier (as in the LLM-modulo framework (Kambhampati et al., 2024)) in order to scale inference-time compute. Our paper covers three major areas:

**Extending Planning and Scheduling Benchmarks:**  We start with existing benchmarks to support baseline comparisons with autoregressive LLMs. For planning, we start with PlanBench (Valmeekam et al., 2024), which consists of both specific test sets and a suite of tools intended for evaluating language models on arbitrary planning domains that were part of International Planning Competitions (IPC, 1998). To evaluate scheduling capabilities, we test on TravelPlanner (Xie et al.), on the the domains from Natural Plan (Zheng et al., 2024), and on graph coloring problems (Stechly et al., 2024b). We also built extensions of these domains to more closely analyze the generalization capabilities of o1-preview and o1-mini as problem difficulty increases.

**Analyzing Inference Cost vs. Accuracy Tradeoffs:**  Current benchmarks designed to test the planning and scheduling capabilities of autoregressive LLMs do not explicitly take efficiency into consideration. As the time taken by a vanilla LLM to produce some output is only dependent on the length of that output, rather than the semantic content or difficulty of the instance, this has not previously mattered. However, when a system adaptively varies time and dollar cost in response to the input, it is crucial to check if the accuracy gains it makes are worth the cost and whether they can be replicated with other techniques when given equivalent resources. In general, we argue that new approaches to measuring LRM reasoning capabilities must take into account efficiency, cost, and guarantees. While OpenAI's announcement does admit that the inference costs for o1 will be higher than the prior autoregressive models, it doesn't give a clear picture of how high the costs can be, nor does it mention the remarkable and unpredictable variance in that cost. As third-party independent evaluators, we quantify the (rather steep) inference cost of these models in dollar terms, and more clearly quantify the trade-offs between LRMs and other ways to scale inference time compute. We find that, in some cases, our baseline (an LLM-Modulo (Kambhampati et al., 2024) approach with a much cheaper base LLM) can beat o1 performance when comparably scaled up.

---

[1]We speculate that the complete system learns to improve its ability to make appropriate Chain-of-Thought (CoT) moves useful for reasoning in a pretraining RL step with synthetic data, and does inference time prompt-specific rollouts; see Appendix B for the details. In other words, it may be an RL-trained system in the same vein as AlphaGo, but where the 'moves' being generated and evaluated are Chains of Thought. Note that this is a very different use of RL from that in RLHF, which can be seen as a fine tuning stage for an LLM that keeps the inference step unchanged.

[2]Per the blogpost announcing the model: "A new series of reasoning models [...] for complex reasoning tasks this is a significant advancement and represents a new level of AI capability. Given this, we are resetting the counter back to 1 and naming this series OpenAI o1."(OpenAI, 2024c)

**Complementing o1 with Other Inference-time Scaling Techniques:** OpenAI released o1-preview with an injunction to keep prompts "simple and direct", claiming that the system worked best when fed questions directly without external augmentation (OpenAI, 2024d). We test whether this is accurate. In particular, we investigate whether inference time scaling techniques that preceded o1 can be used in tandem with it—the idea being to scale further in different ways at inference time. Specifically, we adapt the generate-test style LLM-Modulo framework (Kambhampati et al., 2024) to LRMs. Our results show that such an LRM-Modulo approach can not only provide correctness guarantees, but also further improve LRM performance. In essence, LRMs can replace LLMs as significantly better–but still fallible–generators in the LLM-Modulo framework. This also shows that there is potential in combining multiple inference-time scaling approaches at once when pursuing higher performance.

**Evaluations on DeepSeek R1:** The bulk of this study was done in Fall 2024, right after the release of the "Strawberry" o1 models. Since then, there have been other large reasoning models (see (Kambhampati et al., 2025) for an overview). Of these, DeepSeek's R1 is of particular interest as it is an open weight model with an accompanying full technical report about its internal operations (Guo et al., 2025). We have recently completed our PlanBench evaluations on R1 too, and share those results in Section 6.

## 2 Domains

### 2.1 Planning

We focus on classical planning problems, or STRIPS planning problems, which are a formalism for automated planning in discrete, deterministic spaces. To define a planning problem, we specify an *initial state*, a *domain*, and a *goal*. The domain contains all relevant information about the types of objects that may exist and the allowable actions from any given state, specified by defining the preconditions and effects of each named action. Problems and domains are represented in the flexible PDDL (Planning Domain and Definition Language) framework (McDermott et al., 1998). Solutions to PDDL problems are correct plans–sequences of actions executable from the initial state which arrive at a goal-satisfying final state. These are problems in which the the planner already knows all relevant facts about the world and the possible actions; thus only deliberation is required.

The STRIPS planning evaluation pipeline first introduced in Valmeekam et al. (2024) provides an extensible suite of domains and tools for evaluating language models, a collection of static test sets across these domains, and ways of easily generating additional problems. We draw on these static test sets to provide our initial o1 planning results on Blocksworld and Logistics–commonsense domains from the International Planning Competition (IPC, 1998)–and on Mystery Blocksworld–an obfuscated version of the same.

However, these domains are insufficient for a thorough evaluation of the limitations of o1. We construct new obfuscations of this benchmark. This can be seen as a private test set, generated post-training as a light test of pre-training contamination. Randomized Mystery Blocksworld consists of the same problems as the original Blocksworld test set, but with action and predicate names replaced by randomly generated strings. Randomized Mystery Logistics is constructed in the same way. We also extend the Blocksworld test set by adding 110 harder instances. Problems in this set range from 6 to 20 blocks in length and require 20 to 40 step optimal plans.

While STRIPS planning is in principle PSPACE-complete, the problems presented above are actually of a lower computational complexity. Plan existence for both Blocksworld and Logistics is polynomial (Hoffmann et al., 2006). We add Sokoban, a non-ergodic domain which is known to be PSPACE-complete (Culberson, 1997). In this domain, an agent moves around a constrained grid, pushing boxes by running into them, where the goal is to move every box to one of a set of final locations. As it can be represented in PDDL form, it is amenable to classical planning techniques.

Using a generator from the 2008 International Planning Competition(IPC, 1998), we generate 55 Sokoban instances with grid sizes ranging from $4 \times 4$ to $10 \times 10$, 1 to 4 boxes, and 1 to 4 walls. We then create corresponding prompts in PDDL and test o1 models on them.

Table 1: Performance on 600 instances from the Blocksworld and Mystery Blocksworld domains across large language models from different families[3], using both zero-shot and one-shot prompts. Best-in-class accuracies are bolded.

| Domain | Shots | Claude Models | | OpenAI GPT-4 Models | | | | LLaMA Models | | Gemini Models | |
|---|---|---|---|---|---|---|---|---|---|---|---|
| | | 3.5 Sonnet | 3 Opus | 4o | 4o mini | 4 | 4 Turbo | 3.1 405B | 3 70B | 1.5 Pro | 1 Pro |
| **Blocks world** | One Shot | **346/600 (57.6%)** | 289/600 (48.1%) | 170/600 (28.3%) | 49/600 (8.1%) | 206/600 (34.3%) | 138/600 (23%) | 284/600 (47.3%) | 76/600 (12.6%) | 101/600 (16.8%) | 68/600 (11.3%) |
| | Zero Shot | 329/600 (54.8%) | 356/600 (59.3%) | 213/600 (35.5%) | 53/600 (8.8%) | 210/600 (34.6%) | 241/600 (40.1%) | **376/600 (62.6%)** | 205/600 (34.16%) | 143/600 (23.8%) | 3/600 (0.5%) |
| **Mystery Blocks world** | One Shot | 19/600 (3.1%) | 8/600 (1.3%) | 5/600 (0.83%) | 0/600 (0%) | **26/600 (4.3%)** | 5/600 (0.83%) | 21/600 (3.5%) | 15/600 (2.5%) | - | 2/500 (0.4%) |
| | Zero Shot | 0/600 (0%) | 0/600 (0%) | 0/600 (0%) | 0/600 (0%) | 1/600 (0.16%) | 1/600 (0.16%) | **5/600 (0.8%)** | 0/600 (0%) | - | 0/500 (0%) |

## 2.2 Scheduling

We also evaluate o1 on scheduling problems. Zheng et al. (2024)'s Natural Plan benchmark consists of three domains: trip planning, calendar scheduling, and meeting planning. Xie et al.'s Travel Planning benchmark consists of a large dataset of travel information (flights, accommodations, restaurants, etc.) with prompts that ask the model to create a three to seven day itinerary based on natural language instructions. Note that, despite the word "planning" in these titles, these benchmarks are better characterized as testing scheduling abilities (Ghallab et al., 2016), which are generally much easier than planning and, at worst, only NP-Hard (Carlier, 1982). They are also equivalent to constraint satisfaction problems, such as graph coloring. Stechly et al. (2024b) translate graph coloring into natural language prompts and evaluate GPT-4's accuracy on them. We use their test set to evaluate o1.

We also extend this set to more difficult instances. Using the Erdős–Rényi procedure with $p = 0.4, n = 20$, we generated 50 more graphs, and did not rejection sample for planarity. As in that paper, we precalculate the chromatic number and provide it in the prompt, asking that the model produce a coloring that uses exactly that number of colors. These harder graphs have 20 vertices and around 60 edges each.

## 3 Preliminaries: Auto-regressive LLM's Still Can't Plan

Previous work has already shown that autoregressive LLMs have considerable difficulty on planning (Valmeekam et al., 2024; Silver et al., 2022) and scheduling (Xie et al.; Zheng et al., 2024) benchmarks. However, each piece of previous work evaluated on a different subset of available state-of-the-art LLMs or on previous, no longer relevant models. To provide a clean baseline, we fill in the missing results.

Table 1 shows the results of running current and previous generation LLMs on a static test set of 600 three to five block Blocksworld problems, as well as on a set of 600 semantically identical but syntactically obfuscated instances which Valmeekam et al. (2024) calls Mystery Blocksworld. Across the models tested, the best performance on regular Blocksworld is achieved by LLaMA 3.1 405B with 62.6% accuracy. Despite containing identical underlying problems, Mystery Blocksworld performance lags far behind–no LLM achieves even 5% on our test set–and performance on one version of the domain does not clearly predict performance on the other.

Vanilla language models typically perform better when tested on natural language versions of prompts rather than PDDL (Valmeekam et al., 2024), even though natural language can introduce uncertainty due to polysemanticity and syntactic ambiguity. To make our comparisons "fair" for the models being tested, the

---

[3]We do not provide Mystery Blocksworld data for Gemini 1.5 Pro only because we haven't been able to generate it. The model refuses to produce any output, instead claiming that responding to these queries would be harmful. We include this output in Appendix F.

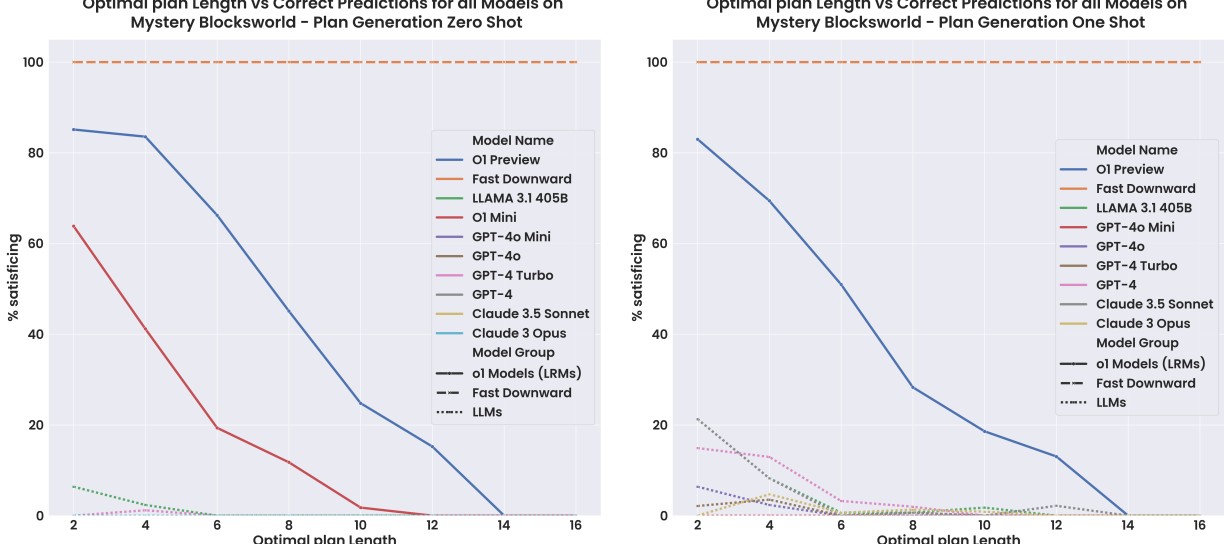

Figure 1: These examples are on Mystery Blocksworld. Fast Downward, a domain-independent planner (Helmert, 2006) solves all given instances near-instantly with guaranteed perfect accuracy. LLMs struggle on even the smallest instances. The two LRMs we tested, o1-preview and o1-mini, are surprisingly effective, but this performance is still not robust, and degrades quickly with length.

results we report are the higher percent accuracy natural language prompting numbers, and can be considered an upper bound on LLM performance on PDDL-specified problems.

The lackluster performance of LLMs on even the easiest static test set leads us to continue to believe that planning cannot be generally and robustly solved by approximate retrieval alone. We relegate additional discussion of the particulars of LLM performance to appendix A.

There have also been works that have attempted to improve the planning performance of LLMs through methods like prompt engineering (Wei et al., 2022; Yao et al., 2023b), self reflection (Huang et al., 2022; Shinn et al., 2024), repeated sampling combined with self-reflection (Hao et al., 2023; Yao et al., 2023a; Hu et al.) among others. However, systematic evaluations (Stechly et al., 2024b;a; Dziri et al., 2024; Prabhakar et al., 2024; Verma et al.) have shown that these methods are brittle and fail to enable LLMs to perform procedural reasoning generally and effectively.

Rather than relying on LLMs to generate solutions, some works have integrated LLMs with external planners. In these settings, LLMs handle tasks like translation (Liu et al., 2023; Dagan et al., 2023; Gestrin et al.) or knowledge acquisition (Guan et al., 2023; Caglar et al., 2024; Tang et al., 2024) while the solvers uses the generated specification to provide solutions. Given that LLMs provide completions without guarantees for almost any problem presented to them, works have explored using LLMs to not only generate plans or models but also search components that can be used as part of planning (Katz et al., 2024) or domain specific plan programs that would produce plans for tasks within the domain (Silver et al., 2024).

## 4 From Approximate Retrieval to Approximate Reasoning

Exact details about o1 are currently sparse, and so we can only speculate about its exact mechanisms. Our best guess is that there are two major differences between this model and vanilla LLMs: an additional reinforcement learning pre-training phase (perhaps to learn the q-values (Watkins, 1989) of different CoTs from massive amounts of synthetic data) and a new adaptively scaling inference procedure (maybe it further refines learned q-values by something like rollout before selecting a particular CoT; see Appendix B).

Table 2: Performance and average time taken on 600 instances from the Blocksworld, Mystery Blocksworld and Randomized Mystery Blocksworld domains and 200 instances from the Logistics and Randomized Logistics domains by OpenAI's o1 family of large reasoning models and satisficing Fast Downward (F.D.)

| Total Instances | Domain | Shots | Instances correct | | | Average Time Taken (in secs) | | |
|---|---|---|---|---|---|---|---|---|
| | | | o1 preview | o1 mini | F.D. | o1 preview | o1 mini | F.D. |
| 600 | **Blocksworld** | Zero Shot | 97.8% | 56.6% | 100% | 40.43 | 10.84 | 0.12 |
| 600 | **Mystery Blocksworld** | One Shot | 41.6% | - | 100% | 82.03 | - | 0.12 |
| | | Zero Shot | 52.8% | 19.1% | 100% | 83.37 | 35.54 | 0.12 |
| 600 | **Randomized Mystery Blocksworld** | Zero Shot | 37.3% | 3.5% | 100% | 111.11 | 55.40 | 0.12 |
| 200 | **Logistics** | Zero Shot | 94% | - | 100% | 84.07 | - | 0.13 |
| 200 | **Randomized Mystery Logistics** | Zero Shot | 52% | - | 100% | 167.41 | - | 0.13 |
| 55 | **Sokoban** | Zero Shot | 12.73% | 10.9% | 100% | 147.98 | 32.31 | 89.65 |

Regardless, what is clear from the detail available is that this model is fundamentally different in nature from previous LLMs. Rather than doing merely approximate retrieval—executing some kind of mildly-generalizing pattern-matching over mixtures of pre-training data—it seems to be doing something closer to approximate reasoning, which necessarily requires the ability to scale inference time. We systematically evaluate the performance difference this provides on planning and scheduling tasks.

## 4.1 Planning

**Evaluating LRMs on PlanBench:** We test o1-preview and o1-mini on the original 600-instance PlanBench test set.[4] The full results can be seen in Table 2. These 600 Blocksworld instances range from 3 to 5 blocks, and require plans of between 2 to 16 steps to solve. Far surpassing any LLM, o1 correctly answers 97.8% of these instances. On Mystery Blocksworld, the model does not maintain this level of performance, but it does far surpass all previous models (which barely managed a few percent), answering 52.8% of the questions correctly. To test whether the exact obfuscation might be compromised because of data contamination, we also generated a new obfuscation using completely random strings, and presented these problems in a new, semantically equivalent prompt format with fully specified and unambiguous PDDL descriptions of both the domain and problem. This is presented in the table as Randomized Mystery Blocksworld. Exact prompts can be seen in the appendix. While performance did dip further, 37.3% of instances are answered correctly, sharply contrasting the flat zeroes of previous models. The same pattern can be seen when evaluating Logistics and a freshly generated obfuscation of that domain. Despite the higher branching factor of the domain, o1-preview solves 94% of all 200 problems tested and achieves 52% on the obfuscated variant.

**Increasing Problem Size:** On the harder Blocksworld set we constructed for this paper, we see performance quickly degrade from the 97.8% reported earlier to 23.63%. Most of this accuracy comes from correctly solving problems which require fewer than 28 steps. While these models are overall impressive, this shows that their performance is still far from robust, especially in comparison to classical solvers as we shall see. These results are collated together with a representative sampling of smaller instances in Figures 1 and 2, showing how performance begins to fall on problems requiring plans of more than 10 steps.

---

[4]While for previous models, the model itself enforced the desired plan format, some modifications had to be made to accurately test o1's abilities. In its current form, o1-preview does not always conform to explicit formatting restrictions. In order to extract the generated plans, we used GPT-4o-mini to translate them into PDDL, and wrote a small Python parser to strip any remaining extraneous symbols before evaluating each proposed plan.

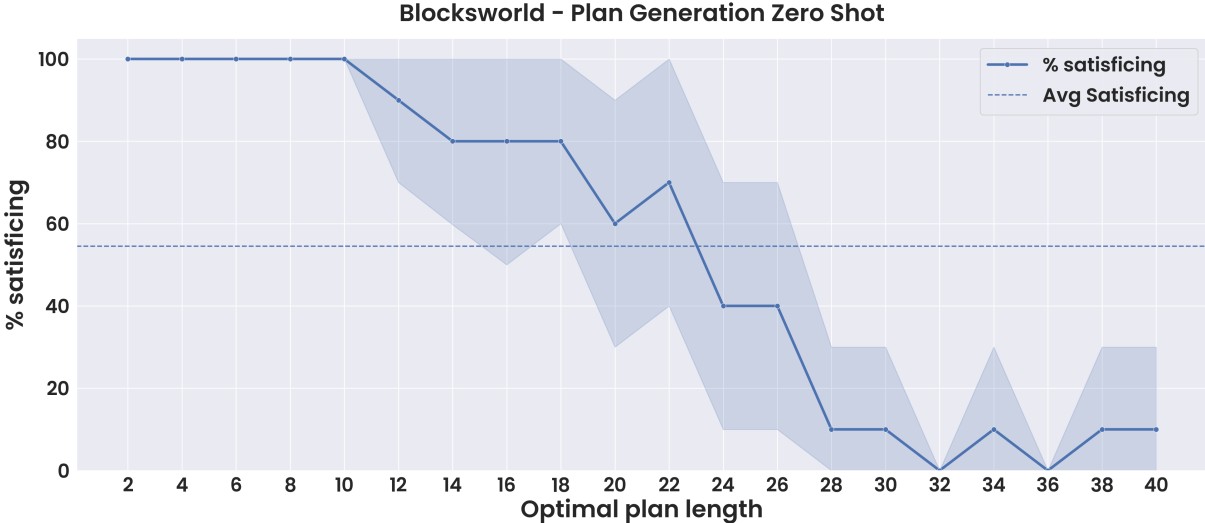

Figure 2: Extending even the (regular, not obfuscated) Blocksworld dataset to problems requiring greater numbers of steps worsens the performance of o1-preview. When tested on 110 instances which each require at least 20 steps to solve, it only manages 23.63%.

**A More Difficult Domain: Sokoban**   We also test on the 55 Sokoban instances that we generated as part of our extension of Planbench. o1-preview answers 12.7% of these instances correctly, while o1-mini is not far behind with 10.9%. However, the only problems on which these models succeed are those in which there is only one box on the grid. For comparison, when evaluated on these same instances, Llama3.1-405B, despite doing the best of all LLMs on the Blocksword sets, does not answer a single question correctly.

**o1's Creative Justifications**   While our main focus has been on providing a quantitative evaluation of o1-mini and o1-preview's performance on PlanBench, we have also noticed an idiosyncrasy that is worth commenting on. When the model gives an incorrect answer, it also sometimes provides a creative, but nonsensical, justification for its decision. It is almost as if o1 has gone from *hallucinating* to *gaslighting*! In one case, it decided that a problem was solved because a goal condition, while not present in the final state, had been true at some point during the execution, and thus should continue to count. In another, it declared that `on(a,c)` was true because, as it explained in a brief parenthetical, a was on b which was on c, and was thus a was somewhere above c, which should count as being "on top" of it. To avoid this, we changed prompts from natural language to PDDL in order to make it extremely clear that divergences from our exact definitions are disallowed.

Table 3: Error analysis of o1-preview

| Total Instances | Domain | Valid plans | Inexecutable plans | Non goal-reaching plans |
|---|---|---|---|---|
| 600 | Blocksworld | 97.83% | 2% | 0.17% |
| 600 | Mystery Blocksworld | 52.83% | 33.67% | 13.50% |
| 600 | Randomized Mystery Blocksworld | 37.33% | 41.50% | 21.17% |
| 200 | Logistics | 94% | 6% | 0% |
| 200 | Randomized Mystery Logistics | 52% | 15% | 33% |
| 55 | Sokoban | 12.73% | 67.27% | 20% |

Table 4: Error analysis of o1-mini

| Total Instances | Domain | Valid plans | Inexecutable plans | Non goal-reaching plans |
|---|---|---|---|---|
| 600 | Blocksworld | 56.67% | 39.50% | 3.83% |
| 600 | Mystery Blocksworld | 19.17% | 57.67% | 23.33% |
| 600 | Randomized Mystery Blocksworld | 3.50% | 19% | 77.50% |
| 55 | Sokoban | 10.9% | 74.54% | 14.5% |

Table 5: Cost per 100 instances (in USD). LRMs are significantly more expensive than LLMs.

| Large Language Models | | | | | | | Large Reasoning Models | |
|---|---|---|---|---|---|---|---|---|
| Claude 3.5 (Sonnet) | Claude 3 (Opus) | GPT 4o | GPT 4o mini | GPT 4 | Gemini 1.5 Pro | Gemini 1 Pro | o1 preview | o1 mini |
| $0.44 | $1.70 | $0.65 | $0.02 | $1.80 | $0.33 | $0.03 | $42.12 | $3.69 |

**Qualitative Analysis Of Errors**   For a closer look at the kind of errors the models make, we also examined what kinds of invalid plans o1-preview and o1-mini generate (see Tables 3 and 4). The invalidity of a plan could be either due to an action having unsatisfied preconditions or the plan not reaching the desired goal state after execution. As shown in the tables, the majority of invalid plans generated by o1-preview and o1-mini are invalid because they are inexecutable.

## 4.2   Scheduling

We also evaluate o1-mini and o1-preview on a set of scheduling problems that have previously been used in testing LLM capabilities.

**Graph Coloring:**   We evaluated o1-mini on the set of 220 problems included in the codebase associated with Stechly et al. (2024b)'s paper, and found that it solved 96%, surpassing the 16% reported by that paper for GPT-4. On the extended graph coloring test set we created for this paper, consisting of 50 graphs with 20 vertices each, o1-mini solves 50%, while o1-preview solves 64%.

**Travel Planning:**   We test o1 models on the 180 instance validation set of the sole-planning mode. In this mode, the model is provided upfront with all relevant information required to construct the requested itinerary. The previous state-of-the-art in direct prompting was 4.4%, achieved by GPT-4-turbo. o1-preview surpasses this solving 10% of all instances. o1-mini does not beat GPT-4-turbo, remaining at 1.67%.

**Natural Plan:**   This benchmark consists of three domains: calendar scheduling, trip planning, and meeting planning. On calendar scheduling, o1-mini solves 94% of all instances correctly. Given cost constraints, we did not test o1-preview on this domain, as o1-mini tends to be a lower bound on its performance. Neither model performs well on trip planning–o1-preview reaches 4% and o1-mini only 1%, both lower than the previous state-of-the-art set by Gemini 1.5 Pro. When provided with prompts from the meeting planning domain, both o1 models refuse to respond and flag the input as a potential terms violation (see Appendix G).

## 4.3   Accuracy/Cost Tradeoffs and Guarantees

With LRMs showing better performance on planning and scheduling problems, our evaluations must explicitly take into account the trade-offs that come from choosing general models over established deep and narrow systems. While in most of our domains o1-preview provides higher accuracy than LLMs, it still fails to

provide any correctness guarantees, and it is unclear that it is at all cost-effective. Unlike previous models, whose APIs only charge based on the number of input tokens and the number of output tokens (usually at a rate that is five times higher for the latter), o1's price-per-call includes a surcharge based on the number of "reasoning tokens"–tokens generated as part of inference and not revealed to the user–it used, which are charged at significantly higher rate. Currently, end users have no control over the number of these tokens generated, a number which is expanded or limited by the model in its own opaque way.

Without exposing the ability to scale inference time to particular specifications, influence the internal 'thinking' process in task-specific ways, or ensure that intermediate steps are evaluated by trusted or sound verifiers, the o1 models are a coarse-grained choice in the space of cost, inference time, guarantees, and performance trade-offs. They aren't, however, the only choices in that space, and reasonable LRM evaluations must take this into account (see similar arguments in Katz et al. (2024) and Kapoor et al. (2024)).

Classical planners like Fast Downward (Helmert, 2006) achieve 100% on our dataset in a fraction of the time, compute, and cost, while providing *guarantees that their answers are correct.* Running Fast Downward on a personal computer was essentially free in dollar terms and averaged 0.12 seconds per instance, which is many orders of magnitude faster than the average o1 clock times listed in table 2. It is also generally predictable, and can be scaled to harder instances very directly. Vanilla LLMs are typically very good at translating problems between formats, and could be used to do so in concert with a classical planner at a significantly lower cost than querying an LRM (e.g. Olmo et al. (2021); Liu et al. (2023)). For problems which don't have simple PDDL domain and instance specifications, LLM-Modulo systems may be a safer and cheaper approach: run a smaller, faster LLM in a loop with a sound verifier, so that the combined system will only output guaranteed correct solutions (e.g. Kambhampati et al. (2024); Romera-Paredes et al. (2024); Trinh et al. (2024)).

The correctness guarantees provided by these latter two methods are sorely lacking in LRMs like o1. A general reasoning system cannot be deployed in safety critical and non-ergodic domains if it continues to confidently make incorrect plans. Even more than previous models, o1 is a fully black box system, and OpenAI's decision to not only keep the architecture under wraps and hide the reasoning traces, but to warn away and even ban anyone who attempts to understand what is going on inside (Edwards, 2024), makes interpretability nearly impossible, and reduces trust in the system overall.[5]

## 5 LRM-Modulo: Inference Scaling Techniques Can Be Combined

Contrary to OpenAI's claims that o1 models work best when prompted simply and without external augmentation (OpenAI, 2024d), we find that the (unknown) inference time scaling o1 executes automatically can in fact be combined with other ways of adaptively increasing LLM compute. The particular method that we investigate is an adaption of Kambhampati et al. (2024)'s LLM-Modulo to LRMs, which we call LRM-Modulo.

We augment o1-preview and o1-mini with external verifiers to endow the combined system with soundness guarantees—this compound system also allows for some degree of control over the total compute used, by just letting the user set limits enforceable between backprompts. While o1 is a stride in the direction of general-purpose, expressive planning systems, our results have shown that it cannot plan robustly when faced with harder instances (see Figure 2). In other words, o1 models, just like LLMs, do not come with guarantees for their outputs. Prior to these models, the best way to coax planning capabilities out of LLMs was to pair them with sound external verifiers in generate-test frameworks, in what are known as LLM-Modulo systems (Kambhampati et al., 2024; Trinh et al., 2024). This framework is broadly applicable even beyond LLMs, and–given a sound verifier for some domain–requires only a generator expressive enough to provide guesses for that domain. Moreover, because of the built-in verification, it guarantees that any answer actually output is correct. For safety-critical systems, this is essential. High accuracies are not sufficient, especially when the underlying system–as is the case for LLMs and even more so for LRMs–is an opaque black box.

---

[5]The current model is also set to a default temperature of 1.0, which further reduces replicability and interpretability–for any given problem, it is never clear whether the result is merely the result of stochasticity. This compounds a problem with OpenAI models that has existed since at least GPT3. Temperature 0 never gave deterministic outputs, and worse, the logprobs provided by the OpenAI API for any given prompt have long been known to fluctuate wildly (Xuan, 2023).

Table 6: Performance of LRM-modulo with o1-preview as the underlying LRM on our hardest test sets: Blocksworld (20+ length plans), Mystery Blocksworld, Sokoban, Graph Coloring (20 vertex), Travel Planning, and Trip Planning (10 cities). Due to cost constraints, we run each problem set for up to ten iterations, stopping early if the improvement per iteration levels off. For all the planning domains prompts were in PDDL.

| Total Instances | Domain | Direct o1-preview | | LRM (o1-preview) Modulo | | |
|---|---|---|---|---|---|---|
| | | % correct | Cost (in $) | % correct | # of iters | Cost (in $) |
| 110 | Blocksworld (hard) | 23.65% | 52.12 | 98.2% | 7 | 139.36 |
| 600 | Mystery Blocksworld | 64.3% | 241.2 | 96% | 6 | 489.21 |
| 55 | Sokoban | 12.73% | 42.89 | 43.6% | 7 | 203.38 |
| 50 | Graph Coloring (hard) | 64% | 26.65 | 94% | 10 | 56.16 |
| 180 | Travel Planning | 10% | 68.21 | 65% | 10 | 431.75 |
| 200 | Trip Planning (10 cities) | 4% | 131.26 | 15.50% | 4 | 535.49 |

Table 7: Performance of LRM-modulo with o1-mini as the underlying LRM on our hardest test sets: Blocksworld (20+ length optimal plans), Sokoban, Graph Coloring (20 vertex), Travel Planning, and Trip Planning (10 cities).

| Total Instances | Domain | Direct o1-mini | | LRM (o1-mini) Modulo | | |
|---|---|---|---|---|---|---|
| | | % correct | Cost (in $) | % correct | # of iters | Cost (in $) |
| 110 | Blocksworld (hard) | 0.90% | 16.30 | 10% | 4 | 40.80 |
| 55 | Sokoban | 10.90% | 5.28 | 12.70% | 4 | 17.49 |
| 50 | Graph Coloring (hard) | 50% | 3.33 | 84% | 15 | 14.18 |
| 180 | Travel Planning | 1.67% | 12.11 | 41.11% | 10 | 97.17 |
| 200 | Trip Planning (10 cities) | 1% | 20.01 | 3.50% | 4 | 77.71 |

Generate-test systems are limited by how good the generator is. A poor but complete generator, such as one that produces completely random strings, may be capable of eventually producing the correct answer, but be so unlikely to do so that it is functionally useless, while an incomplete generator may be incapable of ever outputting the correct answer. LLMs and LRMs can be backprompted–that is, we can take feedback from the sound verifier and send it back to the model or modify the next prompt in some other way to increase the diversity of the responses generated–which may steer their next output towards the correct answer. Based on our results, o1 models are much better generators than anything that came before them.

We test LRM-modulo setups on our five hardest test sets: 20+ length plan Blocksworld, Sokoban, 20 vertex graph coloring, OSU's Travel Planning, and 10 city trip planning. Due to cost constraints, we limit the number of iterations to a maximum of ten, but we stop the system early once the performance increase from round to round has become mostly flat. Even with so few iterations, we see significant jumps in performance across almost all of our domains. o1-preview's performance on harder Blocksworld saturates within 7 iterations, with the combined system achieving 98.2%. Harder graph coloring shows similar results, going up to 94%. Perhaps most surprising, our most difficult domain, Sokoban, shows a significant jump from 12.7% to 43.6%. o1-mini-Modulo performance, while also impressive, only comes close on graph coloring and travel planning. We present these results in Table 6.

Our results imply that these models are not only better generators, but also benefit more from the sound verification signal and provided feedback. However, we can't really know *how* they use the critique provided–

Table 8: Performance and average time taken on (extended) PlanBench by OpenAI's o1-preview and DeepSeek-R1. For all the domains prompts were in PDDL.

| Total Instances | Domain | Instances correct | | Avg. Time (in secs) | |
|---|---|---|---|---|---|
| | | o1-preview | DeepSeek-R1 | o1-preview | DeepSeek-R1 |
| 600 | **Mystery Blocksworld** | 64.3% | 54.4% | 68.14 | 230.87 |
| 600 | **Randomized Mystery Blocksworld** | 37.3% | 47.5% | 111.11 | 343.09 |
| 200 | **Logistics** | 94% | 88% | 84.07 | 153.39 |
| 200 | **Randomized Mystery Logistics** | 52% | 45.26% | 167.41 | 223.56 |
| 55 | **Sokoban** | 12.73% | 14.55% | 147.98 | 240.57 |

Table 9: Performance of LRM-modulo with DeepSeek-R1 as the underlying LRM on Blocksworld (20+ length plans), Mystery Blocksworld and Sokoban. For all the domains prompts were in PDDL.

| Total Instances | Domain | Direct DeepSeek-R1 | | LRM (DeepSeek-R1) Modulo | | |
|---|---|---|---|---|---|---|
| | | % correct | Cost (in $) | % correct | # of iters | Cost (in $) |
| 110 | Blocksworld (hard) | 53.64% | 1.89 | 87.27% | 5 | 3.29 |
| 600 | Mystery Blocksworld | 54.5% | 10.6 | 76.17% | 5 | 28.2 |
| 55 | Sokoban | 14.55% | 1.33 | 30.91% | 5 | 4.29 |

this question is crucial but unanswerable, given OpenAI's current stance against revealing the internal workings of the model or the intermediate reasoning tokens it generates.

**LLM-Modulo vs. LRM:** With the high cost of o1 model queries, it is also essential to examine in which situations these models are not just the best option, but the cost-effective one. Returning to LLM-modulo, where we use a smaller, cheaper, and faster LLM in a loop with a verifier, can provide similar or better performance in some domains. For example, in calendar scheduling, o1-mini costs $2.70 to run over the entire test set, and has a final accuracy of 94%. GPT-4o-mini can be run for 50 iterations in a modulo loop for only $2.48, beating that performance to get 97%.

# 6 PlanBench Evaluations on DeepSeek R1

While o1 seemed like a quantum jump in LLM research when it came out in Fall 2024, there have since been other commercial reasoning models. In Kambhampati et al. (2025), we provide an overview of these developments. Of these models, special mention must be made of DeepSeek R1, an open-weight model whose training regime and architecture are documented in Guo et al. (2025).

We've run the same PlanBench evaluations on this model. As we can see in Table 8, the results are qualitatively very similar. R1 is slightly better on some domains, and slightly worse on others – the major distinction is that it is also significantly better than vanilla LLMs. The bigger difference between the two models (at least when accessed through their respective APIs at the time of writing) is cost and time: R1 is significantly cheaper but noticeably slower, providing another point on the frontier we discussed in subsection 4.3.

We also ran the same LRM-modulo experiments with R1 in the role of LRM. These results are presented in Table 9, showing that significant performance gains can also be made with this newer model, and that combining inference-time scaling techniques is still very much a viable method of further improving performance.

# 7 Conclusion

In this paper, we investigated the performance of o1-preview and o1-mini–the new so-called LRMs–on a variety of planning and scheduling benchmarks. While LLMs have thus far failed to make much progress on obfuscated (or "Mystery") versions of PlanBench domains, these new models show the first bit of real progress. As o1 models are quite different from the previous LLMs, there is a lack of understanding about o1's performance and limitations on reasoning problems. Our systematic study–which starts with existing planning and scheduling benchmarks, but then constructs extensions to better evaluate o1-preview and o1-mini in places where they saturate the existing benchmarks, and carefully analyzes the cost/accuracy tradeoffs, will be a valuable contribution in understanding this new breed of models.

In general, o1-preview has made impressive headway on benchmarks that were previously unassailable. However, when evaluated on longer and harder problems the model's accuracy gains are neither general nor robust. While these models made some gains on scheduling problems, performing much better on graph coloring than previous models, these were not evenly distributed and were somewhat lackluster on OSU's Travel Plan domain and Google's Natural Plan benchmark suite.

We also discussed the critical accuracy/efficiency tradeoffs brought up by the fact that o1 models use (and charge for) significant inference-time compute, as well as how it compares to other inference time scaling approaches (such as LLM-Modulo (Kambhampati et al., 2024)) and dedicated solvers. Future evaluations will have to maintain a focus on these factors if they are to remain meaningful or relevant. We showed that multiple inference time scaling approaches can be combined, and provided a case study of constructing an LRM-Modulo system that further improved performance and, crucially, provided much-needed guarantees. Finally, while the bulk of the paper is focused on o1, we provided similar evaluations on DeepSeek R1, a more recent (and open-weight) LRM. The results for R1 are qualitatively very similar to our o1 results, including on the LRM-Modulo experiments.

# Acknowledgments

We acknowledge support from and spirited discussions with fellow Yochan lab members. Our research is supported in part by DARPA grant HR00112520016, ONR grant N0001423-12409, and a gift from Qualcomm.

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

# Appendix

## A Further Discussion of LLM Planning Performance

LLMs are highly capable at providing translations between equivalent representations (Olmo et al., 2021). This fact, combined with their significantly higher performance on the unobfuscated version of the Blocksworld domain, predicts that–if they are capable of composing reasoning operations–the performance gap between Mystery Blocksworld and classic Blocksworld should shrink substantially if the translation from Mystery Blocksworld back into Blocksworld is explicitly provided. However, when we provide this in the prompt (see Appendix I), performance only improves a very small amount: GPT-4 achieves 10%.

We also find that, contrary to previous claims, one-shot prompting is *not* a strict improvement over zero-shot. In fact, for many models it seems to do significantly worse. (This is most notable in our tests of LLaMA family models.) While the reverse is generally true for Mystery Blocksworld problems, it's important to note that the performance of vanilla LLMs on Mystery Blocksworld has consistently and uniformly been poor (the same as it was when this benchmark was first released), so those results do not provide too clear a picture. Most models do not solve even a single instance in zero-shot mode, and only one (LLaMA 3.1 405B) manages more than one.

## B Speculations about o1 internal operation

While our evaluation of o1 did not depend on any specific assumption about its operation, we did have a working model of o1 based on the very skimpy description that was provided in the blog post that accompanies o1's release (OpenAI, 2024c). Verifying our model is unfortunately made infeasible by the fact that o1 doesn't actually provide any trace of its operations (even during the costly inference stage), and OpenAI warns that API access will be revoked if any attempts are made to surface its reasoning tokens.

There are two things–"reinforcement learning" and "Private Chain-of-Thought (CoT)" that are mentioned in the writeup. So imagine you are trying to transplant a "generalized AlphaGo"–let's call it GPTGo–onto the underlying LLM token prediction substrate.

To do this, you need to know

1. What are the GPTGo moves? For AlphaGo, we had GO moves). What would be the right moves when the task is just "complete the prompt the right way"?

2. Where is it getting its external success/failure signal from? For AlphaGo, we had simulators/verifiers giving the success/failure signal. The most interesting question in transplanting the self-play idea to a general AI agent is where is it getting this signal?

Our guess is that the moves are auto-generated CoTs (thus the moves have a very high branching factor).[6] Let's assume–for simplification–that we have a CoT-generating LLM, that generates these CoTs conditioned on the prompt. (It is not clear if the CoT's are domain independent of the "think step by step" variety (Kojima et al., 2022) or domain/task specific, or a combination.)

The success signal is likely from massive amounts of synthetic training data with correct answers. When the completed prompt is seen to contain the correct answer (presumably judged by the LLM itself), then the episode is considered a success, and a failure otherwise.

The task for the reinforcement learner then is: Given the original problem prompt, generate and select a CoT, and use it to continue to extend the prompt (possibly generating subgoal CoTs after every few stages). Get the final success/failure signal for the example (for which you do have answer).

The RL stage may involve training on a a huge number of training examples with answers. The training examples with answers can either be coming from benchmarks, or from synthetic data with problems and

---

[6]Note that the moves here are "generic" language prompt extension moves. In this sense, they are closer to $\mu_0$ (Schrittwieser et al., 2019) that used generic "shift the piece on the board" moves applicable to all board games than Go or Chess specific moves.

their solutions–using external solvers. In this phase the RL part attempts to learn the q-values of the CoT moves (much like AlphaGo learns the q-values of the moves of the Go). (The q-values learning may be incorporated into the internal weights of the CoT generator LLM). At this point, we have a CoT move generator that is better than the random one before the RL stage

During the inference stage–which OpenAI says can be indefinitely long (although it is currently capped internally by them, with no external control), like AlphaGo, o1 might be further improving its evaluation of the q-values of the CoT moves in the context of the current prompt. While AlphaGo used MCT-based rollouts, we obviously don't know the mechanism o1 uses. The announcement only says that at inference stage a long chain of thought is added to the original prompt (and o1 does charge the end users for its "reasoning tokens," which are never seen by the end user, at the same high rate as the output tokens). In this sense, our speculations seem to be consistent, even though it is not clear whether the reasoning tokens are proportional to the entire inference-stage computation, or just represent the final sequence of CoT moves that get selected after the rollout-like inference stage.

Some corollaries of our speculation are:

1. Note that this use of RL is very different from that in RLHF, which can be seen as a fine tuning stage for an LLM that keeps the inference step unchanged. It is also different from techniques, including OpenAI's, that advocated fine tuning both on synthetic data accompanied with derivational traces–these too will be a form of finetuning that leave inference stage unchanged. Here o1 could, in theory, be getting significantly more leverage out of the data by learning move (auto CoT) generators.

2. There still are no guarantees that the answers provided are "correct"–they may be probabilistically a little more correct (subject to the training data). If you want guarantees, you still will need some sort of LLM-Modulo approach even on top of this.

3. It is certainly not clear that anyone will be willing to really wait for long periods of time during inference (it is already painful to wait for 10 sec for a 10 word last letter concatenation!). The kind of people who will wait for longer periods would certainly want guarantees–and there are deep and narrow System 2's a plenty that can be used for many such cases.

4. There is a bit of a *Ship of Theseus* feel to calling o1 an LLM–considering how far it is from the other LLMs (all of which essentially have teacher-forced training and sub-real-time next token prediction. That said, this is certainly an interesting way to build a generalized system 2'ish component on top of LLM substrates–but without guarantees.

### B.1   Comparison to DeepSeek R1

Our speculations about o1's operations were written in the fall of 2024. While OpenAI o1's internal operations have not been made public to-date, DeepSeek R1 is an open weight model whose internal operations were described in a technical report (Guo et al., 2025). Although we don't know how representative R1 is of o1, R1 does use reinforcement learning to improve the CoTs (intermediate tokens) on synthetic data, with the reward signal coming from the verifiers for that data. Different from our speculation on o1, R1 uses the base model itself to generate the initial CoTs (intermediate tokens). R1 also does not use any test-time inference/adaptive computation (as we speculated for o1); its operation at inference time is thus like that of a standard LLM, albeit one trained to produce intermediate tokens on the way to a solution. We invite the reader to see (Kambhampati et al., 2025) for a more detailed and up-to-date perspective on how, and whether, large reasoning models reason.

### C   Performance on Unsolvable Instances:

While planning problems normally require the agent to formulate a course of action to achieve a goal, an equally valid use of planning abilities is to recognize that a given goal *cannot* be accomplished by any plan. A real-world example of this is network vulnerability analysis, where an agent may wish to certify that no plan of attack exists for a specified system (Boddy et al., 2005). So far, LLMs have struggled to recognize that some

Table 10: Rate of claiming that a problem is impossible by OpenAI's o1-preview on 100 unsolvable and 600 solvable instances in the Blocksworld and Randomized Mystery Blocksworld domains. The True Negative rate is the percent of unsolvable instances that were *correctly* marked as unsolvable. The False Negative rate is the percent of solvable instances that were *incorrectly* marked as unsolvable. Previous models are not shown in this table as their true negative and false negative rates were generally 0% across the board.

| Domain | Shots | o1-preview | |
| --- | --- | --- | --- |
| | | True Negatives | False Negatives |
| **Blocksworld** | 0-Shot | 27% | 0% |
| **Randomized Mystery Blocksworld** | 1-Shot | 16% | 11.5% |

problems cannot be solved, instead confidently confabulating nonsensical answers. o1 was launched with the claim that it has started to overcome this issue, and can now accurately identify unsolvable problems (Brown, 2024). To test this systematically, we modified 100 instances from the easier three to five block test set by adding one `on(x,y)`-type conjunct to each instance's goal state, making the goal unsatisfiable. We ensured our instances were unambiguous by giving the full PDDL representation of both the domain and the instance, to avoid quibbles such as "A is on B because A is on C and C is on B" where the model redefines the meanings of a potentially ambiguous natural language statement. The results are in Table 10. On Blocksworld, only 27% of all instances were correctly and explicitly identified by o1 as unsolvable. In 19% of all cases, the model returned a dot or some kind of "[empty plan]" marker, without any explanation or indication of unsolvability. We consider these incorrect, as "empty plan" is only the correct answer if the goal is already satisfied. In the remaining 54% of cases, the model generated a full (and therefore impossible and incorrect!) plan.

On Randomized Mystery Blocksworld, these numbers are worse: 16% of cases were correctly identified as unsolvable, 5% returned an empty plan, and the remaining 79% were answered with a full (impossible or goal-unsatisfying) plan. Therefore, unsolvable instances continue to be a problem for LRMs. Furthermore, this ability to sometimes note impossible plans correctly comes at a cost: now the model sometimes falsely claims that solvable problems are actually unsolvable. On Randomized Mystery Blocksworld, 11.5% of instances are incorrectly claimed to be impossible. These results can be seen in Table 10.

## D    NL vs PDDL prompts

Table 11: Performance of o1-preview on natural language (NL) and PDDL prompts.

| Total Instances | Domain | Instances correct | |
| --- | --- | --- | --- |
| | | NL | PDDL |
| 600 | Mystery Blocksworld | 52.83% | **64.3%** |
| 230 | Randomized Obfuscated Blocksworld | 36.52% | **47.82%** |

In (Valmeekam et al., 2023), it was shown that LLMs performed better on natural language prompts as opposed to PDDL prompts for planning. With o1 models (specifically o1-preview), we see a reversed trend. As shown in Table 11, o1-preview performs better on PDDL prompts than natural language.

## E    LRM-Modulo performance over iterations

See Figure 3

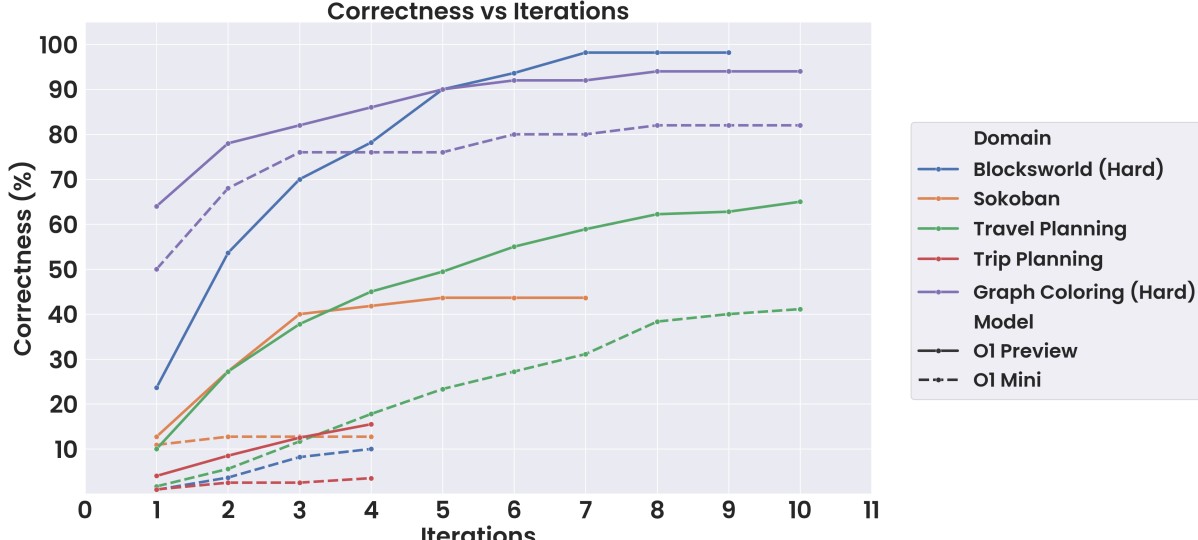

Figure 3: LRM-Modulo significantly improves performance over direct prompting as we increase the number of iterations.

## F  Gemini 1.5 Pro Response to Mystery Blocksworld

When using the default settings, Gemini 1.5 Pro refuses to provide output to our Mystery Blocksworld queries and instead returns the following:

```
finish_reason: SAFETY

safety_ratings {
  category: HARM_CATEGORY_SEXUALLY_EXPLICIT
  probability: NEGLIGIBLE
}
safety_ratings {
  category: HARM_CATEGORY_HATE_SPEECH
  probability: NEGLIGIBLE
}
safety_ratings {
  category: HARM_CATEGORY_HARASSMENT
  probability: NEGLIGIBLE
}
safety_ratings {
  category: HARM_CATEGORY_DANGEROUS_CONTENT
  probability: MEDIUM
}
```

## G  o1-preview and o1-mini Response to Meeting Planning

```
openai.BadRequestError: b'{
    "error": {
      "message": "Invalid prompt: your prompt was flagged as potentially violating our usage
          policy. Please try again with a different prompt.",
   "type": "invalid_request_error",
```

```
    "param": null,
    "code": "invalid_prompt"
  }
}'
```

## H o1 Token Use Versus Problem Difficulty

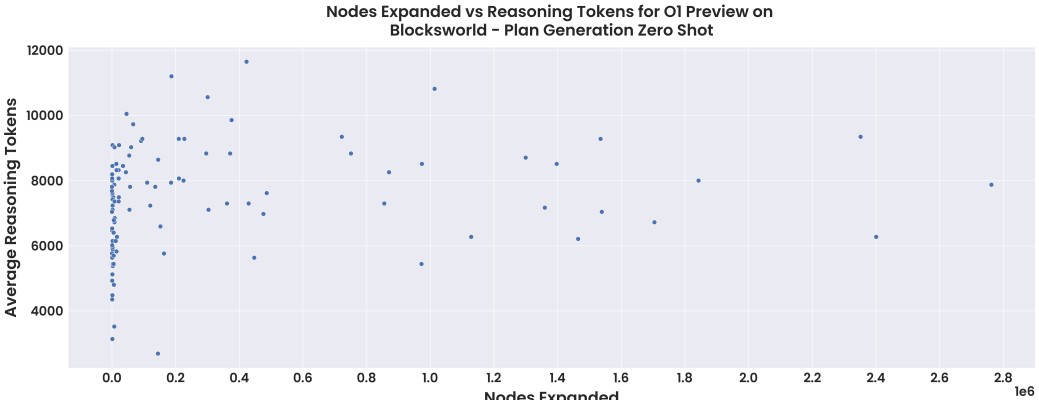

Figure 4: The number of reasoning tokens used by o1-preview when solving Blocksworld instances does not track the number of nodes that need to be expanded to solve the problem.

## I Prompt to Translate From Mystery Back to Blocksworld

I am playing with a set of objects. Here are the actions I can do

    Attack object
    Feast object from another object
    Succumb object
    Overcome object from another object

I have the following restrictions on my actions:
To perform Attack action, the following facts need to be true: Province object, Planet object,
    Harmony.
Once Attack action is performed the following facts will be true: Pain object.
Once Attack action is performed the following facts will be false: Province object, Planet object,
    Harmony.
To perform Succumb action, the following facts need to be true: Pain object.
Once Succumb action is performed the following facts will be true: Province object, Planet object,
    Harmony.
Once Succumb action is performed the following facts will be false: Pain object.
To perform Overcome action, the following needs to be true: Province other object, Pain object.
Once Overcome action is performed the following will be true: Harmony, Province object, Object
    Craves other object.
Once Overcome action is performed the following will be false: Province other object, Pain object.
To perform Feast action, the following needs to be true: Object Craves other object, Province
    object, Harmony.
Once Feast action is performed the following will be true: Pain object, Province other object.
Once Feast action is performed the following will be false:, Object Craves other object, Province
    object, Harmony.

You will be given a set of initial conditions and a goal condition. To solve the problem, you will
    have to tell me which actions to take and in which order in order to achieve the goal.

Please provide your answers using the above terminology. However, you may find it helpful to
    translate the above description into a common-sense format while working out your solution.
    Just remember to translate it back later!
Instead of thinking in terms of "objects", think in terms of different alphabet blocks (block A,
    block B, etc.) which you are stacking (using just one hand) in towers on a table.

Then the "facts" that are true or false at a given time are really facts about the blocks and the
    hand:
      "Province object a" just means that "block A is clear" or, equivalently, "nothing is on top of
          block A"
      "Planet object a" is another way of saying "block A is on the table"
      "Harmony" is a codeword for "my hand isn't holding anything"
      "Pain object a" = "the hand is holding block A"
      "object a Craves object b" translates to "block A is on top of block B"

And the "actions" can be seen as stacking and unstacking of blocks (where the restrictions stop us
    from picking up the bottom block in a tower or holding more than one block in the hand at a
    time):
      "Attack object a" translates to "pick up block A directly from the table"
      "Feast object a from object b" translates to "pick up block A from directly on top of block B"
      "Succumb object a" translates to "put block A directly on the table"
      "Overcome object a from object b" translates to "put block A directly on top of block B"

If you do use this framing, make sure to remember to translate back into the original terms.

## J   Prompts for Blocksworld

### J.1   (Solvable) Blocksworld Instances - Zero-Shot in Natural Language

I am playing with a set of blocks where I need to arrange the blocks into stacks. Here are the
    actions I can do

Pick up a block
Unstack a block from on top of another block
Put down a block
Stack a block on top of another block

I have the following restrictions on my actions:
I can only pick up or unstack one block at a time.
I can only pick up or unstack a block if my hand is empty.
I can only pick up a block if the block is on the table and the block is clear. A block is clear if
     the block has no other blocks on top of it and if the block is not picked up.
I can only unstack a block from on top of another block if the block I am unstacking was really on
    top of the other block.
I can only unstack a block from on top of another block if the block I am unstacking is clear.
Once I pick up or unstack a block, I am holding the block.
I can only put down a block that I am holding.
I can only stack a block on top of another block if I am holding the block being stacked.
I can only stack a block on top of another block if the block onto which I am stacking the block is
      clear.
Once I put down or stack a block, my hand becomes empty.
Once you stack a block on top of a second block, the second block is no longer clear.

[STATEMENT]
As initial conditions I have that, the red block is clear, the blue block is clear, the yellow
    block is clear, the hand is empty, the blue block is on top of the orange block, the red block
    is on the table, the orange block is on the table and the yellow block is on the table.
My goal is to have that the orange block is on top of the blue block.

What is the plan to achieve my goal? Just give the actions in the plan.

==============o1-preview RESPONSE================
To achieve your goal of stacking the orange block on top of the blue block, you can follow these
    steps:

1. **Unstack the blue block from on top of the orange block.**
   - Since the blue block is clear and your hand is empty, you can unstack it from the orange block.

2. **Put down the blue block on the table.**
   - With the blue block in your hand, you can place it onto the table.
3. **Pick up the orange block from the table.**
   - The orange block is now clear (no blocks on top of it), and your hand is empty after putting
       down the blue block, so you can pick it up.
4. **Stack the orange block on top of the blue block.**
   - The blue block is clear on the table, so you can stack the orange block you're holding onto it.

**Actions in the plan:**

- **Unstack** the blue block from the orange block.
- **Put down** the blue block on the table.
- **Pick up** the orange block from the table.
- **Stack** the orange block on top of the blue block.

### J.2 (Solvable) Harder Blocksworld Instances - Zero-Shot in PDDL

The following is a PDDL specification of a planning problem. The first part, under the heading [
    DOMAIN], is the domain file. The second part, under the heading [QUERY PROBLEM], is the problem
     file. Using this information, which is correct, and no further assumptions, find a plan which,
     when run from the specified initial state, satisfies the specified goal. Provide your answer
    as a sequence of actions in PDDL format. An action ACTION which acts on two objects OBJ1 and
    OBJ2 would be written (ACTION OBJ1 OBJ2). Do not provide anything else in your answer.
[DOMAIN]
(define (domain blocksworld-4ops)
  (:requirements :strips)
(:predicates (clear ?x)
           (ontable ?x)
           (handempty)
           (holding ?x)
           (on ?x ?y))

(:action pick-up
  :parameters (?ob)
  :precondition (and (clear ?ob) (ontable ?ob) (handempty))
  :effect (and (holding ?ob) (not (clear ?ob)) (not (ontable ?ob))
              (not (handempty))))

(:action put-down
  :parameters (?ob)
  :precondition (holding ?ob)
  :effect (and (clear ?ob) (handempty) (ontable ?ob)
              (not (holding ?ob))))

(:action stack
  :parameters (?ob ?underob)
  :precondition (and (clear ?underob) (holding ?ob))

```
  :effect (and (handempty) (clear ?ob) (on ?ob ?underob)
               (not (clear ?underob)) (not (holding ?ob)))))

(:action unstack
  :parameters (?ob ?underob)
  :precondition (and (on ?ob ?underob) (clear ?ob) (handempty))
  :effect (and (holding ?ob) (clear ?underob)
               (not (on ?ob ?underob)) (not (clear ?ob)) (not (handempty))))))

[QUERY PROBLEM]
(define(problem BW-rand-6)
(:domain blocksworld-4ops)
(:objects a b c d e f )
(:init
(handempty)
(ontable a)
(on b f)
(on c a)
(on d b)
(on e c)
(on f e)
(clear d)
)
(:goal
(and
(on a f)
(on b c)
(on c a)
(on d b)
(on f e))
)
)
)

[PLAN]

=============o1-preview RESPONSE===============
(unstack d b)
(put-down d)
(unstack b f)
(stack b c)
(pick-up d)
(stack d b)
(unstack f e)
(put-down f)
(unstack e c)
(put-down e)
(pick-up f)
(stack f e)
(unstack c a)
(put-down c)
(pick-up a)
(stack a f)
(pick-up c)
(stack c a)
(pick-up b)
(stack b c)
(pick-up d)
(stack d b)
```

### J.3   (Solvable) Harder Blocksworld Instances - Backprompt

```
The above plan is invalid. The following action at step 4 has an unmet precondition:
stack b c
The unmet precondition is:
(clear c)
Provide your revised plan as a sequence of actions in PDDL format. An action ACTION which acts on
    two objects OBJ1 and OBJ2 would be written (ACTION OBJ1 OBJ2). Do not provide anything else in
    your answer.
```

```
[PLAN]
```

### J.4   Unsolvable Blocksworld Instances - Zero-Shot in PDDL

```
The following is a PDDL specification of a planning problem. The first part, under the heading [
    DOMAIN], is the domain file. The second part, under the heading [QUERY PROBLEM], is the problem
     file. Using this information, which is correct, and no further assumptions, find a plan which,
     when run from the specified initial state, satisfies the specified goal. Provide your answer
    as a sequence of actions in PDDL format. An action ACTION which acts on two objects OBJ1 and
    OBJ2 would be written (ACTION OBJ1 OBJ2). Do not provide anything else in your answer.
[DOMAIN]
(define (domain blocksworld-4ops)
  (:requirements :strips)
(:predicates (clear ?x)
            (ontable ?x)
            (handempty)
            (holding ?x)
            (on ?x ?y))

(:action pick-up
  :parameters (?ob)
  :precondition (and (clear ?ob) (ontable ?ob) (handempty))
  :effect (and (holding ?ob) (not (clear ?ob)) (not (ontable ?ob))
              (not (handempty))))

(:action put-down
  :parameters (?ob)
  :precondition (holding ?ob)
  :effect (and (clear ?ob) (handempty) (ontable ?ob)
              (not (holding ?ob))))

(:action stack
  :parameters (?ob ?underob)
  :precondition (and (clear ?underob) (holding ?ob))
  :effect (and (handempty) (clear ?ob) (on ?ob ?underob)
              (not (clear ?underob)) (not (holding ?ob))))

(:action unstack
  :parameters (?ob ?underob)
  :precondition (and (on ?ob ?underob) (clear ?ob) (handempty))
  :effect (and (holding ?ob) (clear ?underob)
              (not (on ?ob ?underob)) (not (clear ?ob)) (not (handempty)))))

[QUERY PROBLEM]
(define(problem BW-rand-4)
(:domain blocksworld-4ops)
(:objects a b c d )
(:init
(handempty)
```

```
(on a b)
(on b c)
(on c d)
(ontable d)
(clear a)
)
(:goal
      (and
              (on d c)
              (on c d)
      )
)
)
```

```
[PLAN]
==============o1-preview RESPONSE================
**No plan exists that satisfies the goal with the given initial state and domain.**
```

## K   Prompts for Mystery Blocksworld

### K.1   Mystery Blocksworld Instances - Zero-Shot in Natural Language

I am playing with a set of objects. Here are the actions I can do

   Attack object
   Feast object from another object
   Succumb object
   Overcome object from another object

I have the following restrictions on my actions:
   To perform Attack action, the following facts need to be true: Province object, Planet object,
      Harmony.
   Once Attack action is performed the following facts will be true: Pain object.
   Once Attack action is performed the following facts will be false: Province object, Planet
      object, Harmony.
   To perform Succumb action, the following facts need to be true: Pain object.
   Once Succumb action is performed the following facts will be true: Province object, Planet
      object, Harmony.
   Once Succumb action is performed the following facts will be false: Pain object.
   To perform Overcome action, the following needs to be true: Province other object, Pain object.
   Once Overcome action is performed the following will be true: Harmony, Province object, Object
      Craves other object.
   Once Overcome action is performed the following will be false: Province other object, Pain
      object.
   To perform Feast action, the following needs to be true: Object Craves other object, Province
      object, Harmony.
   Once Feast action is performed the following will be true: Pain object, Province other object.
   Once Feast action is performed the following will be false:, Object Craves other object,
      Province object, Harmony.

[STATEMENT]
As initial conditions I have that, object b craves object c, harmony, planet object a, planet
    object c, planet object d, province object a, province object b and province object d.
My goal is to have that object c craves object b.

What is the plan to achieve my goal? Just give the actions in the plan.
==============o1-preview RESPONSE================
To achieve the goal where object c craves object b, you can perform the following sequence of
    actions:

1. **Feast object b from object c**
   - Preconditions:
     - 'Object b craves object c' (initial fact)
     - 'Province object b' (initial fact)
     - 'Harmony' (initial fact)
   - Effects:
     - Adds: 'Pain object b', 'Province object c'
     - Deletes: 'Object b craves object c', 'Province object b', 'Harmony'

2. **Succumb object b**
   - Preconditions:
     - 'Pain object b' (from previous action)
   - Effects:
     - Adds: 'Province object b', 'Planet object b', 'Harmony'
     - Deletes: 'Pain object b'

3. **Attack object c**
   - Preconditions:
     - 'Province object c' (from step 1)
     - 'Planet object c' (initial fact)
     - 'Harmony' (restored in step 2)
   - Effects:
     - Adds: 'Pain object c'
     - Deletes: 'Province object c', 'Planet object c', 'Harmony'

4. **Overcome object c from object b**
   - Preconditions:
     - 'Province object b' (restored in step 2)
     - 'Pain object c' (from step 3)
   - Effects:
     - Adds: 'Harmony', 'Province object c', 'Object c craves object b' (goal achieved)
     - Deletes: 'Province object b', 'Pain object c'

By following this plan, you manipulate the states of the objects through the actions provided, ultimately achieving the desired state where 'Object c craves object b'.

**Answer:**
To achieve the goal, perform these actions in order:

1. Feast object b from object c
2. Succumb object b
3. Attack object c
4. Overcome object c from object b

## L  Prompts for Randomized Mystery Blocksworld

### L.1  Randomized Mystery Blocksworld Instances - Zero-Shot in Natural Language

I am playing with a set of objects. Here are the actions I can do

J4gv801gnu2it0yj object_0 object_1.
U64y1a9apusmslxb object_0.
Kip9uw781pv62umn object_0.
Wio5amhq7814n006 object_0 object_1.

I have the following restrictions on my actions:

To perform j4gv801gnu2it0yj action, the following facts need to be true: tv30k33pzoulql6w object_1,
    and a46zhwykn6jvbabk object_0
Once j4gv801gnu2it0yj is performed the following facts will be true: cql4o62p1yeke3ok,
    tv30k33pzoulql6w object_0, and b6e9q4r60gagvdcn object_0 object_1
Once j4gv801gnu2it0yj is performed the following facts will be false: tv30k33pzoulql6w object_1,
    and a46zhwykn6jvbabk object_0
To perform u64y1a9apusmslxb action, the following facts need to be true: tv30k33pzoulql6w object_0,
    af9tse23ljclsqad object_0, and cql4o62p1yeke3ok
Once u64y1a9apusmslxb is performed the following facts will be true: a46zhwykn6jvbabk object_0
Once u64y1a9apusmslxb is performed the following facts will be false: tv30k33pzoulql6w object_0,
    af9tse23ljclsqad object_0, and cql4o62p1yeke3ok
To perform kip9uw781pv62umn action, the following facts need to be true:a46zhwykn6jvbabk object_0
Once kip9uw781pv62umn is performed the following facts will be true: tv30k33pzoulql6w object_0,
    cql4o62p1yeke3ok, and af9tse23ljclsqad object_0
Once kip9uw781pv62umn is performed the following facts will be false: a46zhwykn6jvbabk object_0
To perform wio5amhq7814n006 action, the following facts need to be true: b6e9q4r60gagvdcn object_0
    object_1, tv30k33pzoulql6w object_0, and cql4o62p1yeke3ok
Once wio5amhq7814n006 is performed the following facts will be true: a46zhwykn6jvbabk object_0, and
    tv30k33pzoulql6w object_1
Once wio5amhq7814n006 is performed the following facts will be false: b6e9q4r60gagvdcn object_0
    object_1, tv30k33pzoulql6w object_0, and cql4o62p1yeke3ok

[STATEMENT]
As initial conditions I have that, af9tse23ljclsqad object_0, af9tse23ljclsqad object_2,
    af9tse23ljclsqad object_3, b6e9q4r60gagvdcn object_1 object_2, cql4o62p1yeke3ok,
    tv30k33pzoulql6w object_0, tv30k33pzoulql6w object_1 and tv30k33pzoulql6w object_3.
My goal is to have that b6e9q4r60gagvdcn object_2 object_1.

What is the plan to achieve my goal? Just give the actions in the plan.
==============o1-preview RESPONSE================
To achieve the goal 'b6e9q4r60gagvdcn object_2 object_1', you need to perform the following
    sequence of actions:

1. **wio5amhq7814n006 object_1 object_2**
   - This action reverses the relationship between 'object_1' and 'object_2', setting up the
       necessary conditions for the next steps.
2. **kip9uw781pv62umn object_1**
   - By performing this action on 'object_1', you restore certain facts ('cql4o62p1yeke3ok')
       required for subsequent actions and adjust the state of 'object_1'.
3. **u64y1a9apusmslxb object_2**
   - This action modifies 'object_2' to have the required property 'a46zhwykn6jvbabk object_2',
       which is a precondition for the final action.
4. **j4gv801gnu2it0yj object_2 object_1**
   - Performing this action with 'object_2' and 'object_1' establishes the goal fact '
       b6e9q4r60gagvdcn object_2 object_1'.

**Plan Summary:**

- **Action 1:** 'wio5amhq7814n006 object_1 object_2'
- **Action 2:** 'kip9uw781pv62umn object_1'
- **Action 3:** 'u64y1a9apusmslxb object_2'
- **Action 4:** 'j4gv801gnu2it0yj object_2 object_1'

This sequence of actions satisfies all the necessary preconditions step-by-step and ultimately
    achieves your goal.

## L.2 Unsolvable Randomized Mystery Blocksworld Instances - Zero-Shot in PDDL

The following is a PDDL specification of a planning problem. The first part, under the heading [
    DOMAIN], is the domain file. The second part, under the heading [QUERY PROBLEM], is the problem
     file. Using this information, which is correct, and no further assumptions, find a plan which,
     when run from the specified initial state, satisfies the specified goal. Provide your answer
     as a sequence of actions in PDDL format. An action ACTION which acts on two objects OBJ1 and
     OBJ2 would be written (ACTION OBJ1 OBJ2). Do not provide anything else in your answer.
[DOMAIN]
(define (domain xaji0y)
    (:requirements :equality)
    (:predicates
        (tv30k33pzoulql6w ?x1 - object)
        (af9tse23ljclsqad ?x1 - object)
        (cql4o62p1yeke3ok )
        (a46zhwykn6jvbabk ?x1 - object)
        (b6e9q4r60gagvdcn ?x1 - object ?x2 - object)
    )

    (:action u64y1a9apusmslxb
     :parameters (?ob1 - object)
     :precondition (and (tv30k33pzoulql6w ?ob1) (af9tse23ljclsqad ?ob1) (cql4o62p1yeke3ok ))
     :effect (and
        (a46zhwykn6jvbabk ?ob1)
        (not (tv30k33pzoulql6w ?ob1))
        (not (af9tse23ljclsqad ?ob1))
        (not (cql4o62p1yeke3ok )))
    )

    (:action kip9uw781pv62umn
     :parameters (?ob1 - object)
     :precondition (a46zhwykn6jvbabk ?ob1)
     :effect (and
        (tv30k33pzoulql6w ?ob1)
        (cql4o62p1yeke3ok )
        (af9tse23ljclsqad ?ob1)
        (not (a46zhwykn6jvbabk ?ob1)))
    )

    (:action j4gv801gnu2it0yj
     :parameters (?ob1 - object ?ob2 - object)
     :precondition (and (tv30k33pzoulql6w ?ob2) (a46zhwykn6jvbabk ?ob1))
     :effect (and
        (cql4o62p1yeke3ok )
        (tv30k33pzoulql6w ?ob1)
        (b6e9q4r60gagvdcn ?ob1 ?ob2)
        (not (tv30k33pzoulql6w ?ob2))
        (not (a46zhwykn6jvbabk ?ob1)))
    )

    (:action wio5amhq7814n006
     :parameters (?ob1 - object ?ob2 - object)
     :precondition (and (b6e9q4r60gagvdcn ?ob1 ?ob2) (tv30k33pzoulql6w ?ob1) (cql4o62p1yeke3ok ))
     :effect (and
        (a46zhwykn6jvbabk ?ob1)
        (tv30k33pzoulql6w ?ob2)
        (not (b6e9q4r60gagvdcn ?ob1 ?ob2))
        (not (tv30k33pzoulql6w ?ob1))

```
        (not (cql4o62p1yeke3ok )))
    )

)

[QUERY PROBLEM]
(define(problem instance-1-4)
(:domain xaji0y)
(:objects a b c d )
(:init
(cql4o62p1yeke3ok)
(b6e9q4r60gagvdcn a d)
(af9tse23ljclsqad b)
(b6e9q4r60gagvdcn c a)
(b6e9q4r60gagvdcn d b)
(tv30k33pzoulql6w c)
)
(:goal
        (and
                (b6e9q4r60gagvdcn a d)
                (b6e9q4r60gagvdcn b a)
                (b6e9q4r60gagvdcn b d)
        )
)
)

[PLAN]
==============o1-preview RESPONSE================
There is no sequence of actions that, when run from the specified initial state, satisfies the
    specified goal. Thus, no plan can be provided.
```

## M   Prompts for Logistics

### M.1   Logistics - Zero-Shot in PDDL

```
The following is a PDDL specification of a planning problem. The first part, under the heading [
    DOMAIN], is the domain file. The second part, under the heading [QUERY PROBLEM], is the problem
     file. Using this information, which is correct, and no further assumptions, find a plan which,
     when run from the specified initial state, satisfies the specified goal. Provide your answer
    as a sequence of actions in PDDL format. An action ACTION which acts on two objects OBJ1 and
    OBJ2 would be written (ACTION OBJ1 OBJ2). Do not provide anything else in your answer.
[DOMAIN]
(define (domain logistics-strips)
  (:requirements :strips)
  (:predicates (OBJ ?obj)
               (TRUCK ?truck)
               (LOCATION ?loc)
               (AIRPLANE ?airplane)
               (CITY ?city)
               (AIRPORT ?airport)
               (at ?obj ?loc)
               (in ?obj1 ?obj2)
               (in-city ?obj ?city))

  ; (:types )         ; default object

(:action LOAD-TRUCK
  :parameters
   (?obj
```

```
   ?truck
   ?loc)
 :precondition
 (and (OBJ ?obj) (TRUCK ?truck) (LOCATION ?loc)
 (at ?truck ?loc) (at ?obj ?loc))
 :effect
 (and (not (at ?obj ?loc)) (in ?obj ?truck)))

(:action LOAD-AIRPLANE
 :parameters
 (?obj
  ?airplane
  ?loc)
 :precondition
 (and (OBJ ?obj) (AIRPLANE ?airplane) (LOCATION ?loc)
 (at ?obj ?loc) (at ?airplane ?loc))
 :effect
 (and (not (at ?obj ?loc)) (in ?obj ?airplane)))

(:action UNLOAD-TRUCK
 :parameters
 (?obj
  ?truck
  ?loc)
 :precondition
 (and (OBJ ?obj) (TRUCK ?truck) (LOCATION ?loc)
      (at ?truck ?loc) (in ?obj ?truck))
 :effect
 (and (not (in ?obj ?truck)) (at ?obj ?loc)))

(:action UNLOAD-AIRPLANE
 :parameters
 (?obj
  ?airplane
  ?loc)
 :precondition
 (and (OBJ ?obj) (AIRPLANE ?airplane) (LOCATION ?loc)
      (in ?obj ?airplane) (at ?airplane ?loc))
 :effect
 (and (not (in ?obj ?airplane)) (at ?obj ?loc)))

(:action DRIVE-TRUCK
 :parameters
 (?truck
  ?loc-from
  ?loc-to
  ?city)
 :precondition
 (and (TRUCK ?truck) (LOCATION ?loc-from) (LOCATION ?loc-to) (CITY ?city)
 (at ?truck ?loc-from)
 (in-city ?loc-from ?city)
 (in-city ?loc-to ?city))
 :effect
 (and (not (at ?truck ?loc-from)) (at ?truck ?loc-to)))

(:action FLY-AIRPLANE
 :parameters
```

```
  (?airplane
   ?loc-from
   ?loc-to)
 :precondition
  (and (AIRPLANE ?airplane) (AIRPORT ?loc-from) (AIRPORT ?loc-to)
       (at ?airplane ?loc-from))
 :effect
  (and (not (at ?airplane ?loc-from)) (at ?airplane ?loc-to)))
)

[QUERY PROBLEM]
(define(problem logistics-c2-s1-p1-a2)
(:domain logistics-strips)
(:objects a0 a1
          c0 c1
          t0 t1
          l0-0 l1-0
          p0
)
(:init
    (AIRPLANE a0)
    (AIRPLANE a1)
    (CITY c0)
    (CITY c1)
    (TRUCK t0)
    (TRUCK t1)
    (LOCATION l0-0)
    (in-city l0-0 c0)
    (LOCATION l1-0)
    (in-city l1-0 c1)
    (AIRPORT l0-0)
    (AIRPORT l1-0)
    (OBJ p0)
    (at t0 l0-0)
    (at t1 l1-0)
    (at p0 l1-0)
    (at a0 l0-0)
    (at a1 l0-0)
)
(:goal
    (and
        (at p0 l0-0)
    )
)
)
)

[PLAN]
=============o1-preview RESPONSE===============
(FLY-AIRPLANE a0 l0-0 l1-0)

(LOAD-AIRPLANE p0 a0 l1-0)

(FLY-AIRPLANE a0 l1-0 l0-0)

(UNLOAD-AIRPLANE p0 a0 l0-0)
```

## M.2 Randomized Logistics - Zero-Shot in PDDL

The following is a PDDL specification of a planning problem. The first part, under the heading
    [DOMAIN], is the domain file. The second part, under the heading [QUERY PROBLEM], is the
    problem file. Using this information, which is correct, and no further assumptions, find a
    plan which, when run from the specified initial state, satisfies the specified goal.
    Provide your answer as a sequence of actions in PDDL format. An action ACTION which acts on
     two objects OBJ1 and OBJ2 would be written (ACTION OBJ1 OBJ2). Do not provide anything
    else in your answer.
[DOMAIN]
(define (domain dxpflc3up53obvi)
    (:requirements :strips)
    (:predicates (a46zhwykn6jvbabk ?airport) (abwaom5mhyf979wg ?obj1 ?obj2) (b6e9q4r60gagvdcn ?city)
        (cql4o62p1yeke3ok ?airplane) (dvr5qjofrygv9zlx ?obj ?loc) (fe8cgp3y92p6lb85 ?loc) (
        j3bxzst7l100cyne ?obj ?city) (lwiiiuk750o3u28e ?obj) (rvgqqfhogwl9yspo ?truck))
    (:action af9tse23ljclsqad
        :parameters (?ob_1 ?ob_2 ?ob_3 ?ob_4)
        :precondition (and (rvgqqfhogwl9yspo ?ob_1) (fe8cgp3y92p6lb85 ?ob_2) (fe8cgp3y92p6lb85 ?ob_3
            ) (b6e9q4r60gagvdcn ?ob_4) (dvr5qjofrygv9zlx ?ob_1 ?ob_2) (j3bxzst7l100cyne ?ob_2 ?ob_4)
             (j3bxzst7l100cyne ?ob_3 ?ob_4))
        :effect (and (not (dvr5qjofrygv9zlx ?ob_1 ?ob_2)) (dvr5qjofrygv9zlx ?ob_1 ?ob_3))
    )
     (:action j4gv801gnu2it0yj
        :parameters (?ob_1 ?ob_2 ?ob_3)
        :precondition (and (lwiiiuk750o3u28e ?ob_1) (rvgqqfhogwl9yspo ?ob_2) (fe8cgp3y92p6lb85 ?ob_3
            ) (dvr5qjofrygv9zlx ?ob_2 ?ob_3) (abwaom5mhyf979wg ?ob_1 ?ob_2))
        :effect (and (not (abwaom5mhyf979wg ?ob_1 ?ob_2)) (dvr5qjofrygv9zlx ?ob_1 ?ob_3))
    )
     (:action kip9uw781pv62umn
        :parameters (?ob_1 ?ob_2 ?ob_3)
        :precondition (and (lwiiiuk750o3u28e ?ob_1) (cql4o62p1yeke3ok ?ob_2) (fe8cgp3y92p6lb85 ?ob_3
            ) (dvr5qjofrygv9zlx ?ob_1 ?ob_3) (dvr5qjofrygv9zlx ?ob_2 ?ob_3))
        :effect (and (not (dvr5qjofrygv9zlx ?ob_1 ?ob_3)) (abwaom5mhyf979wg ?ob_1 ?ob_2))
    )
     (:action tv30k33pzoulql6w
        :parameters (?ob_1 ?ob_2 ?ob_3)
        :precondition (and (cql4o62p1yeke3ok ?ob_1) (a46zhwykn6jvbabk ?ob_2) (a46zhwykn6jvbabk ?ob_3
            ) (dvr5qjofrygv9zlx ?ob_1 ?ob_2))
        :effect (and (not (dvr5qjofrygv9zlx ?ob_1 ?ob_2)) (dvr5qjofrygv9zlx ?ob_1 ?ob_3))
    )
     (:action u64y1a9apusmslxb
        :parameters (?ob_1 ?ob_2 ?ob_3)
        :precondition (and (lwiiiuk750o3u28e ?ob_1) (rvgqqfhogwl9yspo ?ob_2) (fe8cgp3y92p6lb85 ?ob_3
            ) (dvr5qjofrygv9zlx ?ob_2 ?ob_3) (dvr5qjofrygv9zlx ?ob_1 ?ob_3))
        :effect (and (not (dvr5qjofrygv9zlx ?ob_1 ?ob_3)) (abwaom5mhyf979wg ?ob_1 ?ob_2))
    )
     (:action wio5amhq7814n006
        :parameters (?ob_1 ?ob_2 ?ob_3)
        :precondition (and (lwiiiuk750o3u28e ?ob_1) (cql4o62p1yeke3ok ?ob_2) (fe8cgp3y92p6lb85 ?ob_3
            ) (abwaom5mhyf979wg ?ob_1 ?ob_2) (dvr5qjofrygv9zlx ?ob_2 ?ob_3))
        :effect (and (not (abwaom5mhyf979wg ?ob_1 ?ob_2)) (dvr5qjofrygv9zlx ?ob_1 ?ob_3))
    )
)

[QUERY PROBLEM]
(define(problem vd24g3dfuion115)
    (:domain dxpflc3up53obvi)
    (:objects o1 o2 o3 o4 o5 o6 o7 o8 o9)
    (:init (a46zhwykn6jvbabk o4) (a46zhwykn6jvbabk o7) (b6e9q4r60gagvdcn o1) (b6e9q4r60gagvdcn o9)
        (cql4o62p1yeke3ok o2) (cql4o62p1yeke3ok o6) (dvr5qjofrygv9zlx o2 o7) (dvr5qjofrygv9zlx o3
        o4) (dvr5qjofrygv9zlx o5 o4) (dvr5qjofrygv9zlx o6 o7) (dvr5qjofrygv9zlx o8 o7) (

```
        fe8cgp3y92p6lb85 o4) (fe8cgp3y92p6lb85 o7) (j3bxzst7l100cyne o4 o9) (j3bxzst7l100cyne o7 o1
            ) (lwiiiuk750o3u28e o5) (rvgqqfhogwl9yspo o3) (rvgqqfhogwl9yspo o8))
    (:goal (dvr5qjofrygv9zlx o5 o7))
)

[PLAN]
=============o1-preview RESPONSE===============
(tv30k33pzoulql6w o2 o7 o4)
(kip9uw781pv62umn o5 o2 o4)
(tv30k33pzoulql6w o2 o4 o7)
(wio5amhq7814n006 o5 o2 o7)
```

## N  Prompts for Sokoban

### N.1  Sokoban - Zero-Shot in PDDL

```
The following is a PDDL specification of a planning problem. The first part, under the heading [
    DOMAIN], is the domain file. The second part, under the heading [QUERY PROBLEM], is the problem
     file. Using this information, which is correct, and no further assumptions, find a plan which,
     when run from the specified initial state, satisfies the specified goal. Provide your answer
    as a sequence of actions in PDDL format. An action ACTION which acts on two objects OBJ1 and
    OBJ2 would be written (ACTION OBJ1 OBJ2). Do not provide anything else in your answer.
[DOMAIN]
(define (domain typed-sokoban)
(:requirements :typing)
(:types LOC DIR BOX)
(:predicates
        (at-robot ?l - LOC)
        (at ?o - BOX ?l - LOC)
        (adjacent ?l1 - LOC ?l2 - LOC ?d - DIR)
        (clear ?l - LOC)
)

(:action move
:parameters (?from - LOC ?to - LOC ?dir - DIR)
:precondition (and (clear ?to) (at-robot ?from) (adjacent ?from ?to ?dir))
:effect (and (at-robot ?to) (not (at-robot ?from)))
)

(:action push
:parameters (?rloc - LOC ?bloc - LOC ?floc - LOC ?dir - DIR ?b - BOX)
:precondition (and (at-robot ?rloc) (at ?b ?bloc) (clear ?floc)
                (adjacent ?rloc ?bloc ?dir) (adjacent ?bloc ?floc ?dir))

:effect (and (at-robot ?bloc) (at ?b ?floc) (clear ?bloc)
        (not (at-robot ?rloc)) (not (at ?b ?bloc)) (not (clear ?floc)))
)
)

[QUERY PROBLEM]
(define(problem typed-sokoban-grid7-boxes1-walls2)
(:domain typed-sokoban)
(:objects
    up down left right - DIR
    box0 - BOX
    f0-0f f0-1f f0-2f f0-3f f0-4f f0-5f f0-6f
    f1-0f f1-1f f1-2f f1-3f f1-4f f1-5f f1-6f
```

```
        f2-0f f2-1f f2-2f f2-3f f2-4f f2-5f f2-6f
        f3-0f f3-1f f3-2f f3-3f f3-4f f3-5f f3-6f
        f4-0f f4-1f f4-2f f4-3f f4-4f f4-5f f4-6f
        f5-0f f5-1f f5-2f f5-3f f5-4f f5-5f f5-6f
        f6-0f f6-1f f6-2f f6-3f f6-4f f6-5f f6-6f - LOC
)
(:init
(adjacent f0-0f f0-1f right)
(adjacent f0-0f f1-0f down)
(adjacent f0-1f f0-0f left)
(adjacent f0-1f f0-2f right)
(adjacent f0-1f f1-1f down)
(adjacent f0-2f f0-1f left)
(adjacent f0-2f f0-3f right)
(adjacent f0-2f f1-2f down)
(adjacent f0-3f f0-2f left)
(adjacent f0-3f f0-4f right)
(adjacent f0-3f f1-3f down)
(adjacent f0-4f f0-3f left)
(adjacent f0-4f f0-5f right)
(adjacent f0-4f f1-4f down)
(adjacent f0-5f f0-4f left)
(adjacent f0-5f f0-6f right)
(adjacent f0-5f f1-5f down)
(adjacent f0-6f f0-5f left)
(adjacent f0-6f f1-6f down)
(adjacent f1-0f f1-1f right)
(adjacent f1-0f f0-0f up)
(adjacent f1-0f f2-0f down)
(adjacent f1-1f f1-0f left)
(adjacent f1-1f f1-2f right)
(adjacent f1-1f f0-1f up)
(adjacent f1-1f f2-1f down)
(adjacent f1-2f f1-1f left)
(adjacent f1-2f f1-3f right)
(adjacent f1-2f f0-2f up)
(adjacent f1-2f f2-2f down)
(adjacent f1-3f f1-2f left)
(adjacent f1-3f f1-4f right)
(adjacent f1-3f f0-3f up)
(adjacent f1-3f f2-3f down)
(adjacent f1-4f f1-3f left)
(adjacent f1-4f f1-5f right)
(adjacent f1-4f f0-4f up)
(adjacent f1-4f f2-4f down)
(adjacent f1-5f f1-4f left)
(adjacent f1-5f f1-6f right)
(adjacent f1-5f f0-5f up)
(adjacent f1-5f f2-5f down)
(adjacent f1-6f f1-5f left)
(adjacent f1-6f f0-6f up)
(adjacent f1-6f f2-6f down)
(adjacent f2-0f f2-1f right)
(adjacent f2-0f f1-0f up)
(adjacent f2-0f f3-0f down)
(adjacent f2-1f f2-0f left)
(adjacent f2-1f f2-2f right)
(adjacent f2-1f f1-1f up)
(adjacent f2-1f f3-1f down)
```

```
(adjacent f2-2f f2-1f left)
(adjacent f2-2f f2-3f right)
(adjacent f2-2f f1-2f up)
(adjacent f2-2f f3-2f down)
(adjacent f2-3f f2-2f left)
(adjacent f2-3f f2-4f right)
(adjacent f2-3f f1-3f up)
(adjacent f2-3f f3-3f down)
(adjacent f2-4f f2-3f left)
(adjacent f2-4f f2-5f right)
(adjacent f2-4f f1-4f up)
(adjacent f2-4f f3-4f down)
(adjacent f2-5f f2-4f left)
(adjacent f2-5f f2-6f right)
(adjacent f2-5f f1-5f up)
(adjacent f2-5f f3-5f down)
(adjacent f2-6f f2-5f left)
(adjacent f2-6f f1-6f up)
(adjacent f2-6f f3-6f down)
(adjacent f3-0f f3-1f right)
(adjacent f3-0f f2-0f up)
(adjacent f3-0f f4-0f down)
(adjacent f3-1f f3-0f left)
(adjacent f3-1f f3-2f right)
(adjacent f3-1f f2-1f up)
(adjacent f3-1f f4-1f down)
(adjacent f3-2f f3-1f left)
(adjacent f3-2f f3-3f right)
(adjacent f3-2f f2-2f up)
(adjacent f3-2f f4-2f down)
(adjacent f3-3f f3-2f left)
(adjacent f3-3f f3-4f right)
(adjacent f3-3f f2-3f up)
(adjacent f3-3f f4-3f down)
(adjacent f3-4f f3-3f left)
(adjacent f3-4f f3-5f right)
(adjacent f3-4f f2-4f up)
(adjacent f3-4f f4-4f down)
(adjacent f3-5f f3-4f left)
(adjacent f3-5f f3-6f right)
(adjacent f3-5f f2-5f up)
(adjacent f3-5f f4-5f down)
(adjacent f3-6f f3-5f left)
(adjacent f3-6f f2-6f up)
(adjacent f3-6f f4-6f down)
(adjacent f4-0f f4-1f right)
(adjacent f4-0f f3-0f up)
(adjacent f4-0f f5-0f down)
(adjacent f4-1f f4-0f left)
(adjacent f4-1f f4-2f right)
(adjacent f4-1f f3-1f up)
(adjacent f4-1f f5-1f down)
(adjacent f4-2f f4-1f left)
(adjacent f4-2f f4-3f right)
(adjacent f4-2f f3-2f up)
(adjacent f4-2f f5-2f down)
(adjacent f4-3f f4-2f left)
(adjacent f4-3f f4-4f right)
(adjacent f4-3f f3-3f up)
```

```
(adjacent f4-3f f5-3f down)
(adjacent f4-4f f4-3f left)
(adjacent f4-4f f4-5f right)
(adjacent f4-4f f3-4f up)
(adjacent f4-4f f5-4f down)
(adjacent f4-5f f4-4f left)
(adjacent f4-5f f4-6f right)
(adjacent f4-5f f3-5f up)
(adjacent f4-5f f5-5f down)
(adjacent f4-6f f4-5f left)
(adjacent f4-6f f3-6f up)
(adjacent f4-6f f5-6f down)
(adjacent f5-0f f5-1f right)
(adjacent f5-0f f4-0f up)
(adjacent f5-0f f6-0f down)
(adjacent f5-1f f5-0f left)
(adjacent f5-1f f5-2f right)
(adjacent f5-1f f4-1f up)
(adjacent f5-1f f6-1f down)
(adjacent f5-2f f5-1f left)
(adjacent f5-2f f5-3f right)
(adjacent f5-2f f4-2f up)
(adjacent f5-2f f6-2f down)
(adjacent f5-3f f5-2f left)
(adjacent f5-3f f5-4f right)
(adjacent f5-3f f4-3f up)
(adjacent f5-3f f6-3f down)
(adjacent f5-4f f5-3f left)
(adjacent f5-4f f5-5f right)
(adjacent f5-4f f4-4f up)
(adjacent f5-4f f6-4f down)
(adjacent f5-5f f5-4f left)
(adjacent f5-5f f5-6f right)
(adjacent f5-5f f4-5f up)
(adjacent f5-5f f6-5f down)
(adjacent f5-6f f5-5f left)
(adjacent f5-6f f4-6f up)
(adjacent f5-6f f6-6f down)
(adjacent f6-0f f6-1f right)
(adjacent f6-0f f5-0f up)
(adjacent f6-1f f6-0f left)
(adjacent f6-1f f6-2f right)
(adjacent f6-1f f5-1f up)
(adjacent f6-2f f6-1f left)
(adjacent f6-2f f6-3f right)
(adjacent f6-2f f5-2f up)
(adjacent f6-3f f6-2f left)
(adjacent f6-3f f6-4f right)
(adjacent f6-3f f5-3f up)
(adjacent f6-4f f6-3f left)
(adjacent f6-4f f6-5f right)
(adjacent f6-4f f5-4f up)
(adjacent f6-5f f6-4f left)
(adjacent f6-5f f6-6f right)
(adjacent f6-5f f5-5f up)
(adjacent f6-6f f6-5f left)
(adjacent f6-6f f5-6f up)
(at box0 f1-2f)
(clear f0-0f)
```

```
(clear f0-2f)
(clear f0-3f)
(clear f0-4f)
(clear f0-5f)
(clear f0-6f)
(clear f1-0f)
(clear f1-1f)
(clear f1-3f)
(clear f1-4f)
(clear f1-5f)
(clear f1-6f)
(clear f2-0f)
(clear f2-1f)
(clear f2-2f)
(clear f2-3f)
(clear f2-4f)
(clear f2-5f)
(clear f2-6f)
(clear f3-0f)
(clear f3-3f)
(clear f3-4f)
(clear f3-5f)
(clear f3-6f)
(clear f4-0f)
(clear f4-1f)
(clear f4-2f)
(clear f4-3f)
(clear f4-4f)
(clear f4-5f)
(clear f4-6f)
(clear f5-0f)
(clear f5-1f)
(clear f5-2f)
(clear f5-3f)
(clear f5-4f)
(clear f5-5f)
(clear f5-6f)
(at-robot f6-0f)
(clear f6-0f)
(clear f6-1f)
(clear f6-2f)
(clear f6-3f)
(clear f6-4f)
(clear f6-6f)
)
(:goal
(and
(at box0 f6-6f)
)
)
)

[PLAN]
=============o1-preview RESPONSE===============
(move f6-0f f5-0f up)
(move f5-0f f4-0f up)
(move f4-0f f3-0f up)
(move f3-0f f2-0f up)
(move f2-0f f1-0f up)
```

```
(move f1-0f f1-1f right)
(push f1-1f f1-2f f1-3f right box0)
(push f1-2f f1-3f f1-4f right box0)
(push f1-3f f1-4f f1-5f right box0)
(push f1-4f f1-5f f1-6f right box0)
(move f1-5f f0-5f up)
(move f0-5f f0-6f right)
(push f0-6f f1-6f f2-6f down box0)
(push f1-6f f2-6f f3-6f down box0)
(push f2-6f f3-6f f4-6f down box0)
(push f3-6f f4-6f f5-6f down box0)
(push f4-6f f5-6f f6-6f down box0)
```

### N.2 Sokoban - Backprompt

```
The above plan is invalid. The following action at step 20 has an unmet precondition:
push f8-5f f8-6f f8-7f right box1
The unmet precondition is:
(clear f8-7f)
Provide your revised plan as a sequence of actions in PDDL format. An action ACTION which acts on
    two objects OBJ1 and OBJ2 would be written (ACTION OBJ1 OBJ2). Do not provide anything else in
    your answer.

[PLAN]
```

## O   Prompts for Graph Coloring

### O.1   Graph Coloring - Hard

```
    Color the following graph, described as a set of edges, such that no two vertices on the same
        edge share a color.
You may use at most 5 colors.
Vertex 0 is connected to vertex 3.
Vertex 0 is connected to vertex 6.
Vertex 0 is connected to vertex 8.
Vertex 0 is connected to vertex 13.
Vertex 0 is connected to vertex 15.
Vertex 0 is connected to vertex 17.
Vertex 0 is connected to vertex 19.
Vertex 1 is connected to vertex 4.
Vertex 1 is connected to vertex 5.
Vertex 1 is connected to vertex 8.
Vertex 1 is connected to vertex 11.
Vertex 1 is connected to vertex 12.
Vertex 1 is connected to vertex 13.
Vertex 1 is connected to vertex 14.
Vertex 1 is connected to vertex 17.
Vertex 1 is connected to vertex 18.
Vertex 1 is connected to vertex 19.
Vertex 2 is connected to vertex 3.
Vertex 2 is connected to vertex 4.
Vertex 2 is connected to vertex 6.
Vertex 2 is connected to vertex 8.
Vertex 2 is connected to vertex 13.
Vertex 2 is connected to vertex 14.
Vertex 2 is connected to vertex 15.
Vertex 2 is connected to vertex 16.
Vertex 2 is connected to vertex 17.
```

Vertex 3 is connected to vertex 5.
Vertex 3 is connected to vertex 8.
Vertex 3 is connected to vertex 11.
Vertex 3 is connected to vertex 12.
Vertex 3 is connected to vertex 14.
Vertex 3 is connected to vertex 16.
Vertex 4 is connected to vertex 5.
Vertex 4 is connected to vertex 7.
Vertex 4 is connected to vertex 9.
Vertex 4 is connected to vertex 11.
Vertex 4 is connected to vertex 17.
Vertex 4 is connected to vertex 18.
Vertex 5 is connected to vertex 11.
Vertex 5 is connected to vertex 14.
Vertex 5 is connected to vertex 15.
Vertex 6 is connected to vertex 11.
Vertex 6 is connected to vertex 16.
Vertex 6 is connected to vertex 17.
Vertex 7 is connected to vertex 9.
Vertex 7 is connected to vertex 10.
Vertex 7 is connected to vertex 13.
Vertex 7 is connected to vertex 14.
Vertex 7 is connected to vertex 16.
Vertex 7 is connected to vertex 17.
Vertex 8 is connected to vertex 10.
Vertex 8 is connected to vertex 12.
Vertex 8 is connected to vertex 13.
Vertex 8 is connected to vertex 16.
Vertex 8 is connected to vertex 19.
Vertex 9 is connected to vertex 11.
Vertex 9 is connected to vertex 13.
Vertex 9 is connected to vertex 17.
Vertex 10 is connected to vertex 11.
Vertex 10 is connected to vertex 13.
Vertex 10 is connected to vertex 16.
Vertex 10 is connected to vertex 17.
Vertex 10 is connected to vertex 18.
Vertex 10 is connected to vertex 19.
Vertex 11 is connected to vertex 13.
Vertex 11 is connected to vertex 17.
Vertex 11 is connected to vertex 18.
Vertex 11 is connected to vertex 19.
Vertex 12 is connected to vertex 13.
Vertex 12 is connected to vertex 15.
Vertex 12 is connected to vertex 16.
Vertex 12 is connected to vertex 18.
Vertex 12 is connected to vertex 19.
Vertex 13 is connected to vertex 14.
Vertex 13 is connected to vertex 18.
Vertex 14 is connected to vertex 15.
Vertex 14 is connected to vertex 18.
Vertex 14 is connected to vertex 19.
There are a total of 20 vertices. Please label every vertex, even if it is disconnected from the
    rest of the graph.Please provide each vertex's color. Do not skip any vertices. Each color must
    be provided on a new line in the response and should be formatted as "{VERTEX NUMBER}: {VERTEX
    COLOR ASSIGNMENT}". Please do not provide anything else in your response, and end your
    response with '[ANSWER END]'
=============o1-preview RESPONSE===============
0: 1

```
1: 3
2: 1
3: 2
4: 4
5: 5
6: 3
7: 1
8: 4
9: 3
10: 3
11: 1
12: 1
13: 2
14: 4
15: 2
16: 5
17: 2
18: 5
19: 2
[ANSWER END]
```

## P   Graph Coloring Backprompt - Hard

```
    This is incorrect. Feedback:
Vertex 0 was not given a value in the coloring.
Vertex 1 was not given a value in the coloring.
Vertex 0 was not given a value in the coloring.
Vertex 2 was not given a value in the coloring.
Vertex 0 was not given a value in the coloring.
Vertex 3 was not given a value in the coloring.
Vertex 0 was not given a value in the coloring.
Vertex 4 was not given a value in the coloring.
Vertex 0 was not given a value in the coloring.
Vertex 10 was not given a value in the coloring.
Vertex 0 was not given a value in the coloring.
Vertex 12 was not given a value in the coloring.
Vertex 0 was not given a value in the coloring.
Vertex 13 was not given a value in the coloring.
Vertex 0 was not given a value in the coloring.
Vertex 15 was not given a value in the coloring.
Vertex 0 was not given a value in the coloring.
Vertex 16 was not given a value in the coloring.
Vertex 0 was not given a value in the coloring.
Vertex 17 was not given a value in the coloring.
Vertex 0 was not given a value in the coloring.
Vertex 18 was not given a value in the coloring.
Vertex 1 was not given a value in the coloring.
Vertex 2 was not given a value in the coloring.
Vertex 1 was not given a value in the coloring.
Vertex 3 was not given a value in the coloring.
Vertex 1 was not given a value in the coloring.
Vertex 8 was not given a value in the coloring.
Vertex 1 was not given a value in the coloring.
Vertex 9 was not given a value in the coloring.
Vertex 1 was not given a value in the coloring.
Vertex 10 was not given a value in the coloring.
Vertex 1 was not given a value in the coloring.
Vertex 15 was not given a value in the coloring.
```

```
Vertex 1 was not given a value in the coloring.
Vertex 18 was not given a value in the coloring.
Vertex 2 was not given a value in the coloring.
Vertex 3 was not given a value in the coloring.
Vertex 2 was not given a value in the coloring.
Vertex 5 was not given a value in the coloring.
Vertex 2 was not given a value in the coloring.
Vertex 6 was not given a value in the coloring.
Vertex 2 was not given a value in the coloring.
Vertex 8 was not given a value in the coloring.
Vertex 2 was not given a value in the coloring.
Vertex 11 was not given a value in the coloring.
Vertex 2 was not given a value in the coloring.
Vertex 12 was not given a value in the coloring.
Vertex 2 was not given a value in the coloring.
Vertex 14 was not given a value in the coloring.
Vertex 2 was not given a value in the coloring.
Vertex 15 was not given a value in the coloring.
Vertex 2 was not given a value in the coloring.
Vertex 19 was not given a value in the coloring.
Vertex 3 was not given a value in the coloring.
Vertex 18 was not given a value in the coloring.
Vertex 3 was not given a value in the coloring.
Vertex 19 was not given a value in the coloring.
Vertex 4 was not given a value in the coloring.
Vertex 6 was not given a value in the coloring.
Vertex 4 was not given a value in the coloring.
Vertex 7 was not given a value in the coloring.
Vertex 4 was not given a value in the coloring.
Vertex 9 was not given a value in the coloring.
Vertex 4 was not given a value in the coloring.
Vertex 11 was not given a value in the coloring.
Vertex 4 was not given a value in the coloring.
Vertex 13 was not given a value in the coloring.
Vertex 4 was not given a value in the coloring.
Vertex 18 was not given a value in the coloring.
Vertex 4 was not given a value in the coloring.
Vertex 19 was not given a value in the coloring.
Vertex 5 was not given a value in the coloring.
Vertex 6 was not given a value in the coloring.
Vertex 5 was not given a value in the coloring.
Vertex 7 was not given a value in the coloring.
Vertex 5 was not given a value in the coloring.
Vertex 8 was not given a value in the coloring.
Vertex 5 was not given a value in the coloring.
Vertex 10 was not given a value in the coloring.
Vertex 5 was not given a value in the coloring.
Vertex 14 was not given a value in the coloring.
Vertex 6 was not given a value in the coloring.
Vertex 7 was not given a value in the coloring.
Vertex 6 was not given a value in the coloring.
Vertex 9 was not given a value in the coloring.
Vertex 6 was not given a value in the coloring.
Vertex 10 was not given a value in the coloring.
Vertex 6 was not given a value in the coloring.
Vertex 12 was not given a value in the coloring.
Vertex 6 was not given a value in the coloring.
Vertex 13 was not given a value in the coloring.
Vertex 6 was not given a value in the coloring.
```

```
Vertex 14 was not given a value in the coloring.
Vertex 6 was not given a value in the coloring.
Vertex 15 was not given a value in the coloring.
Vertex 6 was not given a value in the coloring.
Vertex 16 was not given a value in the coloring.
Vertex 6 was not given a value in the coloring.
Vertex 17 was not given a value in the coloring.
Vertex 6 was not given a value in the coloring.
Vertex 19 was not given a value in the coloring.
Vertex 7 was not given a value in the coloring.
Vertex 8 was not given a value in the coloring.
Vertex 7 was not given a value in the coloring.
Vertex 9 was not given a value in the coloring.
Vertex 7 was not given a value in the coloring.
Vertex 11 was not given a value in the coloring.
Vertex 7 was not given a value in the coloring.
Vertex 12 was not given a value in the coloring.
Vertex 7 was not given a value in the coloring.
Vertex 13 was not given a value in the coloring.
Vertex 7 was not given a value in the coloring.
Vertex 14 was not given a value in the coloring.
Vertex 7 was not given a value in the coloring.
Vertex 15 was not given a value in the coloring.
Vertex 7 was not given a value in the coloring.
Vertex 18 was not given a value in the coloring.
Vertex 8 was not given a value in the coloring.
Vertex 11 was not given a value in the coloring.
Vertex 8 was not given a value in the coloring.
Vertex 13 was not given a value in the coloring.
Vertex 8 was not given a value in the coloring.
Vertex 14 was not given a value in the coloring.
Vertex 8 was not given a value in the coloring.
Vertex 19 was not given a value in the coloring.
Vertex 9 was not given a value in the coloring.
Vertex 10 was not given a value in the coloring.
Vertex 9 was not given a value in the coloring.
Vertex 11 was not given a value in the coloring.
Vertex 9 was not given a value in the coloring.
Vertex 13 was not given a value in the coloring.
Vertex 9 was not given a value in the coloring.
Vertex 14 was not given a value in the coloring.
Vertex 9 was not given a value in the coloring.
Vertex 15 was not given a value in the coloring.
Vertex 10 was not given a value in the coloring.
Vertex 12 was not given a value in the coloring.
Vertex 10 was not given a value in the coloring.
Vertex 16 was not given a value in the coloring.
Vertex 10 was not given a value in the coloring.
Vertex 17 was not given a value in the coloring.
Vertex 11 was not given a value in the coloring.
Vertex 12 was not given a value in the coloring.
Vertex 11 was not given a value in the coloring.
Vertex 13 was not given a value in the coloring.
Vertex 11 was not given a value in the coloring.
Vertex 14 was not given a value in the coloring.
Vertex 11 was not given a value in the coloring.
Vertex 15 was not given a value in the coloring.
Vertex 11 was not given a value in the coloring.
Vertex 17 was not given a value in the coloring.
```

```
Vertex 11 was not given a value in the coloring.
Vertex 19 was not given a value in the coloring.
Vertex 12 was not given a value in the coloring.
Vertex 13 was not given a value in the coloring.
Vertex 12 was not given a value in the coloring.
Vertex 18 was not given a value in the coloring.
Vertex 13 was not given a value in the coloring.
Vertex 17 was not given a value in the coloring.
Vertex 14 was not given a value in the coloring.
Vertex 19 was not given a value in the coloring.
Vertex 15 was not given a value in the coloring.
Vertex 19 was not given a value in the coloring.
Vertex 16 was not given a value in the coloring.
Vertex 17 was not given a value in the coloring.
Vertex 17 was not given a value in the coloring.
Vertex 18 was not given a value in the coloring.

Using this feedback, please try again.
Please provide each vertex's color. Do not skip any vertices. Each color must be provided on a new
    line in the response and should be formatted as "{VERTEX NUMBER}: {VERTEX COLOR ASSIGNMENT}".
    Please do not provide anything else in your response, and end your response with '[ANSWER END]'
```

## Q Prompts for OSU Travel Planning

### Q.1 First Iteration

```
You are a proficient planner. Based on the provided information and query, please give me a
    detailed plan, including specifics such as flight numbers (e.g., F0123456),
restaurant names, and accommodation names. Note that all the information in your plan should be
    derived from the provided data. You should give a travel plan in JSON format as shown in the
example below. Additionally, all details should align with commonsense. The symbol '-' indicates
    that information is unnecessary. For example, in the provided sample, you do not need to
plan after returning to the departure city. When you travel to two cities in one day, you should
    note it in the 'current_city' section as in the example (i.e., from A to B).

***** Example *****
Query: Could you create a travel plan for 7 people from Ithaca to Charlotte spanning 3 days, from
    March 8th to March 14th, 2022, with a budget of $30,200?
Travel Plan:
[
    {
        "day": 1,
        "people_number": 7,
        "current_city": "from Ithaca to Charlotte",
        "transportation": "Flight Number: F3633413, from Ithaca to Charlotte, Departure Time: 05:38,
             Arrival Time: 07:46",
        "breakfast": "Nagaland's Kitchen, Charlotte",
        "attraction": "The Charlotte Museum of History, Charlotte",
        "lunch": "Cafe Maple Street, Charlotte",
        "dinner": "Bombay Vada Pav, Charlotte",
        "accommodation": "Affordable Spacious Refurbished Room in Bushwick!, Charlotte"
    },
    {
        "day": 2,
        "people_number": 7,
        "current_city": "Charlotte",
        "transportation": "-",
        "breakfast": "Olive Tree Cafe, Charlotte",
        "attraction": "The Mint Museum, Charlotte;Romare Bearden Park, Charlotte",
```

```
        "lunch": "Birbal Ji Dhaba, Charlotte",
        "dinner": "Pind Balluchi, Charlotte",
        "accommodation": "Affordable Spacious Refurbished Room in Bushwick!, Charlotte"
    },
    {
        "day": 3,
        "people_number": 7,
        "current_city": "from Charlotte to Ithaca",
        "transportation": "Flight Number: F3786167, from Charlotte to Ithaca, Departure Time: 21:42,
            Arrival Time: 23:26",
        "breakfast": "Books Monument, Charlotte",
        "attraction": "Books Monument, Charlotte",
        "lunch": "Olive Tree Cafe, Charlotte",
        "dinner": "Kylin Skybar, Charlotte",
        "accommodation": "-"
    }
]
***** Example Ends *****

Given information: [{'Description': 'Attractions in Nashville', 'Content': '
    Name Latitude Longitude                                        Address
Phone
Website    City\nCountry Music Hall of Fame and Museum 36.158263 -86.776126 222 Rep. John Lewis
    Way S, Nashville, TN 37203, USA (615) 416-2001
https://countrymusichalloffame.org/ Nashville\n      Nashville Zoo at Grassmere 36.089705 -86.742096
      3777 Nolensville Pk, Nashville, TN 37211, USA (615) 833-1534
http://www.nashvillezoo.org/ Nashville\n Belle Meade Historic Site & Winery 36.104916 -86.864695
    5025 Harding Pike, Nashville, TN 37205, USA (615) 356-0501
https://visitbellemeade.com/ Nashville\n              Johnny Cash Museum 36.160939 -86.775757
         119 3rd Ave S, Nashville, TN 37201, USA (615) 256-1777
http://www.johnnycashmuseum.com/ Nashville\n              Centennial Park 36.148946 -86.812750
        2500 West End Ave, Nashville, TN 37203, USA (615) 862-8400
https://www.nashville.gov/Parks-and-Recreation/Parks/Centennial-Park.aspx Nashville\n
    Grand Ole Opry 36.206857 -86.692108    600 Opry Mills Dr, Nashville, TN
37214, USA (615) 871-6779
https://www.opry.com/?utm_campaign=opry&utm_medium=organicsearch&utm_source=googlemybusiness&
    utm_audience=tofu_googlemybusiness&utm_content=brandstory_google_my_business_website_link
Nashville\n              Frist Art Museum 36.157897 -86.783853        919 Broadway, Nashville,
    TN 37203, USA (615) 244-3340
https://fristartmuseum.org/ Nashville\n              The Parthenon 36.149674 -86.813347     2500
    West End Ave, Nashville, TN 37203, USA (615) 862-8431
https://www.nashvilleparthenon.com/ Nashville\n              Nashville Shores 36.158209 -86.604915
         4001 Bell Rd, Hermitage, TN 37076, USA (615) 889-7050
http://www.nashvilleshores.com/lodging Nashville\n Musicians Hall of Fame and Museum 36.167668
    -86.782399      401 Gay St, Nashville, TN 37219, USA (615) 244-3263
https://www.musicianshalloffame.com/ Nashville\n Bicentennial Capitol Mall State Park 36.170887
    -86.787589 600 James Robertson Pkwy, Nashville, TN 37243, USA (888) 867-2757
https://tnstateparks.com/parks/bicentennial-mall Nashville\n              Lane Motor Museum 36.140197
     -86.734580 702 Murfreesboro Pike, Nashville, TN 37210, USA (615) 742-7445
http://www.lanemotormuseum.org/ Nashville\n              Honky Tonk Highway 36.160393 -86.778432
         501 Broadway, Nashville, TN 37203, USA (800) 657-6910
https://www.visitmusiccity.com/visitors/honkytonkhighway Nashville\n    Adventure Science Center
    36.146614 -86.775481 800 Fort Negley Blvd, Nashville, TN 37203, USA (615)
862-5160
http://www.adventuresci.org/ Nashville\n              Patsy Cline Museum 36.160904 -86.776030
         119 3rd Ave S, Nashville, TN 37201, USA (615) 454-4722
https://www.patsymuseum.com/ Nashville\n     Andrew Jackson's Hermitage 36.213757 -86.615310
    4580 Rachels Ln, Hermitage, TN 37076, USA (615) 889-2941
```

```
http://www.thehermitage.com/ Nashville\n                NashTrash Tours 36.169474 -86.788254 900
    Rosa L Parks Blvd, Nashville, TN 37208, USA (615) 226-7300
http://www.nashtrash.com/ Nashville\n               Fort Nashborough 36.164268 -86.775430
    170 1st Ave N, Nashville, TN 37201, USA (615) 862-8400
https://www.nashville.gov/Parks-and-Recreation/Historic-Sites/Fort-Nashborough.aspx Nashville\n
            Ryman Auditorium 36.161248 -86.778471        116 5th Ave N,
Nashville, TN 37219, USA (615) 889-3060
https://www.ryman.com/?utm_campaign=ryman&utm_medium=organicsearch&utm_source=googlemybusiness&
    utm_audience=tofu_googlemybusiness&utm_content=brandstory_google_my_business_website_link
Nashville\n        Nashville Public Square Park 36.166611 -86.778126 Union St & 3rd Ave N, Nashville
    , TN 37201, USA (615) 743-3090
https://www.nashvilledowntown.com/go/public-square-park Nashville'}, {'Description': 'Restaurants
    in Nashville', 'Content': "              Name  Average Cost
Cuisines Aggregate Rating    City\n                      Bangkok 1        44
                    Cafe, Seafood           3.3 Nashville\n              Bablu Fast
Food          13    Desserts, Pizza, Mexican, BBQ, Fast Food        0.0 Nashville\n
                Full Dabba       77   Pizza, Mexican, Fast Food, Cafe, American
0.0 Nashville\n                   Twigly        14 Pizza, French, Fast Food, Chinese,
    Seafood       4.5 Nashville\n                   Veg Hut        35
Tea, Mexican, Seafood          0.0 Nashville\n            The Toddy Shop        92
                Desserts, Fast Food       3.4 Nashville\n
GoGourmet       34        Tea, Cafe, Indian, Mediterranean        3.7 Nashville\n
    Govinda's Confectionery       15     Tea, Bakery, Mediterranean, Fast Food
3.1 Nashville\n             Kitchen King       13    Pizza, Italian, BBQ, Cafe,
    Mediterranean        0.0 Nashville\n            Town Hall        55
Tea, Seafood          3.8 Nashville\n         Smoke House Deli        59            Cafe
    , Mediterranean, Seafood        4.0 Nashville\n        Dialogue
Lounge & Caf        83    Tea, Indian, BBQ, Cafe, American, Seafood        3.0 Nashville\n
            Meenakshi Bhawan        24                      Tea,
Desserts          3.1 Nashville\n             Oh! Calcutta        83          Tea,
    Mexican, BBQ, Cafe, Indian        4.4 Nashville\n            Chicago
Pizza        20            Tea, French, Pizza, BBQ       3.2 Nashville\n
    Malhotra Restaurant       76         Tea, Desserts, Seafood
2.9 Nashville\n            Madras Cafe        88           Pizza, Desserts, Seafood
            0.0 Nashville\nDakshin - Sheraton New Delhi Hotel       63
Cafe, Pizza, Indian, Bakery        4.0 Nashville\n    1911 Bar - The Imperial        76
    Desserts, Pizza, Italian, American, Seafood        3.2 Nashville\n
Punjabi Virsa       35       Desserts, Italian, BBQ, Cafe, Seafood        0.0 Nashville\n
            Sagar Dhaba       55     Tea, Bakery, BBQ, Cafe,
Mediterranean          0.0 Nashville\n          Kettle & Kegs        60
                Tea, Seafood        0.0 Nashville\n
Kargo       84          Tea, Pizza, Desserts       4.4 Nashville"}, {'
    Description': 'Accommodations in Nashville', 'Content': '
NAME price     room type                 house_rules minimum nights maximum occupancy
    review rate number  city\nHuge 2 Bedroom, Great Location, Express Metro
745.0 Entire home/apt No smoking & No children under 10 & No pets     4.0            4
            5.0 Nashville\n Clean and large bedroom in a private house 474.0
Private room                No smoking       1.0           1
    4.0 Nashville\n            Brooklyn Heights gem 993.0 Entire
home/apt                No pets        2.0          6          5.0
    Nashville\n    Lovely room in heart of Williamsburg 61.0 Private room
No pets        2.0           1           4.0 Nashville\n *Light & Love* vibrant,
    historic, sleeps 4 679.0 Entire home/apt            No pets
5.0          2          2.0 Nashville\n    FiDi Cozy room overlooking East River 870.0
    Private room               No parties       1.0
2          5.0 Nashville\n    Charming 1BR with sun-nook in Brooklyn 256.0 Entire home/apt No
    children under 10 & No pets & No smoking  3.0          2
4.0 Nashville\n        Cozy bedroom close to Manhattan 576.0 Private room            No
    children under 10        1.0          1          3.0
```

Nashville'}, {'Description': 'Attractions in Knoxville', 'Content': "
    Name Latitude Longitude                                               Address
Phone                                                                       Website
        City\n                      World's Fair Park 35.962577 -83.924192
525 Henley St, Knoxville, TN 37902, USA (865) 215-1158
                                                    http://worldsfairpark.org/
    Knoxville\n
Knoxville Museum of Art 35.962426 -83.925229 1050 Worlds Fair Park Dr, Knoxville, TN 37916, USA
    (865) 525-6101
http://www.knoxart.org/ Knoxville\n                      Sunsphere 35.961707 -83.923353
            810 Clinch Ave, Knoxville, TN 37902, USA (865) 314-0660
http://www.sunspheretickets.com/ Knoxville\n             Ijams Nature Center 35.956454
    -83.866775    2915 Island Home Ave, Knoxville, TN 37920, USA (865) 577-4717
http://www.ijams.org/ Knoxville\n            Knoxville Walking Tours 35.966448 -83.919167
            301 S Gay St, Knoxville, TN 37902, USA (865) 309-4522
http://knoxvillewalkingtours.com/ Knoxville\n                      Muse Knoxville 35.997617
    -83.885467        516 N Beaman St, Knoxville, TN 37914, USA (865) 594-1494
http://www.themuseknoxville.org/ Knoxville\n Knoxville Botanical Garden and Arboretum 35.982160
    -83.881077    2743 Wimpole Ave, Knoxville, TN 37914, USA (865) 862-8717
http://www.knoxgarden.org/ Knoxville\n        Haunted Knoxville Ghost Tours 35.965963 -83.919553
    36 Market Square #1404, Knoxville, TN 37902, USA (865) 377-9677
http://www.hauntedknoxville.net/ Knoxville\n                      Three Rivers Rambler 35.952810
    -83.940376 2560 University Commons Way, Knoxville, TN 37919, USA (865) 524-9411
http://www.threeriversrambler.com/ Knoxville\n             Charles Krutch Park 35.964414
    -83.918695        504 Market St, Knoxville, TN 37902, USA (865) 215-4248
https://www.knoxvilletn.gov/government/city_departments_offices/parks_and_recreation/parks/
    krutch_park Knoxville\n McClung Museum of Natural History & Culture 35.952005 -83.927209
1327 Cir Park Dr, Knoxville, TN 37996, USA (865) 974-2144
                                                    http://mcclungmuseum.utk.edu/
    Knoxville\n
Knoxville Sightseeing 35.992790 -83.904128        2519 Mitchell St, Knoxville, TN 37917, USA (865)
    566-0634
Unknown Knoxville\n      Chilhowee Park & Exposition Center 35.996296 -83.884032    3301 E
    Magnolia Ave, Knoxville, TN 37914, USA (865) 215-1450
https://chilhoweepark.com/ Knoxville\n                      Augusta Quarry 35.944772 -83.911886
    3000 Fort Dickerson Rd SW, Knoxville, TN 37920, USA Unknown
Unknown Knoxville\n                      Zoo Knoxville 35.999812 -83.888250    3500 Knoxville
    Zoo Dr, Knoxville, TN 37914, USA (865) 637-5331
https://www.zooknoxville.org/ Knoxville\n    Outdoor Knoxville Adventure Center 35.961902
    -83.912663 900 Volunteer Landing Ln, Knoxville, TN 37915, USA (865) 228-8424
http://www.outdoorknoxville.com Knoxville\n        James White Fort Association 35.962962
    -83.912169        205 E Hill Ave, Knoxville, TN 37915, USA (865) 525-6514
http://jameswhitesfort.org/ Knoxville\n    Rowing Man Statue in Knoxville, TN 35.963611 -83.917519
     W. Church Avenue &, S Gay St, Knoxville, TN 37902, USA Unknown
Unknown Knoxville\n                      UT Gardens Knoxville 35.944000 -83.938260            2518
    Jacob Dr, Knoxville, TN 37996, USA (865) 974-7324
http://utgardens.tennessee.edu/locations/knoxville Knoxville\nEast Tennessee Historical Society and
     Museum 35.964117 -83.917794   601 S Gay St, Knoxville, TN 37902, USA (865)
215-8830                                                     http://www.easttnhistory.
    org/ Knoxville"}, {'Description': 'Restaurants in Knoxville', 'Content': '
Name  Average Cost                                Cuisines Aggregate Rating    City\n
    Cafe Arabelle          29                French, BBQ, Desserts, Seafood
3.6 Knoxville\n      Les 3 Brasseurs           24          Desserts, Pizza, Italian, Cafe, Indian
            4.6 Knoxville\n        Sky On 57          150                Tea,
Cafe, Pizza, Chinese, Seafood          3.4 Knoxville\n        Cafe El Chico          67
                Tea, French, Indian, Seafood          3.6 Knoxville\n
TcozY          85          Tea, Pizza, Mexican, Fast Food, Cafe          0.0 Knoxville\nRama
    Vaishnav Bhojnalaya          46                Tea, Cafe, Pizza, Desserts

3.0 Knoxville\n        Burger Planet        45                Tea, Bakery, American, Cafe
        3.2 Knoxville\n    Cafe Coffee Day        82            Fast
Food, American, BBQ, Italian        2.8 Knoxville\n        Mamagoto        14
    Indian, Mediterranean, Desserts, Seafood        4.1 Knoxville\n  Ali Baba &
41 Dishes        70            Pizza, Desserts, Fast Food        3.5 Knoxville\n
    Punjabi Flavour        41        Tea, Bakery, Cafe, Indian, Seafood
0.0 Knoxville\n        Tandoori Tadka        23        Chinese, Pizza, Desserts, Fast Food
        0.0 Knoxville\n        Coalition Cafe        35        Desserts,
Tea, BBQ, Cafe, Mediterranean        3.4 Knoxville\n        Chaat Corner        67
                Cafe, Pizza        3.2 Knoxville\n
El Posto        63                Tea, Cafe, Pizza, BBQ        3.5 Knoxville\n
        La-Nawaab        17            French, Bakery, Cafe, Seafood
3.0 Knoxville\n    Peppers & Pipes        86            Chinese, Pizza, Cafe, Italian
        3.1 Knoxville\n        Chit Chat        15
Pizza, Fast Food        3.1 Knoxville\n        Anand Sweets        45        Fast Food,
    Pizza, Indian, Seafood        0.0 Knoxville\n        Open Kitchen
13 Pizza, Bakery, BBQ, Cafe, Indian, Mediterranean        3.0 Knoxville\n    Biryani By Kilo
        17            Chinese, Pizza, Fast Food        4.1
Knoxville\n    Chinese Food Corner        30            Indian, Desserts, Fast Food
        0.0 Knoxville\nThe Indian Kaffe Express        71
Tea, Desserts        3.8 Knoxville'}, {'Description': 'Accommodations in Knoxville', 'Content':
    '                NAME price        room type
house_rules minimum nights maximum occupancy review rate number  city\n  Cozy Private Room in
    Chinatown/ Lower East Side 132.0 Private room                No pets &
No visitors        7.0        2        4.0 Knoxville\n
            The Diamond Room 1008.0 Private room                No parties
    &
No visitors        1.0        1        5.0 Knoxville\n        Light-filled
    Room in Renovated Apt 310.0 Private room
No pets        2.0        2        2.0 Knoxville\n
            Private Room 922.0  Private room
            No
visitors        1.0        1        4.0 Knoxville\n        Beautiful &
    Private Manhattan Room 721.0 Private room            No parties & No
smoking        1.0        1        2.0 Knoxville\n    Lg Quiet Artist
    Home -Ditmas Park - 561.0 Entire home/apt    No visitors & No pets & No
smoking        24.0        4        2.0 Knoxville\n Beautiful 1- bdrm apt in
    tranquil Inwood building 1091.0 Entire home/apt        No smoking &
No pets        2.0        4        4.0 Knoxville\n 1,600sq ft modern duplex in
    new harlem brownstone 1159.0 Entire home/apt No smoking & No children under 10 &
No pets        2.0        5        2.0 Knoxville\n        Brooklyn Sunny room
    5 min to subway 793.0 Private room        No visitors & No children
under 10        2.0        2        5.0 Knoxville\n Amazing Large Sunny Studio in
    Greenwich Village 144.0 Entire home/apt        No parties & No
smoking        30.0        3        4.0 Knoxville\n    Private 1 Bdrm Suite in
    Historic Brownstone 479.0 Private room                No
visitors        2.0        2        2.0 Knoxville\n Charming bedroom with huge
    terrace in Greenpoint 712.0 Private room No parties & No children under 10 & No
visitors        2.0        1        3.0 Knoxville\n                Huge
    quiet modern oasis 1165.0 Entire home/apt        No smoking & No
parties        2.0        8        2.0 Knoxville\nPrivate large room near LGA
    airport with queen bed 552.0 Private room
No pets        1.0        1        2.0 Knoxville\n    12 east 86th Steet Upper
    East Side Building 632.0 Entire home/apt                No
smoking        30.0        3        2.0 Knoxville\n    Spacious Garden Apt in
    Clinton Hill/Ft Greene 1006.0 Entire home/apt No children under 10 & No visitors & No
smoking        3.0        3        3.0 Knoxville\n
            LES Penthouse 993.0 Entire home/apt        No children under
    10 & No

```
parties          2.0              8              4.0 Knoxville'}, {'Description': 'Attractions
    in Chattanooga', 'Content': "
Name Latitude Longitude                                    Address       Phone
Website      City\n                           The Chattanooga Zoo at Warner Park 35.042632
    -85.282099   301 N Holtzclaw Ave, Chattanooga, TN 37404, USA (423) 697-1322
http://www.chattzoo.org/ Chattanooga\n                                    Rock City Gardens
    34.973392 -85.350177    1400 Patten Rd, Lookout Mountain, GA 30750, USA (706)
820-2531                                       http://seerockcity.com/?utm_source=
    gmb&utm_medium=organic Chattanooga\n
Tennessee Aquarium 35.055823 -85.311065          1 Broad St, Chattanooga, TN 37402, USA (423)
    265-0695
http://www.tnaqua.org/ Chattanooga\n                      Lookout Mountain Incline Railway
    35.009546 -85.328564    3917 St Elmo Ave, Chattanooga, TN 37409, USA (423)
821-4224                                                               http://www.
    ridetheincline.com/ Chattanooga\n
Coolidge Park 35.060506 -85.306857           150 River St, Chattanooga, TN 37405, USA (423)
    643-6311
http://www.chattanooga.gov/public-works/parks/directory-of-park-facilities Chattanooga\n
                             Creative Discovery Museum 35.052700 -85.312280
321 Chestnut St, Chattanooga, TN 37402, USA (423) 756-2738
                                                                   http://www.
    cdmfun.org/
Chattanooga\n                          Chattanooga Ghost Tours Inc 35.051231 -85.309057
          57 E 5th St, Chattanooga, TN 37402, USA (423) 800-5998
https://chattanoogaghosttours.com/ Chattanooga\n                                   Ross's
    Landing 35.056741 -85.310302    201 Riverfront Pkwy, Chattanooga, TN 37402,
USA (423) 643-6311                          http://www.chattanooga.gov/public-works/
    parks/directory-of-park-facilities Chattanooga\n
Hunter Museum of American Art 35.055894 -85.306405    10 Bluff View Ave, Chattanooga, TN 37403,
    USA (423) 267-0968
http://www.huntermuseum.org/ Chattanooga\n                              Walnut Street
    Bridge 35.058330 -85.307296         1 Walnut St, Chattanooga, TN 37403, USA
(423) 643-6096                            http://www.chattanooga.gov/public-works/parks
    /directory-of-park-facilities Chattanooga\n
Tennessee Riverpark 35.095938 -85.245311    4301 Amnicola Hwy, Chattanooga, TN 37402, USA (423)
    209-5370
http://parks.hamiltontn.gov/ Chattanooga\n           Moccasin Bend National Archeological
    District 35.053024 -85.328656 301 Manufacturers Rd, Chattanooga, TN 37405, USA
(423) 648-5623                  https://www.nps.gov/chch/learn/historyculture/moccasin-bend-
    national-archeological-district.htm Chattanooga\n
Ruby Falls 35.019084 -85.339355         1720 Scenic Hwy, Chattanooga, TN 37409, USA (423) 821-2544
http://www.rubyfalls.com/ Chattanooga\n                                   Cravens House
    35.013782 -85.341495 Point Park Visitor Ctr, Lookout Mountain, TN 37350, USA
(423) 821-7786                                               https://www.nps.gov/
    places/cravenshouselom.htm Chattanooga\n
Williams Echo Dome 35.054215 -85.304836          430 E 2nd St, Chattanooga, TN 37403, USA
    Unknown
Unknown Chattanooga\n               Reflection Riding Arboretum & Nature Center 35.010130
    -85.364585    400 Garden Rd, Chattanooga, TN 37419, USA (423) 821-1160
https://www.reflectionriding.org/ Chattanooga\n                                  Audubon
    Acres 34.999350 -85.180340    900 N Sanctuary Rd, Chattanooga, TN 37421,
USA (423) 892-1499                                                        http
    ://www.chattanoogaaudubon.org/ Chattanooga\n
Sculpture Fields at Montague Park 35.026175 -85.293624          1800 Polk St, Chattanooga, TN 37408,
     USA (423) 266-7288
http://www.sculpturefields.org/ Chattanooga\nBessie Smith Cultural Center and Chattanooga African
    American Museum 35.044634 -85.306467 200 E M L King Blvd, Chattanooga, TN 37403,
USA (423) 266-8658 http://www.bessiesmithcc.org/?utm_source=google&utm_medium=
    wix_google_business_profile&utm_campaign=3948799342172393035 Chattanooga\n
```

Wooden Horse 35.055836 -85.306713     10 Bluff View Ave, Chattanooga, TN 37403, USA   Unknown
Unknown Chattanooga"}, {'Description': 'Restaurants in Chattanooga', 'Content': "            Name
     Average Cost                            Cuisines Aggregate
Rating     City\n           P.F. Chang's           33                     Cafe, Bakery, BBQ,
    Seafood       4.2 Chattanooga\n           L'amandier
28             Cafe, BBQ, Mediterranean, Seafood         4.3 Chattanooga\n
    Anupam Sweet       82                     Desserts, American, BBQ
3.4 Chattanooga\n           Nirula's       64                     French, Desserts,
    Seafood       2.6 Chattanooga\n Anupam Sweets & Restaurant       62
Tea, Cafe, Desserts, Seafood         3.0 Chattanooga\n         Bisque Bakery       90
                French, Desserts, Seafood         3.9 Chattanooga\n
Nooba       93                     Tea, Mexican, Bakery, Seafood         3.8 Chattanooga\n
          The Royal       39               French, Pizza,
Bakery, Fast Food       3.3 Chattanooga\n       Curry n Phulka       40
                Tea, Chinese, Fast Food       2.8 Chattanooga\n DCK-
Dana Choga's Kitchen       67                     Cafe, Pizza, Mediterranean       3.5
    Chattanooga\n           Chaayos       54
Desserts, Cafe, BBQ, Chinese, Seafood       3.3 Chattanooga\n           Truffles
    53       Cafe, Bakery, BBQ, Fast Food, Chinese, American       3.2
Chattanooga\n           Liquid       15       Tea, Pizza, Bakery, Mediterranean,
    Seafood       4.0 Chattanooga\n           Tpot       14
Desserts, Tea, Pizza, Mexican, Cafe, Indian       0.0 Chattanooga\n   Sardar A Pure Meat Shop
       24               Bakery, Pizza, American
3.4 Chattanooga\n           Warehouse Cafe       90                     Fast Food, Pizza,
    Seafood       3.7 Chattanooga\n           The Beer Cafe       62
Tea, Pizza, Fast Food         3.8 Chattanooga\n           Subway       66
                Tea, Cafe, Bakery, Desserts       2.4 Chattanooga\n
Indi-QUE       57               Tea, Italian, BBQ, Fast Food, Cafe       3.3 Chattanooga\
    nEssex Collections Patisserie       48
Chinese, BBQ, Fast Food       3.4 Chattanooga\n           Food Adda       100
    Desserts, Pizza, Fast Food, Mediterranean, Seafood       0.0 Chattanooga\n
Bikanervala       65                     Cafe, Bakery       3.2
    Chattanooga\n       Havmor Ice Cream       75       Pizza, Bakery, Fast
Food, Chinese, Seafood       3.6 Chattanooga\n       Moti Mahal Delux       52
                Tea, Pizza, BBQ, Cafe, Mediterranean       2.7 Chattanooga\n
Aggarwal Sweet India       30                     Tea, Desserts       2.8
    Chattanooga\n       Jack Po!tato's       63               Tea,
Bakery, Mediterranean, Seafood       3.3 Chattanooga\n           Muradabadi       16
                Cafe, Pizza, Bakery, BBQ       3.0
Chattanooga\n       Vishal Restaurant       27                     Tea, Pizza, BBQ,
    Seafood       3.1 Chattanooga\n       Raj Sweets       69
Desserts, Tea, BBQ, Cafe, Mediterranean       2.8 Chattanooga\n       Punjabee's Darbar
       41           Tea, French, Bakery, Fast Food, Chinese       3.0
Chattanooga\n           Basil Tree       20               Bakery, Indian, American,
    Desserts       2.4 Chattanooga\n       Domino's Pizza       77
Bakery, Pizza, American, BBQ       2.8 Chattanooga\n       Pizza Hut Delivery       25
                Tea, Cafe, American, Indian       3.9 Chattanooga\n
Ichiban       87                     Mexican, Bakery, Desserts       3.9 Chattanooga\n
          Habibi       95
Cafe, BBQ, Seafood       3.7 Chattanooga\n           Cook Du Kdu       69 Pizza, Fast
    Food, Cafe, Indian, Mediterranean, Seafood       3.1 Chattanooga\n
Go Krazy       64                     French, BBQ, Fast Food       3.5 Chattanooga\
    n           Bentoya       41               Tea, Fast
Food, Bakery, Seafood       3.8 Chattanooga\n       Burger's King       16
                Tea, Bakery, Desserts       4.1 Chattanooga"},
{'Description': 'Accommodations in Chattanooga', 'Content': '                     NAME
     price     room type                     house_rules minimum
nights maximum occupancy review rate number     city\n   Affordable Private Spacious Room in
    Brooklyn 790.0 Private room               No parties

2.0                2                3.0 Chattanooga\n                                Sunny One Bedroom
    722.0 Entire home/apt        No pets & No children under 10        1.0
5                3.0 Chattanooga\nUpper West / Morningside Heights Apt, Near Subway 290.0 Entire
    home/apt                No visitors        3.0                3
3.0 Chattanooga\n            Sunny room+Pvte office in huge loft 728.0 Private room
                        No parties        4.0                2                5.0
Chattanooga\n        Extra Cozy Room in Center of Williamsburg 1033.0 Private room
                        No pets        1.0                1                1.0
Chattanooga\n    Luxury & Charm. Steps from Christopher Park! 301.0 Entire home/apt        No pets & No
    visitors & No parties        5.0                2                4.0
Chattanooga\n                                Fort Greene Room 722.0    Private room        No visitors &
        No children under 10        2.0                1                2.0
Chattanooga\n                                [Unicode Sequence Edited Out] 973.0 Entire home/apt
                        No pets        10.0                6
4.0 Chattanooga\n    4 bdrm/2 bath apt. Central Pk, Columbia U. 564.0 Entire home/apt
                No visitors & No pets        30.0                4                3.0
Chattanooga\n        Cozy room in Bushwick- 15 min to the city 933.0 Private room
    No parties & No pets        1.0                2                2.0
Chattanooga\n                Modern apartment w/ gorgeous view 921.0 Private room
                        No pets        3.0                2                1.0
Chattanooga\n    Artsy Private BR in Fort Greene Cumberland 727.0 Private room
                        No visitors        30.0                1                3.0
Chattanooga\n        Studio Deluxe 1 - Wyndham Midtown 45 848.0 Private room No pets & No smoking
    & No children under 10        3.0                1                2.0
Chattanooga\n        Trendy and Stylish Downtown apartment !!! 110.0 Entire home/apt
                        No pets        3.0                2                2.0
Chattanooga\n    Sonder | 180 Water | Luxurious 1BR + Rooftop 676.0 Entire home/apt
    No pets & No smoking        29.0                5                3.0
Chattanooga'}, {'Description': 'Flight from Houston to Nashville on 2022-03-21', 'Content': 'Flight
    Number Price DepTime ArrTime ActualElapsedTime FlightDate OriginCityName DestCityName
Distance\n    F3827247    241    14:22    16:10 1 hours 48 minutes 2022-03-21    Houston    Nashville
    657.0\n    F3827724    147    18:35    20:33 1 hours 58 minutes 2022-03-21
Houston    Nashville    657.0\n    F3827820    157    12:07    14:02 1 hours 55 minutes 2022-03-21
    Houston    Nashville    657.0\n    F3827930    305    07:36    09:41 2 hours 5
minutes 2022-03-21    Houston    Nashville    657.0\n    F3956407    264    11:38    13:15 1 hours 37
    minutes 2022-03-21    Houston    Nashville    670.0\n    F3956408    246
08:08    09:47 1 hours 39 minutes 2022-03-21    Houston    Nashville    670.0\n    F3956409    145
    17:36    19:12 1 hours 36 minutes 2022-03-21    Houston    Nashville    670.0\n
F3956532    225    07:14    08:52 1 hours 38 minutes 2022-03-21    Houston    Nashville    657.0\n
    F3956533    292    11:42    13:19 1 hours 37 minutes 2022-03-21    Houston
Nashville    657.0\n    F3956534    248    18:20    20:06 1 hours 46 minutes 2022-03-21    Houston
    Nashville    657.0\n    F4038756    272    10:11    11:52 1 hours 41 minutes
2022-03-21    Houston    Nashville    657.0\n    F4038966    149    16:33    18:13 1 hours 40 minutes
    2022-03-21    Houston    Nashville    657.0'}, {'Description': 'Self-driving
from Houston to Nashville', 'Content': 'self-driving, from Houston to Nashville, duration: 11 hours
    50 mins, distance: 1,253 km, cost: 62'}, {'Description': 'Taxi from Houston to
Nashville', 'Content': 'taxi, from Houston to Nashville, duration: 11 hours 50 mins, distance:
    1,253 km, cost: 1253'}, {'Description': 'Flight from Nashville to Knoxville on 2022-03-23',
'Content': 'There is no flight from Nashville to Knoxville on 2022-03-23.'}, {'Description': 'Self-
    driving from Nashville to Knoxville', 'Content': 'self-driving, from Nashville to
Knoxville, duration: 2 hours 42 mins, distance: 290 km, cost: 14'}, {'Description': 'Taxi from
    Nashville to Knoxville', 'Content': 'taxi, from Nashville to Knoxville, duration: 2 hours 42
mins, distance: 290 km, cost: 290'}, {'Description': 'Flight from Knoxville to Chattanooga on
    2022-03-25', 'Content': 'There is no flight from Knoxville to Chattanooga on 2022-03-25.'},
{'Description': 'Self-driving from Knoxville to Chattanooga', 'Content': 'self-driving, from
    Knoxville to Chattanooga, duration: 1 hour 41 mins, distance: 180 km, cost: 9'}, {'Description
    ':
'Taxi from Knoxville to Chattanooga', 'Content': 'taxi, from Knoxville to Chattanooga, duration: 1
    hour 41 mins, distance: 180 km, cost: 180'}, {'Description': 'Flight from Chattanooga to

Houston on 2022-03-27', 'Content': 'There is no flight from Chattanooga to Houston on
    2022-03-27.'}, {'Description': 'Self-driving from Chattanooga to Houston', 'Content': 'self-
    driving,
from Chattanooga to Houston, duration: 11 hours 47 mins, distance: 1,309 km, cost: 65'}, {'
    Description': 'Taxi from Chattanooga to Houston', 'Content': 'taxi, from Chattanooga to Houston
    ,
duration: 11 hours 47 mins, distance: 1,309 km, cost: 1309'}]
Query: Could you design a one-week travel itinerary for two, departing from Houston and touring
    three cities in Tennessee from March 21st to March 27th, 2022? Our budget is now $8,200. We
require accommodations that allow smoking and should ideally be private rooms. As for
    transportation, we would prefer not to self-drive.
Travel Plan (please only output the JSON string without explanatory information):

==============RESPONSE OF o1-preview=================

```
[
    {
        "day": 1,
        "people_number": 2,
        "current_city": "from Houston to Nashville",
        "transportation": "Flight Number: F3827820, from Houston to Nashville, Departure Time:
            12:07, Arrival Time: 14:02",
        "breakfast": "-",
        "attraction": "Country Music Hall of Fame and Museum, Nashville",
        "lunch": "Cafe Arabelle, Nashville",
        "dinner": "Les 3 Brasseurs, Nashville",
        "accommodation": "FiDi Cozy room overlooking East River, Nashville"
    },
    {
        "day": 2,
        "people_number": 2,
        "current_city": "Nashville",
        "transportation": "-",
        "breakfast": "Burger Planet, Nashville",
        "attraction": "Johnny Cash Museum; Centennial Park, Nashville",
        "lunch": "Mamagoto, Nashville",
        "dinner": "Oh! Calcutta, Nashville",
        "accommodation": "FiDi Cozy room overlooking East River, Nashville"
    },
    {
        "day": 3,
        "people_number": 2,
        "current_city": "from Nashville to Knoxville",
        "transportation": "Taxi, from Nashville to Knoxville, Departure Time: 09:00, Arrival Time:
            11:42, Duration: 2 hours 42 minutes, Cost: $290",
        "breakfast": "Cafe El Chico, Nashville",
        "attraction": "World's Fair Park, Knoxville",
        "lunch": "Ali Baba & 41 Dishes, Knoxville",
        "dinner": "Sky On 57, Knoxville",
        "accommodation": "Light-filled Room in Renovated Apt, Knoxville"
    },
    {
        "day": 4,
        "people_number": 2,
        "current_city": "Knoxville",
        "transportation": "-",
        "breakfast": "Chaat Corner, Knoxville",
        "attraction": "Ijams Nature Center; Sunsphere, Knoxville",
        "lunch": "Anand Sweets, Knoxville",
```

```
            "dinner": "La-Nawaab, Knoxville",
            "accommodation": "Light-filled Room in Renovated Apt, Knoxville"
        },
        {
            "day": 5,
            "people_number": 2,
            "current_city": "from Knoxville to Chattanooga",
            "transportation": "Taxi, from Knoxville to Chattanooga, Departure Time: 09:00, Arrival Time:
                10:41, Duration: 1 hour 41 minutes, Cost: $180",
            "breakfast": "Anand Sweets, Knoxville",
            "attraction": "Tennessee Aquarium, Chattanooga",
            "lunch": "Anupam Sweet, Chattanooga",
            "dinner": "Bisque Bakery, Chattanooga",
            "accommodation": "Trendy and Stylish Downtown apartment !!!, Chattanooga"
        },
        {
            "day": 6,
            "people_number": 2,
            "current_city": "Chattanooga",
            "transportation": "-",
            "breakfast": "Anupam Sweets & Restaurant, Chattanooga",
            "attraction": "Rock City Gardens; Ruby Falls, Chattanooga",
            "lunch": "Chaayos, Chattanooga",
            "dinner": "L'amandier, Chattanooga",
            "accommodation": "Trendy and Stylish Downtown apartment !!!, Chattanooga"
        },
        {
            "day": 7,
            "people_number": 2,
            "current_city": "from Chattanooga to Houston",
            "transportation": "Taxi, from Chattanooga to Houston, Departure Time: 08:00, Arrival Time:
                19:47, Duration: 11 hours 47 minutes, Cost: $1309",
            "breakfast": "Bisque Bakery, Chattanooga",
            "attraction": "Coolidge Park, Chattanooga",
            "lunch": "Nooba, Chattanooga",
            "dinner": "-",
            "accommodation": "-"
        }
]
```

```
==============FEEDBACK FROM METACONTROLLER=================
By inspecting your plan we find the following issue. Please refine your plan according to the
    feedback below:
1. The room type should be private room :-> Trendy and Stylish Downtown apartment !!!
2. The restaurant in day 5 breakfast is repeated.
3. The accommodation Trendy and Stylish Downtown apartment !!!, Chattanooga do not obey the minumum
    nights rule.
4. The lunch in day 1 is invalid or not in the data provided.
```

## Q.2  Back Prompt

```
You are a proficient planner. Based on the provided information and query, please give me a
    detailed plan, including specifics such as flight numbers (e.g., F0123456), restaurant names,
    and
accommodation names. Note that all the information in your plan should be derived from the provided
    data. You should give a travel plan in JSON format as shown in the example below.
Additionally, all details should align with commonsense. The symbol '-' indicates that information
    is unnecessary. For example, in the provided sample, you do not need to plan after
returning to the departure city. When you travel to two cities in one day, you should note it in
    the 'current_city' section as in the example (i.e., from A to B).
```

```
***** Example *****
Query: Could you create a travel plan for 7 people from Ithaca to Charlotte spanning 3 days, from
    March 8th to March 14th, 2022, with a budget of $30,200?
Travel Plan:
[
    {
        "day": 1,
        "people_number": 7,
        "current_city": "from Ithaca to Charlotte",
        "transportation": "Flight Number: F3633413, from Ithaca to Charlotte, Departure Time: 05:38,
                Arrival Time: 07:46",
        "breakfast": "Nagaland's Kitchen, Charlotte",
        "attraction": "The Charlotte Museum of History, Charlotte",
        "lunch": "Cafe Maple Street, Charlotte",
        "dinner": "Bombay Vada Pav, Charlotte",
        "accommodation": "Affordable Spacious Refurbished Room in Bushwick!, Charlotte"
    },
    {
        "day": 2,
        "people_number": 7,
        "current_city": "Charlotte",
        "transportation": "-",
        "breakfast": "Olive Tree Cafe, Charlotte",
        "attraction": "The Mint Museum, Charlotte;Romare Bearden Park, Charlotte",
        "lunch": "Birbal Ji Dhaba, Charlotte",
        "dinner": "Pind Balluchi, Charlotte",
        "accommodation": "Affordable Spacious Refurbished Room in Bushwick!, Charlotte"
    },
    {
        "day": 3,
        "people_number": 7,
        "current_city": "from Charlotte to Ithaca",
        "transportation": "Flight Number: F3786167, from Charlotte to Ithaca, Departure Time: 21:42,
                Arrival Time: 23:26",
        "breakfast": "Books Monument, Charlotte",
        "attraction": "Books Monument, Charlotte",
        "lunch": "Olive Tree Cafe, Charlotte",
        "dinner": "Kylin Skybar, Charlotte",
        "accommodation": "-"
    }
]
***** Example Ends *****

Given information: [{'Description': 'Attractions in Nashville', 'Content': '
    Name Latitude Longitude                                            Address
Phone
Website     City\nCountry Music Hall of Fame and Museum 36.158263 -86.776126 222 Rep. John Lewis
    Way S, Nashville, TN 37203, USA (615) 416-2001
https://countrymusichalloffame.org/ Nashville\n     Nashville Zoo at Grassmere 36.089705 -86.742096
        3777 Nolensville Pk, Nashville, TN 37211, USA (615) 833-1534
http://www.nashvillezoo.org/ Nashville\n Belle Meade Historic Site & Winery 36.104916 -86.864695
    5025 Harding Pike, Nashville, TN 37205, USA (615) 356-0501
https://visitbellemeade.com/ Nashville\n              Johnny Cash Museum 36.160939 -86.775757
        119 3rd Ave S, Nashville, TN 37201, USA (615) 256-1777
http://www.johnnycashmuseum.com/ Nashville\n               Centennial Park 36.148946 -86.812750
        2500 West End Ave, Nashville, TN 37203, USA (615) 862-8400
https://www.nashville.gov/Parks-and-Recreation/Parks/Centennial-Park.aspx Nashville\n
    Grand Ole Opry 36.206857 -86.692108      600 Opry Mills Dr, Nashville, TN
```

37214, USA (615) 871-6779

https://www.opry.com/?utm_campaign=opry&utm_medium=organicsearch&utm_source=googlemybusiness&
    utm_audience=tofu_googlemybusiness&utm_content=brandstory_google_my_business_website_link
Nashville\n                    Frist Art Museum 36.157897 -86.783853        919 Broadway, Nashville,
    TN 37203, USA (615) 244-3340

https://fristartmuseum.org/ Nashville\n                    The Parthenon 36.149674 -86.813347    2500
    West End Ave, Nashville, TN 37203, USA (615) 862-8431

https://www.nashvilleparthenon.com/ Nashville\n              Nashville Shores 36.158209 -86.604915
        4001 Bell Rd, Hermitage, TN 37076, USA (615) 889-7050

http://www.nashvilleshores.com/lodging Nashville\n Musicians Hall of Fame and Museum 36.167668
    -86.782399      401 Gay St, Nashville, TN 37219, USA (615) 244-3263

https://www.musicianshalloffame.com/ Nashville\n Bicentennial Capitol Mall State Park 36.170887
    -86.787589 600 James Robertson Pkwy, Nashville, TN 37243, USA (888) 867-2757

https://tnstateparks.com/parks/bicentennial-mall Nashville\n        Lane Motor Museum 36.140197
    -86.734580 702 Murfreesboro Pike, Nashville, TN 37210, USA (615) 742-7445

http://www.lanemotormuseum.org/ Nashville\n           Honky Tonk Highway 36.160393 -86.778432
        501 Broadway, Nashville, TN 37203, USA (800) 657-6910

https://www.visitmusiccity.com/visitors/honkytonkhighway Nashville\n    Adventure Science Center
    36.146614 -86.775481 800 Fort Negley Blvd, Nashville, TN 37203, USA (615)
862-5160

http://www.adventuresci.org/ Nashville\n            Patsy Cline Museum 36.160904 -86.776030
        119 3rd Ave S, Nashville, TN 37201, USA (615) 454-4722

https://www.patsymuseum.com/ Nashville\n    Andrew Jackson's Hermitage 36.213757 -86.615310
    4580 Rachels Ln, Hermitage, TN 37076, USA (615) 889-2941

http://www.thehermitage.com/ Nashville\n              NashTrash Tours 36.169474 -86.788254 900
    Rosa L Parks Blvd, Nashville, TN 37208, USA (615) 226-7300

http://www.nashtrash.com/ Nashville\n              Fort Nashborough 36.164268 -86.775430
    170 1st Ave N, Nashville, TN 37201, USA (615) 862-8400

https://www.nashville.gov/Parks-and-Recreation/Historic-Sites/Fort-Nashborough.aspx Nashville\n
        Ryman Auditorium 36.161248 -86.778471        116 5th Ave N,
Nashville, TN 37219, USA (615) 889-3060

https://www.ryman.com/?utm_campaign=ryman&utm_medium=organicsearch&utm_source=googlemybusiness&
    utm_audience=tofu_googlemybusiness&utm_content=brandstory_google_my_business_website_link
Nashville\n      Nashville Public Square Park 36.166611 -86.778126 Union St & 3rd Ave N, Nashville
    , TN 37201, USA (615) 743-3090

https://www.nashvilledowntown.com/go/public-square-park Nashville'}, {'Description': 'Restaurants
    in Nashville', 'Content': "          Name  Average Cost
Cuisines Aggregate Rating    City\n                      Bangkok 1        44
                    Cafe, Seafood        3.3 Nashville\n            Bablu Fast
Food        13    Desserts, Pizza, Mexican, BBQ, Fast Food        0.0 Nashville\n
                Full Dabba        77   Pizza, Mexican, Fast Food, Cafe, American
0.0 Nashville\n                Twigly        14 Pizza, French, Fast Food, Chinese,
    Seafood    4.5 Nashville\n                Veg Hut        35
Tea, Mexican, Seafood        0.0 Nashville\n          The Toddy Shop        92
                Desserts, Fast Food        3.4 Nashville\n
GoGourmet        34        Tea, Cafe, Indian, Mediterranean        3.7 Nashville\n
    Govinda's Confectionery        15    Tea, Bakery, Mediterranean, Fast Food
3.1 Nashville\n            Kitchen King        13    Pizza, Italian, BBQ, Cafe,
    Mediterranean        0.0 Nashville\n            Town Hall        55
Tea, Seafood        3.8 Nashville\n        Smoke House Deli        59        Cafe
    , Mediterranean, Seafood        4.0 Nashville\n        Dialogue
Lounge & Caf        83   Tea, Indian, BBQ, Cafe, American, Seafood        3.0 Nashville\n
        Meenakshi Bhawan        24                Tea,
Desserts        3.1 Nashville\n            Oh! Calcutta        83        Tea,
    Mexican, BBQ, Cafe, Indian        4.4 Nashville\n        Chicago
Pizza        20            Tea, French, Pizza, BBQ        3.2 Nashville\n
    Malhotra Restaurant        76        Tea, Desserts, Seafood
2.9 Nashville\n            Madras Cafe        88        Pizza, Desserts, Seafood
        0.0 Nashville\nDakshin - Sheraton New Delhi Hotel        63

Cafe, Pizza, Indian, Bakery          4.0 Nashville\n        1911 Bar - The Imperial        76
    Desserts, Pizza, Italian, American, Seafood        3.2 Nashville\n
Punjabi Virsa          35        Desserts, Italian, BBQ, Cafe, Seafood          0.0 Nashville\n
                    Sagar Dhaba          55        Tea, Bakery, BBQ, Cafe,
Mediterranean          0.0 Nashville\n            Kettle & Kegs          60
                    Tea, Seafood          0.0 Nashville\n
Kargo          84                    Tea, Pizza, Desserts          4.4 Nashville"}, {'
    Description': 'Accommodations in Nashville', 'Content': '
NAME price      room type                              house_rules minimum nights maximum occupancy
    review rate number  city\nHuge 2 Bedroom, Great Location, Express Metro
745.0 Entire home/apt No smoking & No children under 10 & No pets      4.0              4
                5.0 Nashville\n Clean and large bedroom in a private house 474.0
Private room                          No smoking          1.0              1
    4.0 Nashville\n                Brooklyn Heights gem 993.0 Entire
home/apt                          No pets          2.0              6          5.0
    Nashville\n      Lovely room in heart of Williamsburg 61.0 Private room
No pets          2.0              1              4.0 Nashville\n *Light & Love* vibrant,
    historic, sleeps 4 679.0 Entire home/apt                    No pets
5.0              2              2.0 Nashville\n     FiDi Cozy room overlooking East River 870.0
      Private room                    No parties          1.0
2              5.0 Nashville\n    Charming 1BR with sun-nook in Brooklyn 256.0 Entire home/apt No
    children under 10 & No pets & No smoking   3.0              2
4.0 Nashville\n        Cozy bedroom close to Manhattan 576.0 Private room                    No
    children under 10          1.0              1          3.0
Nashville'}, {'Description': 'Attractions in Knoxville', 'Content': "
    Name Latitude Longitude                              Address
Phone                                                          Website
      City\n                    World's Fair Park 35.962577 -83.924192
525 Henley St, Knoxville, TN 37902, USA (865) 215-1158
                                        http://worldsfairpark.org/
    Knoxville\n
Knoxville Museum of Art 35.962426 -83.925229 1050 Worlds Fair Park Dr, Knoxville, TN 37916, USA
    (865) 525-6101
http://www.knoxart.org/ Knoxville\n                    Sunsphere 35.961707 -83.923353
          810 Clinch Ave, Knoxville, TN 37902, USA (865) 314-0660
http://www.sunspheretickets.com/ Knoxville\n                Ijams Nature Center 35.956454
    -83.866775    2915 Island Home Ave, Knoxville, TN 37920, USA (865) 577-4717
http://www.ijams.org/ Knoxville\n            Knoxville Walking Tours 35.966448 -83.919167
          301 S Gay St, Knoxville, TN 37902, USA (865) 309-4522
http://knoxvillewalkingtours.com/ Knoxville\n                    Muse Knoxville 35.997617
    -83.885467        516 N Beaman St, Knoxville, TN 37914, USA (865) 594-1494
http://www.themuseknoxville.org/ Knoxville\n Knoxville Botanical Garden and Arboretum 35.982160
    -83.881077    2743 Wimpole Ave, Knoxville, TN 37914, USA (865) 862-8717
http://www.knoxgarden.org/ Knoxville\n        Haunted Knoxville Ghost Tours 35.965963 -83.919553
    36 Market Square #1404, Knoxville, TN 37902, USA (865) 377-9677
http://www.hauntedknoxville.net/ Knoxville\n                Three Rivers Rambler 35.952810
    -83.940376 2560 University Commons Way, Knoxville, TN 37919, USA (865) 524-9411
http://www.threeriversrambler.com/ Knoxville\n                Charles Krutch Park 35.964414
    -83.918695        504 Market St, Knoxville, TN 37902, USA (865) 215-4248
https://www.knoxvilletn.gov/government/city_departments_offices/parks_and_recreation/parks/
    krutch_park Knoxville\n McClung Museum of Natural History & Culture 35.952005 -83.927209
1327 Cir Park Dr, Knoxville, TN 37996, USA (865) 974-2144
                                        http://mcclungmuseum.utk.edu/
    Knoxville\n
Knoxville Sightseeing 35.992790 -83.904128        2519 Mitchell St, Knoxville, TN 37917, USA (865)
    566-0634
Unknown Knoxville\n     Chilhowee Park & Exposition Center 35.996296 -83.884032    3301 E
    Magnolia Ave, Knoxville, TN 37914, USA (865) 215-1450

https://chilhoweepark.com/ Knoxville\n                            Augusta Quarry 35.944772 -83.911886
    3000 Fort Dickerson Rd SW, Knoxville, TN 37920, USA Unknown
Unknown Knoxville\n                            Zoo Knoxville 35.999812 -83.888250    3500 Knoxville
    Zoo Dr, Knoxville, TN 37914, USA (865) 637-5331
https://www.zooknoxville.org/ Knoxville\n    Outdoor Knoxville Adventure Center 35.961902
    -83.912663 900 Volunteer Landing Ln, Knoxville, TN 37915, USA (865) 228-8424
http://www.outdoorknoxville.com Knoxville\n        James White Fort Association 35.962962
    -83.912169        205 E Hill Ave, Knoxville, TN 37915, USA (865) 525-6514
http://jameswhitesfort.org/ Knoxville\n    Rowing Man Statue in Knoxville, TN 35.963611 -83.917519
    W. Church Avenue &, S Gay St, Knoxville, TN 37902, USA Unknown
Unknown Knoxville\n                    UT Gardens Knoxville 35.944000 -83.938260            2518
    Jacob Dr, Knoxville, TN 37996, USA (865) 974-7324
http://utgardens.tennessee.edu/locations/knoxville Knoxville\nEast Tennessee Historical Society and
    Museum 35.964117 -83.917794   601 S Gay St, Knoxville, TN 37902, USA (865)
215-8830                                        http://www.easttnhistory.
    org/ Knoxville"}, {'Description': 'Restaurants in Knoxville', 'Content': '
Name  Average Cost                            Cuisines Aggregate Rating   City\n
    Cafe Arabelle        29            French, BBQ, Desserts, Seafood
3.6 Knoxville\n     Les 3 Brasseurs        24        Desserts, Pizza, Italian, Cafe, Indian
            4.6 Knoxville\n        Sky On 57        150            Tea,
Cafe, Pizza, Chinese, Seafood        3.4 Knoxville\n      Cafe El Chico        67
                Tea, French, Indian, Seafood        3.6 Knoxville\n
TcozY        85        Tea, Pizza, Mexican, Fast Food, Cafe        0.0 Knoxville\nRama
    Vaishnav Bhojnalaya        46            Tea, Cafe, Pizza, Desserts
3.0 Knoxville\n        Burger Planet        45            Tea, Bakery, American, Cafe
            3.2 Knoxville\n     Cafe Coffee Day        82            Fast
Food, American, BBQ, Italian        2.8 Knoxville\n        Mamagoto        14
    Indian, Mediterranean, Desserts, Seafood        4.1 Knoxville\n   Ali Baba &
41 Dishes        70            Pizza, Desserts, Fast Food        3.5 Knoxville\n
    Punjabi Flavour        41        Tea, Bakery, Cafe, Indian, Seafood
0.0 Knoxville\n        Tandoori Tadka        23        Chinese, Pizza, Desserts, Fast Food
        0.0 Knoxville\n     Coalition Cafe        35        Desserts,
Tea, BBQ, Cafe, Mediterranean        3.4 Knoxville\n        Chaat Corner        67
                Cafe, Pizza        3.2 Knoxville\n
El Posto        63            Tea, Cafe, Pizza, BBQ        3.5 Knoxville\n
        La-Nawaab        17        French, Bakery, Cafe, Seafood
3.0 Knoxville\n     Peppers & Pipes        86            Chinese, Pizza, Cafe, Italian
        3.1 Knoxville\n        Chit Chat        15
Pizza, Fast Food        3.1 Knoxville\n     Anand Sweets        45            Fast Food,
    Pizza, Indian, Seafood        0.0 Knoxville\n     Open Kitchen
13 Pizza, Bakery, BBQ, Cafe, Indian, Mediterranean        3.0 Knoxville\n     Biryani By Kilo
        17            Chinese, Pizza, Fast Food        4.1
Knoxville\n  Chinese Food Corner        30            Indian, Desserts, Fast Food
        0.0 Knoxville\nThe Indian Kaffe Express        71
Tea, Desserts        3.8 Knoxville'}, {'Description': 'Accommodations in Knoxville', 'Content':
    '                    NAME price        room type
house_rules minimum nights maximum occupancy review rate number  city\n  Cozy Private Room in
    Chinatown/ Lower East Side 132.0 Private room            No pets &
No visitors        7.0        2        4.0 Knoxville\n
                    The Diamond Room 1008.0 Private room            No parties
    &
No visitors        1.0        1        5.0 Knoxville\n        Light-filled
    Room in Renovated Apt 310.0 Private room
No pets        2.0        2        2.0 Knoxville\n
                    Private Room 922.0  Private room
                    No
visitors        1.0        1        4.0 Knoxville\n        Beautiful &
    Private Manhattan Room 721.0 Private room            No parties & No

```
smoking          1.0                   1                2.0 Knoxville\n         Lg Quiet Artist
    Home -Ditmas Park - 561.0 Entire home/apt         No visitors & No pets & No
smoking          24.0                  4                2.0 Knoxville\n Beautiful 1- bdrm apt in
    tranquil Inwood building 1091.0 Entire home/apt              No smoking &
No pets          2.0                   4                4.0 Knoxville\n 1,600sq ft modern duplex in
    new harlem brownstone 1159.0 Entire home/apt No smoking & No children under 10 &
No pets          2.0                   5                2.0 Knoxville\n         Brooklyn Sunny room
     5 min to subway 793.0 Private room         No visitors & No children
under 10         2.0                   2                5.0 Knoxville\n Amazing Large Sunny Studio in
    Greenwich Village 144.0 Entire home/apt             No parties & No
smoking          30.0                  3                4.0 Knoxville\n     Private 1 Bdrm Suite in
    Historic Brownstone 479.0 Private room                          No
visitors         2.0                   2                2.0 Knoxville\n Charming bedroom with huge
    terrace in Greenpoint 712.0 Private room No parties & No children under 10 & No
visitors         2.0                   1                3.0 Knoxville\n               Huge
    quiet modern oasis 1165.0 Entire home/apt             No smoking & No
parties          2.0                   8                2.0 Knoxville\nPrivate large room near LGA
    airport with queen bed 552.0 Private room
No pets          1.0                   1                2.0 Knoxville\n    12 east 86th Steet Upper
    East Side Building 632.0 Entire home/apt                        No
smoking          30.0                  3                2.0 Knoxville\n   Spacious Garden Apt in
    Clinton Hill/Ft Greene 1006.0 Entire home/apt No children under 10 & No visitors & No
smoking          3.0                   3                3.0 Knoxville\n
                                LES Penthouse 993.0 Entire home/apt        No children under
    10 & No
parties          2.0                   8                4.0 Knoxville'}, {'Description': 'Attractions
    in Chattanooga', 'Content': "
Name Latitude Longitude                                     Address       Phone
Website      City\n                        The Chattanooga Zoo at Warner Park 35.042632
    -85.282099   301 N Holtzclaw Ave, Chattanooga, TN 37404, USA (423) 697-1322
http://www.chattzoo.org/ Chattanooga\n                            Rock City Gardens
    34.973392 -85.350177    1400 Patten Rd, Lookout Mountain, GA 30750, USA (706)
820-2531                                         http://seerockcity.com/?utm_source=
    gmb&utm_medium=organic Chattanooga\n
Tennessee Aquarium 35.055823 -85.311065            1 Broad St, Chattanooga, TN 37402, USA (423)
    265-0695
http://www.tnaqua.org/ Chattanooga\n                    Lookout Mountain Incline Railway
    35.009546 -85.328564    3917 St Elmo Ave, Chattanooga, TN 37409, USA (423)
821-4224                                                        http://www.
    ridetheincline.com/ Chattanooga\n
Coolidge Park 35.060506 -85.306857           150 River St, Chattanooga, TN 37405, USA (423)
    643-6311
http://www.chattanooga.gov/public-works/parks/directory-of-park-facilities Chattanooga\n
                                Creative Discovery Museum 35.052700 -85.312280
321 Chestnut St, Chattanooga, TN 37402, USA (423) 756-2738
                                                                http://www.
    cdmfun.org/
Chattanooga\n                              Chattanooga Ghost Tours Inc 35.051231 -85.309057
          57 E 5th St, Chattanooga, TN 37402, USA (423) 800-5998
https://chattanoogaghosttours.com/ Chattanooga\n                            Ross's
    Landing 35.056741 -85.310302    201 Riverfront Pkwy, Chattanooga, TN 37402,
USA (423) 643-6311                            http://www.chattanooga.gov/public-works/
    parks/directory-of-park-facilities Chattanooga\n
Hunter Museum of American Art 35.055894 -85.306405    10 Bluff View Ave, Chattanooga, TN 37403,
    USA (423) 267-0968
http://www.huntermuseum.org/ Chattanooga\n                          Walnut Street
    Bridge 35.058330 -85.307296         1 Walnut St, Chattanooga, TN 37403, USA
(423) 643-6096                            http://www.chattanooga.gov/public-works/parks
    /directory-of-park-facilities Chattanooga\n
```

Tennessee Riverpark 35.095938 -85.245311        4301 Amnicola Hwy, Chattanooga, TN 37402, USA (423)
    209-5370
http://parks.hamiltontn.gov/ Chattanooga\n            Moccasin Bend National Archeological
    District 35.053024 -85.328656 301 Manufacturers Rd, Chattanooga, TN 37405, USA
(423) 648-5623            https://www.nps.gov/chch/learn/historyculture/moccasin-bend-
    national-archeological-district.htm Chattanooga\n
Ruby Falls 35.019084 -85.339355        1720 Scenic Hwy, Chattanooga, TN 37409, USA (423) 821-2544
http://www.rubyfalls.com/ Chattanooga\n                    Cravens House
    35.013782 -85.341495 Point Park Visitor Ctr, Lookout Mountain, TN 37350, USA
(423) 821-7786                    https://www.nps.gov/
    places/cravenshouselom.htm Chattanooga\n
Williams Echo Dome 35.054215 -85.304836        430 E 2nd St, Chattanooga, TN 37403, USA
    Unknown
Unknown Chattanooga\n            Reflection Riding Arboretum & Nature Center 35.010130
    -85.364585    400 Garden Rd, Chattanooga, TN 37419, USA (423) 821-1160
https://www.reflectionriding.org/ Chattanooga\n                    Audubon
     Acres 34.999350 -85.180340    900 N Sanctuary Rd, Chattanooga, TN 37421,
USA (423) 892-1499                    http
    ://www.chattanoogaaudubon.org/ Chattanooga\n
Sculpture Fields at Montague Park 35.026175 -85.293624        1800 Polk St, Chattanooga, TN 37408,
    USA (423) 266-7288
http://www.sculpturefields.org/ Chattanooga\nBessie Smith Cultural Center and Chattanooga African
    American Museum 35.044634 -85.306467 200 E M L King Blvd, Chattanooga, TN 37403,
USA (423) 266-8658 http://www.bessiesmithcc.org/?utm_source=google&utm_medium=
    wix_google_business_profile&utm_campaign=39487993421723930355 Chattanooga\n
Wooden Horse 35.055836 -85.306713        10 Bluff View Ave, Chattanooga, TN 37403, USA   Unknown
Unknown Chattanooga"}, {'Description': 'Restaurants in Chattanooga', 'Content': "        Name
     Average Cost                Cuisines Aggregate
Rating    City\n        P.F. Chang's        33                Cafe, Bakery, BBQ,
    Seafood        4.2 Chattanooga\n        L'amandier
28            Cafe, BBQ, Mediterranean, Seafood        4.3 Chattanooga\n
    Anupam Sweet        82                Desserts, American, BBQ
3.4 Chattanooga\n        Nirula's        64                French, Desserts,
    Seafood        2.6 Chattanooga\n Anupam Sweets & Restaurant        62
Tea, Cafe, Desserts, Seafood        3.0 Chattanooga\n        Bisque Bakery        90
                French, Desserts, Seafood        3.9 Chattanooga\n
Nooba        93                Tea, Mexican, Bakery, Seafood        3.8 Chattanooga\n
        The Royal        39                French, Pizza,
Bakery, Fast Food        3.3 Chattanooga\n        Curry n Phulka        40
                Tea, Chinese, Fast Food        2.8 Chattanooga\n DCK-
Dana Choga's Kitchen        67                Cafe, Pizza, Mediterranean        3.5
    Chattanooga\n        Chaayos        54
Desserts, Cafe, BBQ, Chinese, Seafood        3.3 Chattanooga\n        Truffles
    53        Cafe, Bakery, BBQ, Fast Food, Chinese, American        3.2
Chattanooga\n        Liquid        15        Tea, Pizza, Bakery, Mediterranean,
    Seafood        4.0 Chattanooga\n        Tpot        14
Desserts, Tea, Pizza, Mexican, Cafe, Indian        0.0 Chattanooga\n   Sardar A Pure Meat Shop
        24                Bakery, Pizza, American
3.4 Chattanooga\n        Warehouse Cafe        90                Fast Food, Pizza,
    Seafood        3.7 Chattanooga\n        The Beer Cafe        62
Tea, Pizza, Fast Food        3.8 Chattanooga\n        Subway        66
                Tea, Cafe, Bakery, Desserts        2.4 Chattanooga\n
Indi-QUE        57                Tea, Italian, BBQ, Fast Food, Cafe        3.3 Chattanooga\
    nEssex Collections Patisserie        48
Chinese, BBQ, Fast Food        3.4 Chattanooga\n        Food Adda        100
    Desserts, Pizza, Fast Food, Mediterranean, Seafood        0.0 Chattanooga\n
Bikanervala        65                Cafe, Bakery        3.2
    Chattanooga\n        Havmor Ice Cream        75        Pizza, Bakery, Fast

Food, Chinese, Seafood          3.6 Chattanooga\n       Moti Mahal Delux        52
                Tea, Pizza, BBQ, Cafe, Mediterranean        2.7 Chattanooga\n
Aggarwal Sweet India        30                          Tea, Desserts          2.8
    Chattanooga\n         Jack Po!tato's        63              Tea,
Bakery, Mediterranean, Seafood          3.3 Chattanooga\n          Muradabadi       16
                Cafe, Pizza, Bakery, BBQ        3.0
Chattanooga\n        Vishal Restaurant        27                    Tea, Pizza, BBQ,
    Seafood          3.1 Chattanooga\n         Raj Sweets        69
Desserts, Tea, BBQ, Cafe, Mediterranean        2.8 Chattanooga\n      Punjabee's Darbar
        41          Tea, French, Bakery, Fast Food, Chinese        3.0
Chattanooga\n        Basil Tree        20              Bakery, Indian, American,
    Desserts        2.4 Chattanooga\n        Domino's Pizza        77
Bakery, Pizza, American, BBQ        2.8 Chattanooga\n     Pizza Hut Delivery        25
                Tea, Cafe, American, Indian        3.9 Chattanooga\n
Ichiban        87                  Mexican, Bakery, Desserts        3.9 Chattanooga\n
            Habibi        95
Cafe, BBQ, Seafood        3.7 Chattanooga\n        Cook Du Kdu        69 Pizza, Fast
    Food, Cafe, Indian, Mediterranean, Seafood        3.1 Chattanooga\n
Go Krazy        64                  French, BBQ, Fast Food        3.5 Chattanooga\
    n        Bentoya        41                  Tea, Fast
Food, Bakery, Seafood        3.8 Chattanooga\n        Burger's King        16
                Tea, Bakery, Desserts        4.1 Chattanooga"},
{'Description': 'Accommodations in Chattanooga', 'Content': '                       NAME
    price        room type                  house_rules minimum
nights maximum occupancy review rate number        city\n      Affordable Private Spacious Room in
    Brooklyn 790.0 Private room                  No parties
2.0            2            3.0 Chattanooga\n                          Sunny One Bedroom
    722.0 Entire home/apt        No pets & No children under 10        1.0
5            3.0 Chattanooga\nUpper West / Morningside Heights Apt, Near Subway 290.0 Entire
    home/apt              No visitors        3.0            3
3.0 Chattanooga\n        Sunny room+Pvte office in huge loft 728.0 Private room
                No parties        4.0            2            5.0
Chattanooga\n    Extra Cozy Room in Center of Williamsburg 1033.0 Private room
                No pets        1.0            1            1.0
Chattanooga\n   Luxury & Charm. Steps from Christopher Park! 301.0 Entire home/apt      No pets & No
    visitors & No parties        5.0            2            4.0
Chattanooga\n                  Fort Greene Room 722.0  Private room        No visitors &
    No children under 10        2.0            1            2.0
Chattanooga\n                      [Unicode Sequence Edited Out] 973.0 Entire home/apt
                No pets        10.0            6
4.0 Chattanooga\n    4 bdrm/2 bath apt. Central Pk, Columbia U. 564.0 Entire home/apt
                No visitors & No pets        30.0            4            3.0
Chattanooga\n     Cozy room in Bushwick- 15 min to the city 933.0 Private room
    No parties & No pets        1.0            2            2.0
Chattanooga\n          Modern apartment w/ gorgeous view 921.0 Private room
                No pets        3.0            2            1.0
Chattanooga\n     Artsy Private BR in Fort Greene Cumberland 727.0 Private room
                No visitors        30.0            1            3.0
Chattanooga\n        Studio Deluxe 1 - Wyndham Midtown 45 848.0 Private room No pets & No smoking
    & No children under 10        3.0            1            2.0
Chattanooga\n     Trendy and Stylish Downtown apartment !!! 110.0 Entire home/apt
                No pets        3.0            2            2.0
Chattanooga\n   Sonder | 180 Water | Luxurious 1BR + Rooftop 676.0 Entire home/apt
    No pets & No smoking        29.0            5            3.0
Chattanooga'}, {'Description': 'Flight from Houston to Nashville on 2022-03-21', 'Content': 'Flight
    Number Price DepTime ArrTime ActualElapsedTime FlightDate OriginCityName DestCityName
Distance\n   F3827247   241   14:22   16:10 1 hours 48 minutes 2022-03-21    Houston   Nashville
    657.0\n    F3827724   147   18:35   20:33 1 hours 58 minutes 2022-03-21

Houston    Nashville    657.0\n    F3827820    157    12:07    14:02 1 hours 55 minutes 2022-03-21
     Houston    Nashville    657.0\n    F3827930    305    07:36    09:41 2 hours 5
minutes 2022-03-21    Houston    Nashville    657.0\n    F3956407    264    11:38    13:15 1 hours 37
     minutes 2022-03-21    Houston    Nashville    670.0\n    F3956408    246
08:08    09:47 1 hours 39 minutes 2022-03-21    Houston    Nashville    670.0\n    F3956409    145
     17:36    19:12 1 hours 36 minutes 2022-03-21    Houston    Nashville    670.0\n
F3956532    225    07:14    08:52 1 hours 38 minutes 2022-03-21    Houston    Nashville    657.0\n
     F3956533    292    11:42    13:19 1 hours 37 minutes 2022-03-21    Houston
Nashville    657.0\n    F3956534    248    18:20    20:06 1 hours 46 minutes 2022-03-21    Houston
     Nashville    657.0\n    F4038756    272    10:11    11:52 1 hours 41 minutes
2022-03-21    Houston    Nashville    657.0\n    F4038966    149    16:33    18:13 1 hours 40 minutes
     2022-03-21    Houston    Nashville    657.0'}, {'Description': 'Self-driving
from Houston to Nashville', 'Content': 'self-driving, from Houston to Nashville, duration: 11 hours
     50 mins, distance: 1,253 km, cost: 62'}, {'Description': 'Taxi from Houston to
Nashville', 'Content': 'taxi, from Houston to Nashville, duration: 11 hours 50 mins, distance:
     1,253 km, cost: 1253'}, {'Description': 'Flight from Nashville to Knoxville on 2022-03-23',
'Content': 'There is no flight from Nashville to Knoxville on 2022-03-23.'}, {'Description': 'Self-
     driving from Nashville to Knoxville', 'Content': 'self-driving, from Nashville to
Knoxville, duration: 2 hours 42 mins, distance: 290 km, cost: 14'}, {'Description': 'Taxi from
     Nashville to Knoxville', 'Content': 'taxi, from Nashville to Knoxville, duration: 2 hours 42
mins, distance: 290 km, cost: 290'}, {'Description': 'Flight from Knoxville to Chattanooga on
     2022-03-25', 'Content': 'There is no flight from Knoxville to Chattanooga on 2022-03-25.'},
{'Description': 'Self-driving from Knoxville to Chattanooga', 'Content': 'self-driving, from
     Knoxville to Chattanooga, duration: 1 hour 41 mins, distance: 180 km, cost: 9'}, {'Description
     ':
'Taxi from Knoxville to Chattanooga', 'Content': 'taxi, from Knoxville to Chattanooga, duration: 1
     hour 41 mins, distance: 180 km, cost: 180'}, {'Description': 'Flight from Chattanooga to
Houston on 2022-03-27', 'Content': 'There is no flight from Chattanooga to Houston on
     2022-03-27.'}, {'Description': 'Self-driving from Chattanooga to Houston', 'Content': 'self-
     driving,
from Chattanooga to Houston, duration: 11 hours 47 mins, distance: 1,309 km, cost: 65'}, {'
     Description': 'Taxi from Chattanooga to Houston', 'Content': 'taxi, from Chattanooga to Houston
     ',
duration: 11 hours 47 mins, distance: 1,309 km, cost: 1309'}]
Query: Could you design a one-week travel itinerary for two, departing from Houston and touring
     three cities in Tennessee from March 21st to March 27th, 2022? Our budget is now $8,200. We
require accommodations that allow smoking and should ideally be private rooms. As for
     transportation, we would prefer not to self-drive.
Travel Plan:
[
    {
        "day": 1,
        "people_number": 2,
        "current_city": "from Houston to Nashville",
        "transportation": "Flight Number: F3827820, from Houston to Nashville, Departure Time:
            12:07, Arrival Time: 14:02",
        "breakfast": "-",
        "attraction": "Country Music Hall of Fame and Museum, Nashville",
        "lunch": "Cafe Arabelle, Nashville",
        "dinner": "Les 3 Brasseurs, Nashville",
        "accommodation": "FiDi Cozy room overlooking East River, Nashville"
    },
    {
        "day": 2,
        "people_number": 2,
        "current_city": "Nashville",
        "transportation": "-",
        "breakfast": "Burger Planet, Nashville",
        "attraction": "Johnny Cash Museum; Centennial Park, Nashville",

```
                "lunch": "Mamagoto, Nashville",
                "dinner": "Oh! Calcutta, Nashville",
                "accommodation": "FiDi Cozy room overlooking East River, Nashville"
            },
            {
                "day": 3,
                "people_number": 2,
                "current_city": "from Nashville to Knoxville",
                "transportation": "Taxi, from Nashville to Knoxville, Departure Time: 09:00, Arrival Time:
                    11:42, Duration: 2 hours 42 minutes, Cost: $290",
                "breakfast": "Cafe El Chico, Nashville",
                "attraction": "World's Fair Park, Knoxville",
                "lunch": "Ali Baba & 41 Dishes, Knoxville",
                "dinner": "Sky On 57, Knoxville",
                "accommodation": "Light-filled Room in Renovated Apt, Knoxville"
            },
            {
                "day": 4,
                "people_number": 2,
                "current_city": "Knoxville",
                "transportation": "-",
                "breakfast": "Chaat Corner, Knoxville",
                "attraction": "Ijams Nature Center; Sunsphere, Knoxville",
                "lunch": "Anand Sweets, Knoxville",
                "dinner": "La-Nawaab, Knoxville",
                "accommodation": "Light-filled Room in Renovated Apt, Knoxville"
            },
            {
                "day": 5,
                "people_number": 2,
                "current_city": "from Knoxville to Chattanooga",
                "transportation": "Taxi, from Knoxville to Chattanooga, Departure Time: 09:00, Arrival Time:
                    10:41, Duration: 1 hour 41 minutes, Cost: $180",
                "breakfast": "Anand Sweets, Knoxville",
                "attraction": "Tennessee Aquarium, Chattanooga",
                "lunch": "Anupam Sweet, Chattanooga",
                "dinner": "Bisque Bakery, Chattanooga",
                "accommodation": "Trendy and Stylish Downtown apartment !!!, Chattanooga"
            },
            {
                "day": 6,
                "people_number": 2,
                "current_city": "Chattanooga",
                "transportation": "-",
                "breakfast": "Anupam Sweets & Restaurant, Chattanooga",
                "attraction": "Rock City Gardens; Ruby Falls, Chattanooga",
                "lunch": "Chaayos, Chattanooga",
                "dinner": "L'amandier, Chattanooga",
                "accommodation": "Trendy and Stylish Downtown apartment !!!, Chattanooga"
            },
            {
                "day": 7,
                "people_number": 2,
                "current_city": "from Chattanooga to Houston",
                "transportation": "Taxi, from Chattanooga to Houston, Departure Time: 08:00, Arrival Time:
                    19:47, Duration: 11 hours 47 minutes, Cost: $1309",
                "breakfast": "Bisque Bakery, Chattanooga",
                "attraction": "Coolidge Park, Chattanooga",
                "lunch": "Nooba, Chattanooga",
```

```
            "dinner": "-",
            "accommodation": "-"
    }
]
```

By inspecting your plan we find the following issue. Please refine your plan according to the
     feedback below:
1. The room type should be private room :-> Trendy and Stylish Downtown apartment !!!
2. The restaurant in day 5 breakfast is repeated.
3. The accommodation Trendy and Stylish Downtown apartment !!!, Chattanooga do not obey the minumum
     nights rule.
4. The lunch in day 1 is invalid or not in the data provided.

Travel Plan (please only output the JSON string without explanatory information):

==============RESPONSE OF o1-preview=================

```
[
    {
        "day": 1,
        "people_number": 2,
        "current_city": "from Houston to Nashville",
        "transportation": "Flight Number: F3827820, from Houston to Nashville, Departure Time:
            12:07, Arrival Time: 14:02",
        "breakfast": "-",
        "attraction": "Country Music Hall of Fame and Museum, Nashville",
        "lunch": "Smoke House Deli, Nashville",
        "dinner": "Les 3 Brasseurs, Nashville",
        "accommodation": "FiDi Cozy room overlooking East River, Nashville"
    },
    {
        "day": 2,
        "people_number": 2,
        "current_city": "Nashville",
        "transportation": "-",
        "breakfast": "Burger Planet, Nashville",
        "attraction": "Johnny Cash Museum; Centennial Park, Nashville",
        "lunch": "Mamagoto, Nashville",
        "dinner": "Oh! Calcutta, Nashville",
        "accommodation": "FiDi Cozy room overlooking East River, Nashville"
    },
    {
        "day": 3,
        "people_number": 2,
        "current_city": "from Nashville to Knoxville",
        "transportation": "Taxi, from Nashville to Knoxville, Departure Time: 09:00, Arrival Time:
            11:42, Duration: 2 hours 42 minutes, Cost: $290",
        "breakfast": "Cafe El Chico, Nashville",
        "attraction": "World's Fair Park, Knoxville",
        "lunch": "Ali Baba & 41 Dishes, Knoxville",
        "dinner": "Sky On 57, Knoxville",
        "accommodation": "Light-filled Room in Renovated Apt, Knoxville"
    },
    {
        "day": 4,
        "people_number": 2,
        "current_city": "Knoxville",
        "transportation": "-",
        "breakfast": "Chaat Corner, Knoxville",
```

```
            "attraction": "Ijams Nature Center; Sunsphere, Knoxville",
            "lunch": "Anand Sweets, Knoxville",
            "dinner": "La-Nawaab, Knoxville",
            "accommodation": "Light-filled Room in Renovated Apt, Knoxville"
    },
    {
        "day": 5,
        "people_number": 2,
        "current_city": "from Knoxville to Chattanooga",
        "transportation": "Taxi, from Knoxville to Chattanooga, Departure Time: 09:00, Arrival Time:
                10:41, Duration: 1 hour 41 minutes, Cost: $180",
        "breakfast": "Burger Planet, Knoxville",
        "attraction": "Tennessee Aquarium, Chattanooga",
        "lunch": "Anupam Sweet, Chattanooga",
        "dinner": "Bisque Bakery, Chattanooga",
        "accommodation": "Affordable Private Spacious Room in Brooklyn, Chattanooga"
    },
    {
        "day": 6,
        "people_number": 2,
        "current_city": "Chattanooga",
        "transportation": "-",
        "breakfast": "Anupam Sweets & Restaurant, Chattanooga",
        "attraction": "Rock City Gardens; Ruby Falls, Chattanooga",
        "lunch": "Chaayos, Chattanooga",
        "dinner": "L'amandier, Chattanooga",
        "accommodation": "Affordable Private Spacious Room in Brooklyn, Chattanooga"
    },
    {
        "day": 7,
        "people_number": 2,
        "current_city": "from Chattanooga to Houston",
        "transportation": "Taxi, from Chattanooga to Houston, Departure Time: 08:00, Arrival Time:
                19:47, Duration: 11 hours 47 minutes, Cost: $1309",
        "breakfast": "Bisque Bakery, Chattanooga",
        "attraction": "Coolidge Park, Chattanooga",
        "lunch": "Nooba, Chattanooga",
        "dinner": "-",
        "accommodation": "-"
    }
]
```

# R   Prompts for Trip Planning

## R.1   First Iteration

```
You are an expert at planning trips. You are given a few constraints regarding the cities to visit
    and the durations of staying at each city. You are also given the flight information between
    the cities('and' suggests flights are possible both ways and 'from - to -' suggests
one way flights). You can travel to any city only once.
Produce only one plan, strictly adhering to the format shown in the examples below, without any
    extra information. Your answer shold start with 'SOLUTION:'.
Here are a few example tasks and solutions:

TASK: You plan to visit 10 European cities for 21 days in total. You only take direct flights to
    commute between cities. You plan to stay in Vienna for 3 days. You want to spend 5 days in
    Frankfurt. You want to spend 2 days in Oslo. You are going to attend a wedding in Oslo
```

between day 20 and day 21. You want to spend 3 days in Prague. You would like to visit Valencia for
    2 days. You want to meet a friend in Valencia between day 17 and day 18. You plan to stay in
    Dubrovnik for 2 days. You would like to visit Edinburgh for 5 days. From day 1 to
day 5, there is a annual show you want to attend in Edinburgh. You plan to stay in London for 2
    days. You plan to visit relatives in London between day 12 and day 13. You would like to visit
    Munich for 3 days. You would like to meet your friends at Munich between day 18 and
day 20 to tour together. You would like to visit Budapest for 3 days.

Here are the cities that have direct flights:
Valencia and Munich, Vienna and Munich, Vienna and Valencia, London and Budapest, London and Oslo,
    Edinburgh and Budapest, Frankfurt and Budapest, Frankfurt and London, Prague and Oslo,
    Edinburgh and Oslo, Edinburgh and Munich, Prague and Munich, London and Prague, Edinburgh
and London, Edinburgh and Frankfurt, Dubrovnik and Munich, Dubrovnik and Vienna, Munich and Oslo,
    Dubrovnik and Oslo, Budapest and Munich, Frankfurt and Prague, Vienna and London, Frankfurt and
        Vienna, Frankfurt and Oslo, Frankfurt and Munich, Vienna and Oslo, Vienna and
Prague, Budapest and Oslo, Budapest and Prague, London and Valencia, London and Munich, Frankfurt
    and Dubrovnik, Prague and Valencia, Frankfurt and Valencia, Edinburgh and Prague, Vienna and
    Budapest.

Find a trip plan of visiting the cities for 21 days by taking direct flights to commute between
    them.
SOLUTION: Here is the trip plan for visiting the 10 European cities for 21 days:

**Day 1-5:** Arriving in Edinburgh and visit Edinburgh for 5 days.
**Day 5:** Fly from Edinburgh to Frankfurt.
**Day 5-9:** Visit Frankfurt for 5 days.
**Day 9:** Fly from Frankfurt to Dubrovnik.
**Day 9-10:** Visit Dubrovnik for 2 days.
**Day 10:** Fly from Dubrovnik to Vienna.
**Day 10-12:** Visit Vienna for 3 days.
**Day 12:** Fly from Vienna to London.
**Day 12-13:** Visit London for 2 days.
**Day 13:** Fly from London to Budapest.
**Day 13-15:** Visit Budapest for 3 days.
**Day 15:** Fly from Budapest to Prague.
**Day 15-17:** Visit Prague for 3 days.
**Day 17:** Fly from Prague to Valencia.
**Day 17-18:** Visit Valencia for 2 days.
**Day 18:** Fly from Valencia to Munich.
**Day 18-20:** Visit Munich for 3 days.
**Day 20:** Fly from Munich to Oslo.
**Day 20-21:** Visit Oslo for 2 days.

TASK: You plan to visit 10 European cities for 22 days in total. You only take direct flights to
    commute between cities. You want to spend 4 days in Helsinki. You plan to stay in Vilnius for 2
    days. You plan to stay in Istanbul for 5 days. From day 17 to day 21, there is a
annual show you want to attend in Istanbul. You want to spend 2 days in Hamburg. You plan to stay
    in Porto for 2 days. During day 11 and day 12, you have to attend a conference in Porto. You
    want to spend 4 days in Athens. You would like to visit Prague for 2 days. You plan
to visit relatives in Prague between day 13 and day 14. You want to spend 2 days in Frankfurt. You
    want to spend 3 days in Krakow. You want to spend 5 days in Munich. You would like to meet your
    friends at Munich between day 1 and day 5 to tour together.

Here are the cities that have direct flights:
from Krakow to Vilnius, Helsinki and Hamburg, Hamburg and Athens, Munich and Frankfurt, Hamburg and
    Porto, Munich and Istanbul, Prague and Athens, Frankfurt and Athens, Munich and Athens, Munich
    and Prague, from Vilnius to Munich, Hamburg and Istanbul, Frankfurt and

Istanbul, Munich and Krakow, Munich and Hamburg, Munich and Helsinki, Prague and Istanbul,
     Frankfurt and Vilnius, Helsinki and Istanbul, Athens and Vilnius, Krakow and Frankfurt,
     Helsinki and Frankfurt, Porto and Frankfurt, Frankfurt and Prague, Istanbul and Vilnius, Krakow
and Istanbul, Krakow and Prague, Munich and Porto, Helsinki and Vilnius, Helsinki and Prague, Porto
     and Istanbul, Hamburg and Frankfurt, Krakow and Helsinki, Athens and Istanbul.

Find a trip plan of visiting the cities for 22 days by taking direct flights to commute between
     them.
SOLUTION: Here is the trip plan for visiting the 10 European cities for 22 days:

**Day 1-5:** Arriving in Munich and visit Munich for 5 days.
**Day 5:** Fly from Munich to Krakow.
**Day 5-7:** Visit Krakow for 3 days.
**Day 7:** Fly from Krakow to Helsinki.
**Day 7-10:** Visit Helsinki for 4 days.
**Day 10:** Fly from Helsinki to Hamburg.
**Day 10-11:** Visit Hamburg for 2 days.
**Day 11:** Fly from Hamburg to Porto.
**Day 11-12:** Visit Porto for 2 days.
**Day 12:** Fly from Porto to Frankfurt.
**Day 12-13:** Visit Frankfurt for 2 days.
**Day 13:** Fly from Frankfurt to Prague.
**Day 13-14:** Visit Prague for 2 days.
**Day 14:** Fly from Prague to Athens.
**Day 14-17:** Visit Athens for 4 days.
**Day 17:** Fly from Athens to Istanbul.
**Day 17-21:** Visit Istanbul for 5 days.
**Day 21:** Fly from Istanbul to Vilnius.
**Day 21-22:** Visit Vilnius for 2 days.

TASK: You plan to visit 10 European cities for 23 days in total. You only take direct flights to
     commute between cities. You would like to visit Stuttgart for 2 days. You would like to visit
     Split for 2 days. You are going to attend a wedding in Split between day 22 and day
23. You would like to visit Vienna for 5 days. You want to spend 4 days in Madrid. You plan to stay
     in Athens for 2 days. You would like to visit London for 3 days. During day 8 and day 10, you
     have to attend a conference in London. You plan to stay in Paris for 3 days. You
want to meet a friend in Paris between day 10 and day 12. You plan to stay in Reykjavik for 2 days.
     You have to attend a workshop in Reykjavik between day 16 and day 17. You want to spend 4 days
     in Seville. You want to spend 5 days in Milan. You would like to meet your
friends at Milan between day 17 and day 21 to tour together.

Here are the cities that have direct flights:
Athens and Paris, Athens and Vienna, Madrid and Vienna, Madrid and Split, Vienna and Stuttgart,
     Paris and Milan, London and Vienna, London and Milan, Paris and Reykjavik, Athens and London,
     from Reykjavik to Stuttgart, Seville and Milan, from Reykjavik to Madrid, London and
Stuttgart, Milan and Stuttgart, Vienna and Reykjavik, Athens and Split, Athens and Milan, Madrid
     and Athens, Madrid and London, Paris and Split, London and Paris, Seville and Vienna, Vienna
     and Milan, Athens and Stuttgart, Madrid and Paris, Seville and Madrid, from Reykjavik
to Athens, Vienna and Split, London and Split, Stuttgart and Split, Seville and Paris, Paris and
     Stuttgart, Reykjavik and Milan, London and Reykjavik, Madrid and Milan, Paris and Vienna, Milan
      and Split.

Find a trip plan of visiting the cities for 23 days by taking direct flights to commute between
     them.
SOLUTION: Here is the trip plan for visiting the 10 European cities for 23 days:

**Day 1-4:** Arriving in Seville and visit Seville for 4 days.
**Day 4:** Fly from Seville to Madrid.
**Day 4-7:** Visit Madrid for 4 days.

**Day 7:** Fly from Madrid to Athens.
**Day 7-8:** Visit Athens for 2 days.
**Day 8:** Fly from Athens to London.
**Day 8-10:** Visit London for 3 days.
**Day 10:** Fly from London to Paris.
**Day 10-12:** Visit Paris for 3 days.
**Day 12:** Fly from Paris to Vienna.
**Day 12-16:** Visit Vienna for 5 days.
**Day 16:** Fly from Vienna to Reykjavik.
**Day 16-17:** Visit Reykjavik for 2 days.
**Day 17:** Fly from Reykjavik to Milan.
**Day 17-21:** Visit Milan for 5 days.
**Day 21:** Fly from Milan to Stuttgart.
**Day 21-22:** Visit Stuttgart for 2 days.
**Day 22:** Fly from Stuttgart to Split.
**Day 22-23:** Visit Split for 2 days.

TASK: You plan to visit 10 European cities for 25 days in total. You only take direct flights to
     commute between cities. You would like to visit Berlin for 2 days. You would like to visit Riga
      for 2 days. During day 5 and day 6, you have to attend a conference in Riga. You
want to spend 3 days in Barcelona. You would like to visit Lyon for 4 days. You would like to meet
     your friends at Lyon between day 8 and day 11 to tour together. You plan to stay in Naples for
     2 days. You plan to stay in Venice for 5 days. You want to spend 5 days in
Helsinki. You plan to visit relatives in Helsinki between day 21 and day 25. You plan to stay in
     Rome for 5 days. You want to spend 3 days in Vilnius. You want to spend 3 days in Amsterdam.
     You are going to attend a wedding in Amsterdam between day 19 and day 21.

Here are the cities that have direct flights:
Berlin and Amsterdam, Rome and Helsinki, Rome and Lyon, Naples and Amsterdam, Riga and Barcelona,
     Rome and Venice, Riga and Amsterdam, from Riga to Vilnius, Barcelona and Berlin, Rome and
     Barcelona, Rome and Amsterdam, Barcelona and Venice, Berlin and Helsinki, Amsterdam and
Helsinki, Vilnius and Helsinki, Rome and Berlin, from Rome to Riga, Barcelona and Amsterdam, Venice
      and Naples, Barcelona and Lyon, Naples and Berlin, Barcelona and Helsinki, Venice and Helsinki
     , Barcelona and Naples, Vilnius and Amsterdam, Venice and Amsterdam, Lyon and
Venice, Naples and Helsinki, Riga and Berlin, Rome and Naples, Venice and Berlin, Riga and Helsinki
     , Berlin and Vilnius, Lyon and Amsterdam.

Find a trip plan of visiting the cities for 25 days by taking direct flights to commute between
     them.
SOLUTION: Here is the trip plan for visiting the 10 European cities for 25 days:

**Day 1-5:** Arriving in Rome and visit Rome for 5 days.
**Day 5:** Fly from Rome to Riga.
**Day 5-6:** Visit Riga for 2 days.
**Day 6:** Fly from Riga to Barcelona.
**Day 6-8:** Visit Barcelona for 3 days.
**Day 8:** Fly from Barcelona to Lyon.
**Day 8-11:** Visit Lyon for 4 days.
**Day 11:** Fly from Lyon to Venice.
**Day 11-15:** Visit Venice for 5 days.
**Day 15:** Fly from Venice to Naples.
**Day 15-16:** Visit Naples for 2 days.
**Day 16:** Fly from Naples to Berlin.
**Day 16-17:** Visit Berlin for 2 days.
**Day 17:** Fly from Berlin to Vilnius.
**Day 17-19:** Visit Vilnius for 3 days.
**Day 19:** Fly from Vilnius to Amsterdam.
**Day 19-21:** Visit Amsterdam for 3 days.
**Day 21:** Fly from Amsterdam to Helsinki.

**Day 21-25:** Visit Helsinki for 5 days.

TASK: You plan to visit 10 European cities for 27 days in total. You only take direct flights to
    commute between cities. You would like to visit Prague for 5 days. You have to attend a
    workshop in Prague between day 7 and day 11. You would like to visit Helsinki for 3 days.
You are going to attend a wedding in Helsinki between day 15 and day 17. You plan to stay in
    Tallinn for 2 days. You want to meet a friend in Tallinn between day 6 and day 7. You want to
    spend 4 days in Edinburgh. You want to spend 5 days in Paris. You want to spend 4 days
in Vienna. You plan to stay in Lisbon for 5 days. From day 11 to day 15, there is a annual show you
    want to attend in Lisbon. You want to spend 4 days in Budapest. You plan to stay in Lyon for 2
    days. You plan to stay in Brussels for 2 days. You would like to meet your
friends at Brussels between day 1 and day 2 to tour together.

Here are the cities that have direct flights:
Prague and Lyon, Brussels and Lisbon, Helsinki and Budapest, Vienna and Lyon, Paris and Tallinn,
    Brussels and Prague, Brussels and Helsinki, Prague and Helsinki, Brussels and Vienna, Brussels
    and Budapest, Lisbon and Budapest, Tallinn and Helsinki, Brussels and Paris,
Brussels and Tallinn, Lisbon and Lyon, Prague and Lisbon, Paris and Prague, Helsinki and Edinburgh,
    Prague and Edinburgh, Tallinn and Prague, Brussels and Lyon, Paris and Lisbon, Helsinki and
    Vienna, Paris and Helsinki, Paris and Budapest, Edinburgh and Budapest, Brussels
and Edinburgh, Lisbon and Vienna, Paris and Lyon, Lisbon and Helsinki, Prague and Vienna, Paris and
    Vienna, Prague and Budapest, Paris and Edinburgh, Budapest and Vienna.

Find a trip plan of visiting the cities for 27 days by taking direct flights to commute between
    them.
SOLUTION: Here is the trip plan for visiting the 10 European cities for 27 days:

**Day 1-2:** Arriving in Brussels and visit Brussels for 2 days.
**Day 2:** Fly from Brussels to Paris.
**Day 2-6:** Visit Paris for 5 days.
**Day 6:** Fly from Paris to Tallinn.
**Day 6-7:** Visit Tallinn for 2 days.
**Day 7:** Fly from Tallinn to Prague.
**Day 7-11:** Visit Prague for 5 days.
**Day 11:** Fly from Prague to Lisbon.
**Day 11-15:** Visit Lisbon for 5 days.
**Day 15:** Fly from Lisbon to Helsinki.
**Day 15-17:** Visit Helsinki for 3 days.
**Day 17:** Fly from Helsinki to Edinburgh.
**Day 17-20:** Visit Edinburgh for 4 days.
**Day 20:** Fly from Edinburgh to Budapest.
**Day 20-23:** Visit Budapest for 4 days.
**Day 23:** Fly from Budapest to Vienna.
**Day 23-26:** Visit Vienna for 4 days.
**Day 26:** Fly from Vienna to Lyon.
**Day 26-27:** Visit Lyon for 2 days.

Query:
You plan to visit 10 European cities for 25 days in total. You only take direct flights to commute
    between cities. You plan to stay in Berlin for 4 days. You have to attend a workshop in Berlin
    between day 12 and day 15. You would like to visit Prague for 2 days. You plan to
stay in Stuttgart for 5 days. You would like to visit Manchester for 3 days. You want to spend 2
    days in Nice. You would like to visit Reykjavik for 2 days. You would like to visit Florence
    for 3 days. You plan to stay in Vilnius for 5 days. You would like to meet your
friends at Vilnius between day 15 and day 19 to tour together. You plan to stay in Oslo for 4 days.
    You would like to visit Dubrovnik for 4 days. You plan to visit relatives in Dubrovnik between
    day 1 and day 4.

Here are the cities that have direct flights:
from Reykjavik to Stuttgart, Manchester and Stuttgart, Nice and Berlin, Oslo and Prague, Stuttgart
    and Berlin, Manchester and Nice, Reykjavik and Oslo, Reykjavik and Prague, Manchester and
    Prague, Reykjavik and Berlin, Dubrovnik and Manchester, Manchester and Oslo,
Manchester and Berlin, Prague and Florence, Berlin and Vilnius, Dubrovnik and Oslo, Nice and Oslo,
    Berlin and Oslo, Nice and Reykjavik, Vilnius and Oslo.

Find a trip plan of visiting the cities for 25 days by taking direct flights to commute between
    them.

===============RESPONSE OF o1-preview==================
SOLUTION: Here is the trip plan for visiting the 10 European cities for 25 days:

**Day 1-4:** Arriving in Dubrovnik and visit Dubrovnik for 4 days.

**Day 4:** Fly from Dubrovnik to Manchester.

**Day 4-7:** Visit Manchester for 3 days.

**Day 7:** Fly from Manchester to Stuttgart.

**Day 7-12:** Visit Stuttgart for 5 days.

**Day 12:** Fly from Stuttgart to Berlin.

**Day 12-15:** Visit Berlin for 4 days. (Attend workshop between day 12 and day 15)

**Day 15:** Fly from Berlin to Vilnius.

**Day 15-19:** Visit Vilnius for 5 days. (Meet friends between day 15 and day 19)

**Day 19:** Fly from Vilnius to Oslo via Berlin.

**Day 19-23:** Visit Oslo for 4 days.

**Day 23:** Fly from Oslo to Reykjavik.

**Day 23-25:** Visit Reykjavik for 2 days.

===============FEEDBACK FROM METACONTROLLER=================
Number of cities in plan is 7, expected 10

### R.2 Back Prompt

You are an expert at planning trips. Fix the below given trip schedule such that it meets as many
    specified constraints as possible. You are given a few constraints regarding the cities to
    visit and the durations of staying at each city. You are also given the flight
information between the cities.
Produce only one plan, strictly adhering to the format shown in the examples below, without any
    extra information.

The produced plan will be parsed using the following python code:

```python
def parse_response(response: str):
  """Parse the response.

  Returns a parsed plan in a list of (city, stay_days) tuples.
```

```
  Args:
    response: Raw response from the model.

  Returns:
    Structured plan after parsing.
  """
  pattern_visit = r'\d+-\d+'
  pattern_flight = r'.*Day (\d+).*from (\w+) to (\w+)'
  pattern_days = r'European cities for (\d+) days'

  days, flights, flight_days = [], [], []
  total_days = None
  for piece in response.split('\n'):
    days_match = re.findall(pattern_days, piece)
    if days_match:
      total_days = int(days_match[0])

    visit_match = re.findall(pattern_visit, piece)
    if visit_match:
      days.append(visit_match[0])
      end_day = int(visit_match[0].split('-')[1])
      # Reach the end of the plan, stop to avoid parsing alternative plans.
      if end_day == total_days:
        break
    flight_match = re.findall(pattern_flight, piece)
    if flight_match:
      flights.append(flight_match[0])

  visit_cities, parsed_plan = [], []
  for flight_day, begin_city, end_city in flights:
    flight_days.append(int(flight_day))
    if not visit_cities:
      visit_cities.append(begin_city)
      visit_cities.append(end_city)
    else:
      visit_cities.append(end_city)

  if not days or not flights or not visit_cities:
    return []
  last_day = int(days[-1].split('-')[1])
  flight_days = [1] + flight_days +
  for i, visit_city in enumerate(visit_cities):
    city_stay = flight_days - flight_days + 1
    parsed_plan.append((visit_city, city_stay))

  return parsed_plan
```

Here are a few example tasks and solutions:

TASK: You plan to visit 10 European cities for 21 days in total. You only take direct flights to commute between cities. You plan to stay in Vienna for 3 days. You want to spend 5 days in Frankfurt. You want to spend 2 days in Oslo. You are going to attend a wedding in Oslo between day 20 and day 21. You want to spend 3 days in Prague. You would like to visit Valencia for 2 days. You want to meet a friend in Valencia between day 17 and day 18. You plan to stay in Dubrovnik for 2 days. You would like to visit Edinburgh for 5 days. From day 1 to

day 5, there is a annual show you want to attend in Edinburgh. You plan to stay in London for 2
    days. You plan to visit relatives in London between day 12 and day 13. You would like to visit
    Munich for 3 days. You would like to meet your friends at Munich between day 18 and
day 20 to tour together. You would like to visit Budapest for 3 days.

Here are the cities that have direct flights:
Valencia and Munich, Vienna and Munich, Vienna and Valencia, London and Budapest, London and Oslo,
    Edinburgh and Budapest, Frankfurt and Budapest, Frankfurt and London, Prague and Oslo,
    Edinburgh and Oslo, Edinburgh and Munich, Prague and Munich, London and Prague, Edinburgh
and London, Edinburgh and Frankfurt, Dubrovnik and Munich, Dubrovnik and Vienna, Munich and Oslo,
    Dubrovnik and Oslo, Budapest and Munich, Frankfurt and Prague, Vienna and London, Frankfurt and
     Vienna, Frankfurt and Oslo, Frankfurt and Munich, Vienna and Oslo, Vienna and
Prague, Budapest and Oslo, Budapest and Prague, London and Valencia, London and Munich, Frankfurt
    and Dubrovnik, Prague and Valencia, Frankfurt and Valencia, Edinburgh and Prague, Vienna and
    Budapest.

Find a trip plan of visiting the cities for 21 days by taking direct flights to commute between
    them.
SOLUTION: Here is the trip plan for visiting the 10 European cities for 21 days:

**Day 1-5:** Arriving in Edinburgh and visit Edinburgh for 5 days.
**Day 5:** Fly from Edinburgh to Frankfurt.
**Day 5-9:** Visit Frankfurt for 5 days.
**Day 9:** Fly from Frankfurt to Dubrovnik.
**Day 9-10:** Visit Dubrovnik for 2 days.
**Day 10:** Fly from Dubrovnik to Vienna.
**Day 10-12:** Visit Vienna for 3 days.
**Day 12:** Fly from Vienna to London.
**Day 12-13:** Visit London for 2 days.
**Day 13:** Fly from London to Budapest.
**Day 13-15:** Visit Budapest for 3 days.
**Day 15:** Fly from Budapest to Prague.
**Day 15-17:** Visit Prague for 3 days.
**Day 17:** Fly from Prague to Valencia.
**Day 17-18:** Visit Valencia for 2 days.
**Day 18:** Fly from Valencia to Munich.
**Day 18-20:** Visit Munich for 3 days.
**Day 20:** Fly from Munich to Oslo.
**Day 20-21:** Visit Oslo for 2 days.

TASK: You plan to visit 10 European cities for 22 days in total. You only take direct flights to
    commute between cities. You want to spend 4 days in Helsinki. You plan to stay in Vilnius for 2
     days. You plan to stay in Istanbul for 5 days. From day 17 to day 21, there is a
annual show you want to attend in Istanbul. You want to spend 2 days in Hamburg. You plan to stay
    in Porto for 2 days. During day 11 and day 12, you have to attend a conference in Porto. You
    want to spend 4 days in Athens. You would like to visit Prague for 2 days. You plan
to visit relatives in Prague between day 13 and day 14. You want to spend 2 days in Frankfurt. You
    want to spend 3 days in Krakow. You want to spend 5 days in Munich. You would like to meet your
     friends at Munich between day 1 and day 5 to tour together.

Here are the cities that have direct flights:
from Krakow to Vilnius, Helsinki and Hamburg, Hamburg and Athens, Munich and Frankfurt, Hamburg and
     Porto, Munich and Istanbul, Prague and Athens, Frankfurt and Athens, Munich and Athens, Munich
     and Prague, from Vilnius to Munich, Hamburg and Istanbul, Frankfurt and
Istanbul, Munich and Krakow, Munich and Hamburg, Munich and Helsinki, Prague and Istanbul,
    Frankfurt and Vilnius, Helsinki and Istanbul, Athens and Vilnius, Krakow and Frankfurt,
    Helsinki and Frankfurt, Porto and Frankfurt, Frankfurt and Prague, Istanbul and Vilnius, Krakow
and Istanbul, Krakow and Prague, Munich and Porto, Helsinki and Vilnius, Helsinki and Prague, Porto
     and Istanbul, Hamburg and Frankfurt, Krakow and Helsinki, Athens and Istanbul.

Find a trip plan of visiting the cities for 22 days by taking direct flights to commute between
    them.
SOLUTION: Here is the trip plan for visiting the 10 European cities for 22 days:

**Day 1-5:** Arriving in Munich and visit Munich for 5 days.
**Day 5:** Fly from Munich to Krakow.
**Day 5-7:** Visit Krakow for 3 days.
**Day 7:** Fly from Krakow to Helsinki.
**Day 7-10:** Visit Helsinki for 4 days.
**Day 10:** Fly from Helsinki to Hamburg.
**Day 10-11:** Visit Hamburg for 2 days.
**Day 11:** Fly from Hamburg to Porto.
**Day 11-12:** Visit Porto for 2 days.
**Day 12:** Fly from Porto to Frankfurt.
**Day 12-13:** Visit Frankfurt for 2 days.
**Day 13:** Fly from Frankfurt to Prague.
**Day 13-14:** Visit Prague for 2 days.
**Day 14:** Fly from Prague to Athens.
**Day 14-17:** Visit Athens for 4 days.
**Day 17:** Fly from Athens to Istanbul.
**Day 17-21:** Visit Istanbul for 5 days.
**Day 21:** Fly from Istanbul to Vilnius.
**Day 21-22:** Visit Vilnius for 2 days.

TASK: You plan to visit 10 European cities for 23 days in total. You only take direct flights to
    commute between cities. You would like to visit Stuttgart for 2 days. You would like to visit
    Split for 2 days. You are going to attend a wedding in Split between day 22 and day
23. You would like to visit Vienna for 5 days. You want to spend 4 days in Madrid. You plan to stay
    in Athens for 2 days. You would like to visit London for 3 days. During day 8 and day 10, you
    have to attend a conference in London. You plan to stay in Paris for 3 days. You
want to meet a friend in Paris between day 10 and day 12. You plan to stay in Reykjavik for 2 days.
    You have to attend a workshop in Reykjavik between day 16 and day 17. You want to spend 4 days
    in Seville. You want to spend 5 days in Milan. You would like to meet your
friends at Milan between day 17 and day 21 to tour together.

Here are the cities that have direct flights:
Athens and Paris, Athens and Vienna, Madrid and Vienna, Madrid and Split, Vienna and Stuttgart,
    Paris and Milan, London and Vienna, London and Milan, Paris and Reykjavik, Athens and London,
    from Reykjavik to Stuttgart, Seville and Milan, from Reykjavik to Madrid, London and
Stuttgart, Milan and Stuttgart, Vienna and Reykjavik, Athens and Split, Athens and Milan, Madrid
    and Athens, Madrid and London, Paris and Split, London and Paris, Seville and Vienna, Vienna
    and Milan, Athens and Stuttgart, Madrid and Paris, Seville and Madrid, from Reykjavik
to Athens, Vienna and Split, London and Split, Stuttgart and Split, Seville and Paris, Paris and
    Stuttgart, Reykjavik and Milan, London and Reykjavik, Madrid and Milan, Paris and Vienna, Milan
    and Split.

Find a trip plan of visiting the cities for 23 days by taking direct flights to commute between
    them.
SOLUTION: Here is the trip plan for visiting the 10 European cities for 23 days:

**Day 1-4:** Arriving in Seville and visit Seville for 4 days.
**Day 4:** Fly from Seville to Madrid.
**Day 4-7:** Visit Madrid for 4 days.
**Day 7:** Fly from Madrid to Athens.
**Day 7-8:** Visit Athens for 2 days.
**Day 8:** Fly from Athens to London.
**Day 8-10:** Visit London for 3 days.
**Day 10:** Fly from London to Paris.

**Day 10-12:** Visit Paris for 3 days.
**Day 12:** Fly from Paris to Vienna.
**Day 12-16:** Visit Vienna for 5 days.
**Day 16:** Fly from Vienna to Reykjavik.
**Day 16-17:** Visit Reykjavik for 2 days.
**Day 17:** Fly from Reykjavik to Milan.
**Day 17-21:** Visit Milan for 5 days.
**Day 21:** Fly from Milan to Stuttgart.
**Day 21-22:** Visit Stuttgart for 2 days.
**Day 22:** Fly from Stuttgart to Split.
**Day 22-23:** Visit Split for 2 days.

TASK: You plan to visit 10 European cities for 25 days in total. You only take direct flights to commute between cities. You would like to visit Berlin for 2 days. You would like to visit Riga for 2 days. During day 5 and day 6, you have to attend a conference in Riga. You want to spend 3 days in Barcelona. You would like to visit Lyon for 4 days. You would like to meet your friends at Lyon between day 8 and day 11 to tour together. You plan to stay in Naples for 2 days. You plan to stay in Venice for 5 days. You want to spend 5 days in Helsinki. You plan to visit relatives in Helsinki between day 21 and day 25. You plan to stay in Rome for 5 days. You want to spend 3 days in Vilnius. You want to spend 3 days in Amsterdam. You are going to attend a wedding in Amsterdam between day 19 and day 21.

Here are the cities that have direct flights:
Berlin and Amsterdam, Rome and Helsinki, Rome and Lyon, Naples and Amsterdam, Riga and Barcelona, Rome and Venice, Riga and Amsterdam, from Riga to Vilnius, Barcelona and Berlin, Rome and Barcelona, Rome and Amsterdam, Barcelona and Venice, Berlin and Helsinki, Amsterdam and Helsinki, Vilnius and Helsinki, Rome and Berlin, from Rome to Riga, Barcelona and Amsterdam, Venice and Naples, Barcelona and Lyon, Naples and Berlin, Barcelona and Helsinki, Venice and Helsinki, Barcelona and Naples, Vilnius and Amsterdam, Venice and Amsterdam, Lyon and Venice, Naples and Helsinki, Riga and Berlin, Rome and Naples, Venice and Berlin, Riga and Helsinki, Berlin and Vilnius, Lyon and Amsterdam.

Find a trip plan of visiting the cities for 25 days by taking direct flights to commute between them.
SOLUTION: Here is the trip plan for visiting the 10 European cities for 25 days:

**Day 1-5:** Arriving in Rome and visit Rome for 5 days.
**Day 5:** Fly from Rome to Riga.
**Day 5-6:** Visit Riga for 2 days.
**Day 6:** Fly from Riga to Barcelona.
**Day 6-8:** Visit Barcelona for 3 days.
**Day 8:** Fly from Barcelona to Lyon.
**Day 8-11:** Visit Lyon for 4 days.
**Day 11:** Fly from Lyon to Venice.
**Day 11-15:** Visit Venice for 5 days.
**Day 15:** Fly from Venice to Naples.
**Day 15-16:** Visit Naples for 2 days.
**Day 16:** Fly from Naples to Berlin.
**Day 16-17:** Visit Berlin for 2 days.
**Day 17:** Fly from Berlin to Vilnius.
**Day 17-19:** Visit Vilnius for 3 days.
**Day 19:** Fly from Vilnius to Amsterdam.
**Day 19-21:** Visit Amsterdam for 3 days.
**Day 21:** Fly from Amsterdam to Helsinki.
**Day 21-25:** Visit Helsinki for 5 days.

TASK: You plan to visit 10 European cities for 27 days in total. You only take direct flights to commute between cities. You would like to visit Prague for 5 days. You have to attend a workshop in Prague between day 7 and day 11. You would like to visit Helsinki for 3 days.

You are going to attend a wedding in Helsinki between day 15 and day 17. You plan to stay in
    Tallinn for 2 days. You want to meet a friend in Tallinn between day 6 and day 7. You want to
    spend 4 days in Edinburgh. You want to spend 5 days in Paris. You want to spend 4 days
in Vienna. You plan to stay in Lisbon for 5 days. From day 11 to day 15, there is a annual show you
    want to attend in Lisbon. You want to spend 4 days in Budapest. You plan to stay in Lyon for 2
    days. You plan to stay in Brussels for 2 days. You would like to meet your
friends at Brussels between day 1 and day 2 to tour together.

Here are the cities that have direct flights:
Prague and Lyon, Brussels and Lisbon, Helsinki and Budapest, Vienna and Lyon, Paris and Tallinn,
    Brussels and Prague, Brussels and Helsinki, Prague and Helsinki, Brussels and Vienna, Brussels
    and Budapest, Lisbon and Budapest, Tallinn and Helsinki, Brussels and Paris,
Brussels and Tallinn, Lisbon and Lyon, Prague and Lisbon, Paris and Prague, Helsinki and Edinburgh,
     Prague and Edinburgh, Tallinn and Prague, Brussels and Lyon, Paris and Lisbon, Helsinki and
     Vienna, Paris and Helsinki, Paris and Budapest, Edinburgh and Budapest, Brussels
and Edinburgh, Lisbon and Vienna, Paris and Lyon, Lisbon and Helsinki, Prague and Vienna, Paris and
     Vienna, Prague and Budapest, Paris and Edinburgh, Budapest and Vienna.

Find a trip plan of visiting the cities for 27 days by taking direct flights to commute between
    them.
SOLUTION: Here is the trip plan for visiting the 10 European cities for 27 days:

**Day 1-2:** Arriving in Brussels and visit Brussels for 2 days.
**Day 2:** Fly from Brussels to Paris.
**Day 2-6:** Visit Paris for 5 days.
**Day 6:** Fly from Paris to Tallinn.
**Day 6-7:** Visit Tallinn for 2 days.
**Day 7:** Fly from Tallinn to Prague.
**Day 7-11:** Visit Prague for 5 days.
**Day 11:** Fly from Prague to Lisbon.
**Day 11-15:** Visit Lisbon for 5 days.
**Day 15:** Fly from Lisbon to Helsinki.
**Day 15-17:** Visit Helsinki for 3 days.
**Day 17:** Fly from Helsinki to Edinburgh.
**Day 17-20:** Visit Edinburgh for 4 days.
**Day 20:** Fly from Edinburgh to Budapest.
**Day 20-23:** Visit Budapest for 4 days.
**Day 23:** Fly from Budapest to Vienna.
**Day 23-26:** Visit Vienna for 4 days.
**Day 26:** Fly from Vienna to Lyon.
**Day 26-27:** Visit Lyon for 2 days.

Query:
You plan to visit 10 European cities for 25 days in total. You only take direct flights to commute
    between cities. You plan to stay in Berlin for 4 days. You have to attend a workshop in Berlin
    between day 12 and day 15. You would like to visit Prague for 2 days. You plan to
stay in Stuttgart for 5 days. You would like to visit Manchester for 3 days. You want to spend 2
    days in Nice. You would like to visit Reykjavik for 2 days. You would like to visit Florence
    for 3 days. You plan to stay in Vilnius for 5 days. You would like to meet your
friends at Vilnius between day 15 and day 19 to tour together. You plan to stay in Oslo for 4 days.
     You would like to visit Dubrovnik for 4 days. You plan to visit relatives in Dubrovnik between
     day 1 and day 4.

Here are the cities that have direct flights:
from Reykjavik to Stuttgart, Manchester and Stuttgart, Nice and Berlin, Oslo and Prague, Stuttgart
    and Berlin, Manchester and Nice, Reykjavik and Oslo, Reykjavik and Prague, Manchester and
    Prague, Reykjavik and Berlin, Dubrovnik and Manchester, Manchester and Oslo,

Manchester and Berlin, Prague and Florence, Berlin and Vilnius, Dubrovnik and Oslo, Nice and Oslo,
     Berlin and Oslo, Nice and Reykjavik, Vilnius and Oslo.

Find a trip plan of visiting the cities for 25 days by taking direct flights to commute between
     them.

Incorrect plan in natural language:
SOLUTION: Here is the trip plan for visiting the 10 European cities for 25 days:

**Day 1-4:** Arriving in Dubrovnik and visit Dubrovnik for 4 days.

**Day 4:** Fly from Dubrovnik to Manchester.

**Day 4-7:** Visit Manchester for 3 days.

**Day 7:** Fly from Manchester to Stuttgart.

**Day 7-12:** Visit Stuttgart for 5 days.

**Day 12:** Fly from Stuttgart to Berlin.

**Day 12-15:** Visit Berlin for 4 days. (Attend workshop between day 12 and day 15)

**Day 15:** Fly from Berlin to Vilnius.

**Day 15-19:** Visit Vilnius for 5 days. (Meet friends between day 15 and day 19)

**Day 19:** Fly from Vilnius to Oslo via Berlin.

**Day 19-23:** Visit Oslo for 4 days.

**Day 23:** Fly from Oslo to Reykjavik.

**Day 23-25:** Visit Reykjavik for 2 days.

Incorrect plan in parsed format:
[['Dubrovnik', 4], ['Manchester', 4], ['Stuttgart', 6], ['Berlin', 4], ['Vilnius', 5], ['Oslo', 5],
     ['Reykjavik', 3]]

Errors with the above plan:
Number of cities in plan is 7, expected 10

Give the corrected plan in natural language such that it can be parsed by the above python code.
Strictly adhere to the format shown in the examples below, without any extra information.

Corrected plan:

===============RESPONSE OF o1-preview==================
SOLUTION: Here is the trip plan for visiting the 10 European cities for 25 days:

**Day 1-4:** Arriving in Dubrovnik and visit Dubrovnik for 4 days. (Visit relatives between day 1
     and day 4)

**Day 4:** Fly from Dubrovnik to Manchester.

**Day 4-6:** Visit Manchester for 3 days.

**Day 6:** Fly from Manchester to Nice.

**Day 6-7:** Visit Nice for 2 days.

**Day 7:** Fly from Nice to Reykjavik.

**Day 7-8:** Visit Reykjavik for 2 days.

**Day 8:** Fly from Reykjavik to Stuttgart.

**Day 8-10:** Visit Stuttgart for 3 days.

**Day 10:** Fly from Stuttgart to Prague.

**Day 10-11:** Visit Prague for 2 days.

**Day 11:** Fly from Prague to Berlin.

**Day 11-15:** Visit Berlin for 4 days. (Attend workshop between day 12 and day 15)

**Day 15:** Fly from Berlin to Vilnius.

**Day 15-19:** Visit Vilnius for 5 days. (Meet friends between day 15 and day 19)

**Day 19:** Fly from Vilnius to Oslo.

**Day 19-22:** Visit Oslo for 4 days.

**Day 22:** Fly from Oslo to Florence via Prague.

**Day 22-25:** Visit Florence for 3 days.

## S   Prompts for Calendar Scheduling

### S.1   First Iteration

You are an expert at scheduling meetings. You are given a few constraints on the existing schedule
    of each participant, the meeting duration, and possibly some preferences on the meeting time.
    Note there exists a solution that works with existing schedule of every
participant. Produce only one meeting time, strictly adhering to the format shown in the examples
    below, without providing any extra information. Here are a few example tasks and solutions:

TASK: You need to schedule a meeting for George, Steven, Aaron, Patrick and Cynthia for half an
    hour between the work hours of 9:00 to 17:00 on Monday.

Here are the existing schedules for everyone during the day:
Georgehas no meetings the whole day.
Steven is free the entire day.
Aaron has blocked their calendar on Monday during 9:00 to 10:00, 11:30 to 12:00, 15:30 to 17:00;
Patrick has blocked their calendar on Monday during 9:00 to 9:30, 10:00 to 11:00, 11:30 to 12:00,
    12:30 to 14:00, 15:00 to 15:30;
Cynthia is busy on Monday during 9:00 to 9:30, 10:30 to 11:30, 12:30 to 14:30, 15:00 to 16:30;

Aaron can not meet on Monday after 12:30. Find a time that works for everyone's schedule and
    constraints.
SOLUTION: Here is the proposed time: Monday, 12:00 - 12:30

TASK: You need to schedule a meeting for Elizabeth, Eugene, Nancy, Justin and Roy for half an hour
    between the work hours of 9:00 to 17:00 on Monday.

Here are the existing schedules for everyone during the day:

Elizabeth's calendar is wide open the entire day.
Eugene has blocked their calendar on Monday during 12:00 to 12:30, 13:30 to 14:00, 15:00 to 16:00;
Nancy has meetings on Monday during 10:30 to 11:00, 12:00 to 13:00, 14:00 to 14:30, 15:00 to 16:00;
Justin has meetings on Monday during 9:30 to 10:00, 10:30 to 11:30, 12:00 to 12:30, 13:00 to 13:30,
    14:30 to 17:00;
Roy is busy on Monday during 10:00 to 11:00, 11:30 to 12:30, 13:30 to 14:00, 14:30 to 16:00;

Find a time that works for everyone's schedule and constraints.
SOLUTION: Here is the proposed time: Monday, 9:00 - 9:30

TASK: You need to schedule a meeting for Mason, Linda, Barbara, Roger and Lisa for half an hour
    between the work hours of 9:00 to 17:00 on Monday.

Here are the existing schedules for everyone during the day:
Masonhas no meetings the whole day.
Linda is busy on Monday during 9:30 to 11:00, 12:00 to 12:30, 13:00 to 13:30, 16:30 to 17:00;
Barbara has blocked their calendar on Monday during 9:00 to 9:30, 12:00 to 13:00, 15:30 to 16:00;
Roger has meetings on Monday during 9:00 to 9:30, 13:00 to 14:30, 15:00 to 17:00;
Lisa has blocked their calendar on Monday during 9:00 to 10:00, 10:30 to 11:30, 12:00 to 12:30,
    13:00 to 14:30, 15:00 to 17:00;

Mason would rather not meet on Monday after 13:30. Find a time that works for everyone's schedule
    and constraints.
SOLUTION: Here is the proposed time: Monday, 11:30 - 12:00

TASK: You need to schedule a meeting for Thomas, Olivia, Doris, Amber and Ronald for half an hour
    between the work hours of 9:00 to 17:00 on Monday.

Here are the existing schedules for everyone during the day:
Thomas has meetings on Monday during 12:00 to 12:30, 16:30 to 17:00;
Olivia has blocked their calendar on Monday during 10:30 to 11:30, 13:00 to 14:00, 16:00 to 17:00;
Doris is busy on Monday during 13:30 to 14:00, 16:30 to 17:00;
Amber is busy on Monday during 10:00 to 10:30, 12:00 to 13:00, 14:00 to 15:00, 15:30 to 17:00;
Ronald has meetings on Monday during 9:30 to 11:30, 12:00 to 13:00, 13:30 to 14:00, 15:00 to 15:30;

Ronald would rather not meet on Monday before 11:30. Find a time that works for everyone's schedule
    and constraints.
SOLUTION: Here is the proposed time: Monday, 11:30 - 12:00

TASK: You need to schedule a meeting for Patrick, Kathleen, Frank, Patricia and Carolyn for half an
    hour between the work hours of 9:00 to 17:00 on Monday.

Here are the existing schedules for everyone during the day:
Patrick is free the entire day.
Kathleen has meetings on Monday during 9:30 to 10:00, 15:00 to 15:30;
Frank has blocked their calendar on Monday during 11:00 to 11:30, 15:30 to 17:00;
Patricia is busy on Monday during 9:00 to 12:00, 14:30 to 15:00, 16:00 to 16:30;
Carolyn has meetings on Monday during 9:00 to 9:30, 11:00 to 12:30, 13:00 to 17:00;

Find a time that works for everyone's schedule and constraints.
SOLUTION: Here is the proposed time: Monday, 12:30 - 13:00

Query:
TASK: You need to schedule a meeting for Stephen, Elijah, William, Jeremy and Timothy for half an
    hour between the work hours of 9:00 to 17:00 on Monday.

Here are the existing schedules for everyone during the day:
Stephen is free the entire day.
Elijah is busy on Monday during 9:00 to 9:30, 12:30 to 13:00, 14:30 to 15:00, 16:00 to 16:30;

William is busy on Monday during 9:30 to 10:00, 15:30 to 16:00;
Jeremy is busy on Monday during 9:00 to 9:30, 10:00 to 12:00, 13:00 to 15:00, 15:30 to 17:00;
Timothy is busy on Monday during 10:00 to 10:30, 11:30 to 14:30, 15:30 to 16:00;

Find a time that works for everyone's schedule and constraints.
SOLUTION:

==============RESPONSE OF o1-mini================
SOLUTION: Here is the proposed time: Monday, 15:00 - 15:30

