# OpenReview forum: "A Systematic Evaluation of the Planning and Scheduling Abilities of the Reasoning Model o1"
_TMLR — Accepted by TMLR_

### Review · Reviewer_tRV3 · 2024-12-20

**Summary Of Contributions:**

This paper evaluates and compares o1 models, as sota LRMs, with previous LLMs. It shows that while LRMs can generally outperform previous models, they require significantly more cost while still failing in extremely difficult instances.

**Audience:**

Yes

**Claims And Evidence:**

Yes

**Requested Changes:**

The authors are encouraged to discuss more, diverse works in LLM planning/scheduling.

**Strengths And Weaknesses:**

Strengths:
The task (planning/scheduling) itself is important for the ml community.

It is important to understand the limitations of a new model paradigm--reasoning model.

The paper is easy to follow.

The experiments are comprehensive and interesting.

Weaknesses:
The literature review is a little bit insufficient. This paper seems to only be discussing a limited amount of papers from very similar research veins, while there are a lot of other LLM planning papers worth citing.

---

> ### Author Response · Authors · 2025-01-09
>
> Thank you for your encouraging review. We are gratified that you found our study to be comprehensive and interesting.
>
> Regarding your comment about discussing more diverse works in LLM planning/scheduling, we have expanded our discussion (at the end of page 4) to include works from multiple veins of LLM planning research, covering various tool augmentations, diverse domains, prompt engineering techniques, and relevant compound systems.

---

> > ### Comment · Reviewer_tRV3 · 2025-01-11
> >
> > Thanks to the authors for the revision! I have no concerns now.
> >
> > One minor issue: citation of Katz et al. misses year at the end of page 5.

---

> > > ### Author Response · Authors · 2025-01-13
> > >
> > > Thank you for your prompt response. We are glad that your concerns have been addressed.
> > >
> > > > **One minor issue: citation of Katz et al. misses year at the end of page 5.**
> > >
> > > Thank you for noticing it! We will fix the issue with that citation.

---

### Review · Reviewer_wCEF · 2024-12-24

**Summary Of Contributions:**

This article investigates the reasoning capabilities of GPT o1 preview and GPT o1 mini on some reasoning and planning benchmarks including PlanBench, Sokoban, and planning problems. The authors conclude that the GPT o1-series models surpass the performance of their predecessors by a large margin but at the cost of an increase in price and efficiency.

**Audience:**

Yes

**Claims And Evidence:**

Yes

**Requested Changes:**

- It is not suggested that abbreviations be put in titles, especially when those are from new concepts.
- I don't think it's a good practice to call GPT o1 "large reasoning models". First, I don't think LRM is a widely-recognized term. In addition, previous works have adopted the same term for completely different concepts. This may cause confusion.
- Minor typos such as quotation marks ("" in LaTeX source), "LLM models" (page 14), etc.

**Strengths And Weaknesses:**

The presentation of this article is largely clear, and the results included could be considered comprehensive.

However, my biggest concern about this article is its (lack of) contribution to the community. People have a consensus that o1 is much better at reasoning and planning. Although this article provides concrete numbers on some of the benchmarks, it is hard to imagine how these numbers can provide us with additional insights into the usage of such models other than those we are already aware of. The analysis, unfortunately, largely builds upon guesswork and therefore is not robust.

---

> ### Author Response · Authors · 2025-01-09
>
> We thank the reviewer for their valuable comments. Here are some clarifications for the concerns raised.
>
> > **However, my biggest concern about this article is its (lack of) contribution to the community. People have a consensus that o1 is much better at reasoning and planning. Although this article provides concrete numbers on some of the benchmarks, it is hard to imagine how these numbers can provide us with additional insights into the usage of such models other than those we are already aware of.**
>
> Our paper presents a systematic evaluation of the cost-quality performance of o1 on a spectrum of planning instances, and also shows how o1’s accuracy can be further improved–with guarantees by combining with other inference-time scaling techniques such as LLM-Modulo.  We strongly believe that such independent and objective third party analyses are critical especially given the lack of technical documentation of the inner workings of o1, and the influence of commercial interests which cannot be discounted. We are not sure how to interpret the “People have consensus that o1 is much better at reasoning and planning” statement by the reviewer. That the general gestalt impressions of capabilities can be quite wrong has been amply shown by the initial optimism and the later realization of the planning (in)capabilities of vanilla LLMs. The latter happened only when the models were subjected to systematic third party analyses. We are certainly not aware of any clear, quantitative, and comprehensive analyses of the planning capabilities of o1.
>
> In the same vein, while o1 is understood to be “costlier”, there certainly was no quantitative understanding of how much costlier it can be—especially not on problems requiring planning capabilities. Our evaluation provides significant insight into this and highlights the importance of comparing not just raw performance but also taking into account other trade-offs, especially when there is the potential for creating similar or lower cost alternatives that use smaller or cheaper models in augmented or modulo systems.
>
> > **The analysis, unfortunately, largely builds upon guesswork and therefore is not robust.**
>
> While we did present speculations on the inner workings of o1--detailed in the appendix rather than the main paper--we would like to clarify that these speculations have no bearing on our claims about o1's performance in planning tasks. Given this, we are a little confused as to what the reviewer considers "not robust" in our analysis, as we believe our main arguments are well-supported by the direct and copious empirical evidence presented in the paper.
>
> > **It is not suggested that abbreviations be put in titles, especially when those are from new concepts.  I don't think it's a good practice to call GPT o1 "large reasoning models". First, I don't think LRM is a widely-recognized term. In addition, previous works have adopted the same term for completely different concepts. This may cause confusion.**
>
> We would like to point out that OpenAI themselves have used “Reasoning Models” terminology to refer to o1 models [1]. Additionally, we believe that the term “large reasoning models” is consistent with the terminology the community has lately been using to refer to such models [2, 3, 4]. That said, we acknowledge that the abbreviation “LRM” is not yet widely recognized. To address this, we have primarily used it as a short form within the paper and have revised the title to avoid using abbreviations directly.
>
> [1] https://openai.com/index/introducing-openai-o1-preview/
>
> [2] Nasir, M. U., James, S., & Togelius, J. GameTraversalBenchmark: Evaluating Planning Abilities Of Large Language Models Through Traversing 2D Game Maps. In The Thirty-eight Conference on Neural Information Processing Systems Datasets and Benchmarks Track.
>
> [3] Zhao, Y., Yin, H., Zeng, B., Wang, H., Shi, T., Lyu, C., ... & Zhang, K. (2024). Marco-o1: Towards open reasoning models for open-ended solutions. arXiv preprint arXiv:2411.14405.
>
> [4] https://github.com/SimpleBerry/LLaMA-O1
>
>
> > **Minor typos such as quotation marks ("" in LaTeX source), "LLM models" (page 14), etc.**
>
> Thank you for pointing them out. We have fixed these typos in the revised version.

---

### Review · Reviewer_Bwjz · 2024-12-29

**Summary Of Contributions:**

The authors evaluate o1-preview and o1-mini on planning and scheduling tasks, with and without obfuscated names. They find the o1 series to outperform GPT-4 but to deteriorate with plan length or problem size.

They point out that it is important to evaluate the accuracy vs cost trade-off, and perform experiments with GPT-4 (to generate candidate solutions) coupled with a guaranteed external verifier as well as with o1-preview in this setup, finding that the advantage of o1 against GPT-4 vanishes at at least one accuracy-cost trade-off point.

**Audience:**

Yes

**Broader Impact Concerns:**

None.

**Claims And Evidence:**

No

**Requested Changes:**

The title is not appropriate. The paper is not about LLM planning (this has been discussed in other works). I would just keep the second part: "A Systematic Evaluation of the Planning and Scheduling Abilities of o1".

Is OAI actually making the claim that o1 is "generally capable of tackling procedural reasoning taks"? I have not seen that.
Either document your claim or remove it.

Make a note in section 5 that the modulo approach can only be used when a verifier exists and is computationally efficient, whereas o1 can be used even without a verifier.

Modify the language to address the issues above, avoiding claims not justified in this paper.

For example rephrase "it cannot plan robustly when faced with harder instances" into something like "it outperforms LLMs but results degrade when faced with longer or larger instances".

It is also not very neutral to write "o1 is still fallible and without guarantees" because OAI did not claim that it was infallible or that it came with guarantees. Please soften the tone more generally.

The authors sometimes talk about o1 and sometimes about o1-preview and o1-mini. It may be good to clarify that they use the term o1 generically to talk about these two versions (since another version, o1 full, came out recently).

Define the IPC abbreviation before using it.

Define q-values.

Typos:
"did not rejection" missing "do" in the middle
"language typically models" should be "language models typically"

**Strengths And Weaknesses:**

In summary, useful evaluations but inappropriate, unjustified or even misleading statements, which can all be fixed.

Some of the results are not surprising, e.g., that o1 outperforms GPT-4 on most of the tasks, since this is coherent with OAI's own evaluations. It is also not surprising that results deteriorate with plan length, but it is good to quantify this on these tasks.

Some of the results are much more significant, in particular on the importance of measuring and taking into account the accuracy-cost trade-off.

In general, the language used in the paper is highly depreciative of o1 and OpenAI, which is not appropriate for TMLR.
Several value judgments are made, without justification, which is not acceptable, including calling the state-of-the-art LLMs 'approximate retrieval' systems doing 'mildly-generating pattern matching'. In other places, the authors seem to imply that OpenAI made claims about o1 which I have not seen previously, e.g., that it performs robust and general reasoning and planning. They also claim that Chain-of-Thought did not improve on planning tasks, but fail to mention that these systems do improve on other reasoning tasks.

In fact, my recollection from OAI's documentation is that they claimed advances in reasoning, but not notable advances in planning.
It is true that planning is a special form of reasoning, but it may require particular skills that were not particularly developed in the training of o1. OAI did not claim that o1 was a general reasoning system. Be careful to not give that impression and other similar inappropriately phrased statements.

Also, one should be careful when comparing a generic natural-language based system like o1 and a formal and somewhat task-specific search algorithm. In cases where one can use the latter with success, there is indeed no reason to use the former, but one should also point out where systems like o1 can be used but not the classical planning or reasoning systems.

---

> ### Author Response · Authors · 2025-01-09
>
> We thank the reviewer for their detailed feedback. Below, we provide responses to the requested changes.
>
> > **The title is not appropriate. The paper is not about LLM planning (this has been discussed in other works). I would just keep the second part: "A Systematic Evaluation of the Planning and Scheduling Abilities of o1".**
>
> We thank the reviewer for their suggestion. We agree that our primary contribution is presenting a systematic and comprehensive evaluation of the o1 models on planning. We have modified the title as suggested by the reviewer to reflect that.
>
> > **Is OAI actually making the claim that o1 is "generally capable of tackling procedural reasoning taks"? I have not seen that. Either document your claim or remove it.**
>
> We would like to clarify that OpenAI has positioned o1 models as capable of handling complex reasoning tasks and planning is one such task. The following quotes from OpenAI’s o1 release site [1,2] support this claim:
>
> - “We are introducing OpenAI o1, a new large language model trained with reinforcement learning to perform complex reasoning.”
>
> - “OpenAI o1, our reasoning model designed to handle complex multi-step tasks with advanced accuracy, is rolling out to developers on usage tier 5 in the API.”
> We have documented this in the revised version.
>
> We’d also like to point out that, planning is included in the tasks that o1 is claimed to have significantly improved upon compared to LLMs and is given as a specific example in their launch tweet thread [3].
>
> [1] https://openai.com/index/learning-to-reason-with-llms/
>
> [2] https://openai.com/index/o1-and-new-tools-for-developers/
>
> [3] https://x.com/polynoamial/status/1834280155730043108?lang=en
>
>
> > **Make a note in section 5 that the modulo approach can only be used when a verifier exists and is computationally efficient, whereas o1 can be used even without a verifier.**
>
> We would like to highlight that the second paragraph of Section 5 already makes this distinction, noting that we augment o1 with external verifiers to ensure soundness guarantees.
>
> > **Modify the language to address the issues above, avoiding claims not justified in this paper. For example rephrase "it cannot plan robustly when faced with harder instances" into something like "it outperforms LLMs but results degrade when faced with longer or larger instances".**
>
> We agree with your point and would like to clarify that the paper already acknowledges this nuance. The full sentence that the phrase is quoted from is “While o1 is a stride in the direction of general-purpose, expressive planning systems, our results have shown that it cannot plan robustly when faced with harder instances.” (Last paragraph of page 9)
>
> > **It is also not very neutral to write "o1 is still fallible and without guarantees" because OAI did not claim that it was infallible or that it came with guarantees. Please soften the tone more generally.**
>
> We agree with the reviewer that OpenAI has not claimed o1 to be infallible or to come with guarantees. We did not attribute any such claims to OpenAI in the paper as we believe that the value of our study is not so much to critique OpenAI but to give more realistic expectations to potential users and to provide an outside evaluation. In particular, we take the position that guarantees are necessary for any safety-critical system, and thus that any planning system (formal or general) must provide some level of them to be truly useful, especially in real-world applications. Therefore, we believe it’s essential to highlight the model's limitations and the value of modulo systems. That said, we acknowledge the importance of adopting a softer tone. In the revised version, we have adjusted the language to ensure a more neutral tone while still addressing this important limitation.
>
> > **The authors sometimes talk about o1 and sometimes about o1-preview and o1-mini. It may be good to clarify that they use the term o1 generically to talk about these two versions (since another version, o1 full, came out recently).**
>
> Thank you for pointing this out. We have made this clear in the revised version.
>
> > **Definitions and Typos**
>
> Thank you for pointing these out. We have provided the required definitions and fixed the typos.

---

> > ### Comment · Reviewer_Bwjz · 2025-01-29
> > **Fine with me**
> >
> > The modified paper is fine with me as far as  publication in TMLR goes.

---

### Decision · Action_Editor_Cgzc · 2025-03-09

**Recommendation:** Accept with minor revision

**Comment:**

Even though one of the reviewers (wCEF) questioned the value of analyzing a closed-source model, which is already outdated before the review process is complete, I think there is some value in publishing this work to shed some light on the planning ability of these models. I think TMLR is a perfect venue for such work.

Please incorporate all the changes you made during the rebuttal to the final version of the paper. Also, please try to be more precise in your claims and add a discussion on how this study is still valuable while there are new models coming out and making the previous generation models outdated in a matter of weeks or months.

**Audience:**

This paper will be of interest to the TMLR audience since this is a timely work that evaluates the planning and scheduling capabilities of OpenAI O1 models.

**Claims And Evidence:**

The paper systematically evaluates the planning and scheduling capabilities of OpenAI O1 models and shows that while they achieve better performance than other models, the performance gap with GPT4 reduces as GPT-4 is provided with an external verifier with similar compute and cost. This is a timely and systematic study of a state-of-the-art proprietary model.

The reviewers initially had concerns about the paper's grand claims. However, based on their feedback, the authors have toned down the claims.